# Deciphering spatial domains from spatial multi-omics with SpatialGlue

Yahui Long [1], Kok Siong Ang [1], Raman Sethi [2], Sha Liao[3,4], Yang Heng[3,4], Lynn van Olst [5], Shuchen Ye[2], Chengwei Zhong[1], Hang Xu [2], Di Zhang[6], Immanuel Kwok[7], Nazihah Husna[7,8,9], Min Jian[3,10], Lai Guan Ng [11], Ao Chen [3,4,12], Nicholas R. J. Gascoigne [8,9,13], David Gate[5], Rong Fan [6], Xun Xu [3] & Jinmiao Chen [1,2,8,9,14] ✉

Advances in spatial omics technologies now allow multiple types of data to be acquired from the same tissue slice. To realize the full potential of such data, we need spatially informed methods for data integration. Here, we introduce SpatialGlue, a graph neural network model with a dual-attention mechanism that deciphers spatial domains by intra-omics integration of spatial location and omics measurement followed by cross-omics integration. We demonstrated SpatialGlue on data acquired from different tissue types using different technologies, including spatial epigenome–transcriptome and transcriptome–proteome modalities. Compared to other methods, SpatialGlue captured more anatomical details and more accurately resolved spatial domains such as the cortex layers of the brain. Our method also identified cell types like spleen macrophage subsets located at three different zones that were not available in the original data annotations. SpatialGlue scales well with data size and can be used to integrate three modalities. Our spatial multi-omics analysis tool combines the information from complementary omics modalities to obtain a holistic view of cellular and tissue properties.

Spatial transcriptomics is the next major development in analyzing biological samples since the advent of single-cell transcriptomics. Currently, spatial technologies are expanding to spatial multi-omics with the simultaneous profiling of different omics on a single tissue section. These technologies can be roughly divided into two categories, sequencing based and imaging based. Sequencing-based techniques include DBiT-seq[1], spatial-CITE-seq[2], spatial assay for transposase-accessible chromatin and RNA using sequencing (spatial ATAC–RNA-seq) and CUT&Tag-RNA-seq[3], SPOTS[4], SM-Omics[5], Stereo-CITE-seq[6], spatial RNA-TCR-seq[7] and 10x Genomics Xenium[8], while imaging-based techniques include DNA seqFISH+[9], DNA-MERFISH-based DNA and RNA profiling[10], MERSCOPE[11] and Nanostring CosMx[12]. With these technologies, we can now acquire multiple complementary views of each cell within their spatial context. This offers the potential for developing deeper insights into cellular and emergent tissue properties.

To fully utilize spatial multi-omics data to construct a coherent picture of the tissue under study, spatially aware integration of heterogeneous data modalities is required. Such multi-omics data integration poses a major challenge as different modalities have feature counts that can vary enormously (for example, number of proteins versus transcripts measured) and possess different statistical distributions. This challenge is deepened when integrating spatial information with feature counts within each data modality. To our knowledge, there is no tool designed specifically for spatial multi-omics acquired from the same tissue section. Existing methods either are unimodal or do not use spatial information, except for one tool with functionality for spatial multi-omics integration, MEFISTO, which has only been previously demonstrated on single-cell multi-omics or spatial transcriptomics separately. For the non-spatial multi-omics data integration methods, a wide range of algorithms are available. These include Seurat WNN[13],

MOFA+[14], StabMap[15], totalVI[16], MultiVI[17] and scMM[18]. Moreover, some of these methods are designed for specific data modalities, which can be restrictive. For example, totalVI is designed for CITE-seq data of RNA and protein modalities, while MultiVI is optimized for gene expression and chromatin accessibility. For spatial omics tools, examples include STAGATE[19], SpaGCN[20] and GraphST[21], which integrate spatial information and single omics modalities. Such single omics methods can only handle spatial multi-omics data by concatenating the feature count data from different omics modalities. This approach assumes that features across different omics have the same importance, which may not be true. Therefore, tools specifically tailored for spatial multi-omics data are needed to handle the challenges of integrating spatial multi-omics data for downstream analyses. In particular, we need new methods that are capable of spatially aware cross-omics integration.

Here we present SpatialGlue for spatial multi-omics analysis. Specifically, SpatialGlue is a spatially aware method that integrates multiple spatial omics data modalities, acquired from the same tissue slice, to decipher spatial domains of tissue samples at a higher spatial resolution. SpatialGlue uses graph neural networks to learn a low-dimensional embedding for each data modality, followed by data integration across modalities. To integrate spatial information with individual omics data and integrate across omics, we adopted a dual-attention aggregation mechanism to adaptively capture the importance of different modalities, resulting in more accurate integration. We first tested SpatialGlue on simulated and experimentally acquired data of the human lymph node with ground truth to benchmark its performance with other methods. SpatialGlue achieved better quantitative performance than the other methods and captured more anatomical details. We then tested SpatialGlue and competing methods on integrating the spatial epigenome and transcriptome of the mouse brain, and spatial transcriptome and proteome data acquired from the mouse thymus and spleen. SpatialGlue leveraged the epigenome–transcriptome data to differentiate more cortex layers than the original data annotation, and the transcriptome–proteome to distinguish macrophage subsets within the spleen. These results highlight the power of multimodal spatial omics for analyzing biological complexity.

## Results

### SpatialGlue model structure

SpatialGlue deciphers spatial domains of tissue samples at a higher resolution by effectively integrating multi-omics modalities data with spatial information (Fig. 1a). SpatialGlue is a graph neural network (GNN)-based deep learning model (Fig. 1b). The input data to SpatialGlue can be feature matrices of segmented cells or capture locations (beads, voxels, pixels, bins or spots), with accompanying spatial coordinates. We refer to the cells and the capture locations as spots hereafter for brevity and not to restrict SpatialGlue to any specific technological platform or resolution. For data integration, SpatialGlue uses a dual-attention mechanism at two levels, within-modality spatial information and measurement feature integration first, and then between-modality multi-omics integration.

SpatialGlue first learns a low-dimensional embedding within each modality using the spatial coordinates and omics data. Within each modality, SpatialGlue constructs a spatial proximity graph and a feature similarity graph, which are used separately to encode the preprocessed feature count data into a common low-dimensional embedding space. Here the spatial proximity graph captures spatial relationships between spots, while the feature graph captures feature similarities. These constructed graphs possess unique semantic information that can be integrated to better capture cellular heterogeneity. However, the different graphs can contribute differential importance to each spot, posing a challenge to capture this difference. Therefore, we adopted a within-modality attention aggregation layer to adaptively integrate the spatial and feature graph-specific

representations and derive modality-specific representations. Specifically, the model learns graph-specific weights to assign importance to each graph. Similarly, the different omics modalities can have distinct and complementary contributions to each spot. Thus, we further designed a between-modality attention aggregation layer that learns modality-specific importance weights and adaptively integrates the modality-specific representations to generate the final cross-modality integrated latent representation. The learned weights illustrate the contribution of each modality to the learned latent representation of each spot and, consequently, the demarcation of different spatial domains or cell types. After obtaining SpatialGlue's integrated multi-omics representation, we can then use clustering to identify biologically relevant spatial domains, which consist of cells that are coherent spatially and across the measured omics. Such spatial domains can range from local clusters of distinct cell states to functionally distinct anatomical structures. We believe this attention-based approach enables more accurate integration than summation or concatenation of the feature matrices.

To evaluate the effectiveness of the proposed SpatialGlue model, we initially validated the importance of attention and other components with a series of ablation studies using simulated data (Supplementary Fig. 1; full experimental details are in the Supplementary Information). Subsequently, we characterized SpatialGlue's sensitivity to the number of neighbors and principal component analysis (PCA) dimensions of the input, and the number of GNN layers (Extended Data Fig. 1; full results are in the Supplementary Information).

### Benchmarking SpatialGlue and existing methods on simulated and experimental spatial multi-omics data

We first benchmarked SpatialGlue with competing methods using simulated data and experimentally acquired data with ground truth labels. With the ground truth available, we could assess performance with supervised metrics, namely homogeneity, mutual information, v measure, AMI (adjusted mutual information), NMI (normalized mutual information) and ARI (adjusted rand index). We generated a set of simulated data consisting of two modalities that contained unique and complementary information of the ground truth (Fig. 2a). Specifically, factors 1, 3 and 4 were determined by modality 1, while factor 2 was uniquely identified through modality 2. The modalities were designed to simulate the transcriptome and proteome, respectively, with the first modality following the zero-inflated negative binomial distribution and the second following the negative binomial distribution (Fig. 2b). For comparison, we tested seven competing methods: Seurat, totalVI, MultiVI, MOFA+, MEFISTO, scMM and StabMap, alongside SpatialGlue. Visually, SpatialGlue was able to clearly recover all four spatial factors to closely match the ground truth (Fig. 2a). Seurat and MEFISTO were able to clearly recover two factors (factors 2 and 4 for Seurat, 3 and 4 for MEFISTO). Other methods were able to recover some of the factors but with much higher levels of noise (factor 2 for totalVI, 1 and 2 for MOFA+, MultiVI and scMM, 2 and 3 for StabMap). The metrics confirmed the visuals with SpatialGlue scoring top in all metrics, followed by Seurat and MEFISTO (Fig. 2c). We further tested all methods with four more datasets generated with modified distribution parameters (Supplementary Figs. 2–4) and measured their performance with the same metrics, summarizing the results with box plots (Fig. 2d). Again, SpatialGlue performed the best with little variance between different datasets. Lastly, we have also demonstrated using simulated data that the SpatialGlue framework is extensible to three or more modalities (Extended Data Fig. 2).

For the second example, we benchmarked SpatialGlue and the same competing methods with an in-house human lymph node dataset generated using 10x Genomics Visium RNA and protein co-profiling technology (section A1). Here we used the hematoxylin and eosin (H&E)-based annotation as the ground truth (Fig. 2e). In the annotation, the major structures include the pericapsular adipose tissue and capsule that form the outer layers of the bulb, while the cortex and medulla (sinuses, cords and vessels) form the core internal structures.

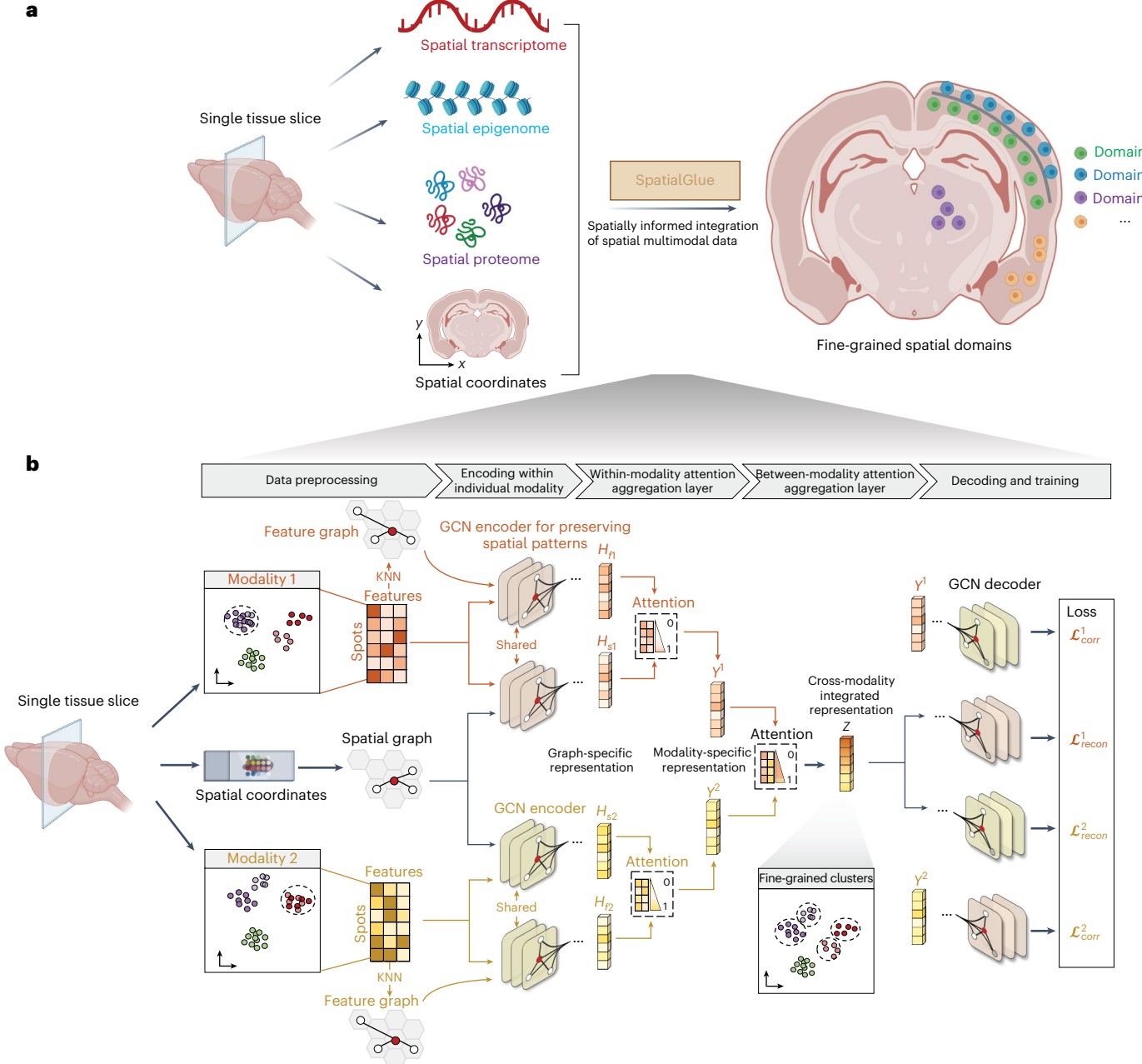

**Fig. 1 | Interpretable deep dual-attention model for spatial multi-omics data analysis. a**, SpatialGlue overview. SpatialGlue is designed to integrate the different omics modalities with spatial information to obtain a comprehensive molecular view of the tissue under study. **b**, SpatialGlue model structure. SpatialGlue first uses the *k*-nearest-neighbor (KNN) algorithm to construct a spatial neighbor graph using the spatial coordinates and a feature neighbor graph with the normalized expression data for each omics modality. Then for each modality, a GNN encoder takes in the normalized expressions and

the neighbor graph to learn two graph-specific representations by iteratively aggregating representations of neighbors. To capture the importance of different graphs, we designed a within-modality attention aggregation layer to adaptively integrate graph-specific representations and obtain a modality-specific representation. Finally, to preserve the importance of different modalities, SpatialGlue uses a between-modality attention aggregation layer to adaptively integrate modality-specific representations and output the final integrated representation of spots.

For comparison, we also plotted the single modality PCA-based clustering of RNA and protein (Fig. 2f). All the methods were able to isolate the paracortex (T cell zone, SpatialGlue cluster 1) that more resembled the RNA-specific and protein-specific modalities than the H&E annotation, which is unsurprising because T cells can be better identified by protein and gene markers such CD8A, CD3E and CCR7 (Supplementary Fig. 5). The methods were also unable to differentiate capsule layers from the pericapsular adipose tissue, which were not well captured in the RNA and protein modalities either. Among the tested methods, SpatialGlue, Seurat, totalVI and MOFA+ were able to identify the follicle regions,

while MultiVI, scMM, MEFISTO and StabMap could not. The hilum, which normally accumulates fat, is only visible in the RNA modality, and only MOFA+ and SpatialGlue could separate it from the pericapsular layer. To assess performance quantitatively, we used both unsupervised and supervised metrics. We first used the unsupervised Moran's *I* score and Jaccard similarity coefficient to assess spatial autocorrelation of clusters and preservation of distance in the joint latent space, respectively. The Moran's *I* score was computed for each cluster and visualized as a box plot for each method. SpatialGlue outperformed all other methods with a median score of 0.62 (Fig. 2g). We computed

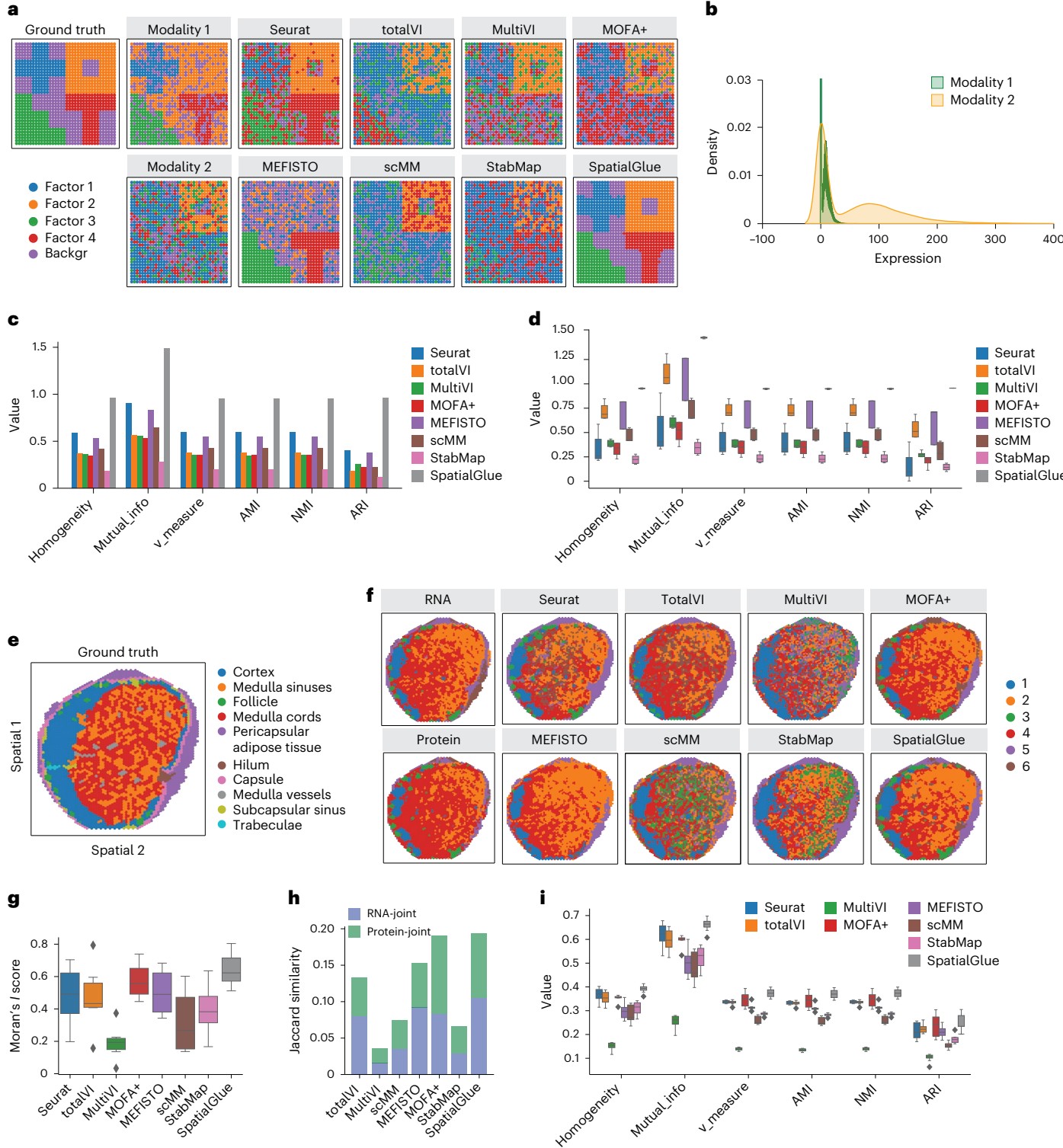

**Fig. 2 | SpatialGlue accurately identified spatial domains in simulated and real data. a**, Spatial plots of the simulated data, from left to right: ground truth, generated raw data of individual modalities, and clustering results by single-cell and spatial multi-omics integration methods—Seurat, totalVI, MultiVI, MOFA+, MEFISTO, scMM, StabMap and SpatialGlue. 'backgr' means background. **b**, Density distribution of the simulated data modalities. **c**, Quantitative evaluation of methods with six supervised metrics. **d**, Box plots of the six metrics with the scores from five simulated datasets. In the box plot, the center line denotes the median, box limits denote the upper and lower quartiles, and whiskers denote the 1.5 times the interquartile range. *n* = 5 simulated samples. **e**, Manual annotation of human lymph node sample A1. **f**, Spatial plots of lymph node sample A1, clustering of individual RNA and protein modalities

(left), clustering results (right) from single and spatial multi-omics integration methods—Seurat, totalVI, MultiVI, MOFA+, MEFISTO, scMM, StabMap and SpatialGlue. Note that the colors of clusters do not directly correspond to the same captured structures across different methods. **g**, Box plots of Moran's *I* score of the eight methods. **h**, Jaccard similarity scores of the eight methods. In the box plot, the center line denotes the median, box limits denote the upper and lower quartiles, and whiskers denote the 1.5 times the interquartile range. *n* = 6 clusters. **i**, Box plots of six supervised metrics with scores of clustering results with the number of clusters ranging from 4 to 11. In the box plot, the center line denotes the median, box limits denote the upper and lower quartiles, and whiskers denote 1.5 times the interquartile range. *n* = 12 clustering results with different resolutions for each method.

a Jaccard similarity score to quantify the overlap of neighbor sets between the joint space and each modality. Summed together, the total Jaccard similarity of SpatialGlue also outperformed the other methods with MOFA+ as a close second (Fig. 2h). For the supervised metrics computed with respect to the ground truth, SpatialGlue likewise outperformed all other methods with six clusters (Extended Data Fig. 3c). We further generated different numbers of clusters and the resulting box plots of supervised metrics showed results stability regardless of clustering resolution (Fig. 2i). To ensure that the results were not predicated on a specific tissue section, we applied the same methods to another human lymph node section (D1). With this data, SpatialGlue showed comparable scores with six clusters, but achieved more stable performance across different clustering resolutions than other methods (Extended Data Fig. 3d–i).

## Capturing mouse brain anatomy from spatial epigenome–transcriptome data at higher resolution than with individual modalities

Next, we applied SpatialGlue to mouse brain epigenome–transcriptome datasets to showcase its ability to resolve spatial domains at a higher resolution than methods used in the original study. We first tested SpatialGlue on a postnatal day (P)22 mouse brain coronal section dataset acquired using spatial ATAC–RNA-seq[3] to measure mRNA and open chromatin regions. We used the Allen brain atlas reference to annotate anatomical regions such as the cortex layers (ctx), genu of corpus callosum (ccg), lateral septal nucleus (ls) and nucleus accumbens (acb; Fig. 3a). For benchmarking, we tested SpatialGlue against Seurat, MultiVI, MOFA+, scMM and StabMap. We did not include MEFISTO and totalVI because we could not finish running MEFISTO within 12 h, and totalVI was designed only for CITE-seq. We first visualized the individual modalities (Fig. 3b), where we see that they captured various regions with differing accuracy. While both modalities captured the lateral ventricle (vl) and the lateral preoptic area (lpo), the RNA modality clearly captured the ccg but was unable to differentiate the ctx layers. Meanwhile, the ATAC modality was able to isolate the caudoputamen (cp) as well as some of the ctx layers. SpatialGlue captured all the aforementioned anatomical regions (2-acb, 4-cp/13-cp, 9-vl, 11-ccg/aco, 12-ls and 18-lpo) and produced better defined layers in the ctx and anterior cingulate area (aca) regions. Notably, SpatialGlue was able to differentiate more ctx layers than all other methods including the original analysis by Zhang et al.[3]. Seurat was able to capture the vl, acb, cp and ctx layers, making it the second-best method, while the other methods could only capture the ccg and a few of the other structures. In general, the outputs of competing methods presented more noise than SpatialGlue, which was quantitatively confirmed by the Moran's *I* score (Fig. 3c). For the Jaccard similarity metric, SpatialGlue again ranked top (Fig. 3d). We next examined the cross-modality and intra-modality weights learned by SpatialGlue in the aggregation layers. These weights denote the contribution of individual modality's features and spatial information toward the integrated

output (Fig. 3e and Extended Data Fig. 4c). For the cross-modality weights, the RNA modality better segregated the ccg region and thus was assigned a heavier weight. On the other hand, for the ctx and vl, the ATAC modality showed more contribution and thus a heavier weight was assigned. It should be noted that the cross-modality weights are calculated based on the latent embeddings of each modality, rather than raw feature matrices. Therefore, observing some degree of discrepancies between the cross-modality weights and unimodal clusters is expected.

We extended the analysis to another P22 mouse brain dataset of a highly similar coronal section but with RNA-seq and CUT&Tag (acetylated histone H3 Lys27 (H3K27ac) histone modification) modalities. This dataset also does not have an annotated ground truth; therefore, we used the Allen brain atlas reference again for annotation. In this dataset, SpatialGlue captured the major structures of the ctx layers (clusters 1, 2, 5, 6, 11 and 12), 8-aca, 10-ccg/aco, cp (7,14), vl (9,16), 3-ls and 4-acb (Fig. 3f). By contrast, all other methods were unable to clearly capture many structures such as the acb and ls. The output of SpatialGlue also had the least noise, which was reflected in Moran's *I* score (Fig. 3g). For the Jaccard similarity, SpatialGlue achieved the highest score, highlighting that SpatialGlue's integrated output was able to best preserve the between-spot distance from the original individual data modalities (Fig. 3h). We also examined the modality weights for the contribution of the different modalities toward each cluster (Fig. 3i). For most clusters, the histone modification modality made a similar or greater contribution. For example, clusters ctx-1, ctx-5, ctx-6 and ctx-12 had heavier weights assigned to the histone modification modality. To ensure that the results were not contingent on dataset selection, we again tested on two other P22 mouse brain datasets with RNA-seq and CUT&Tag (H3K4me3 and H3K27me3 histone modification) modalities. SpatialGlue was again the top method in both Moran's *I* score and Jaccard similarity for these datasets (Extended Data Figs. 5 and 6).

We further analyzed the differentially expressed genes (DEGs) of each cluster (Fig. 3j) and found known markers for the different brain regions such as myelin-related genes, *Tspan2*, *Cldn11* and *Ugt8a*, expressed in the postnatal developing corpus callosum (10-ccg/aco), and *Olfm1*, *Cux2* and *Rorb* in the cortex layers. We next examined the differentially expressed peaks in the H3K27ac histone modification modality (Fig. 3k), where we found strong peaks in the clusters 12-ctx, 10-ccg/aco, 4-acb and 7-cp. Finally, following the original study[3], we plotted the peak-to-gene links heat map (Fig. 3l). Here, there are two major groups appearing in both data modalities, the first primarily consisting of acb/cp structures (4-acb, 7-cp and 14-cp), and the second comprising ctx-related clusters (6-ctx, 11-ctx and 12-ctx). This illustrated SpatialGlue's success in combining information from both modalities into the latent space to enable biologically relevant clusters. We believe such information combination has also contributed to the detection of the four cortical layers (clusters 5, 12, 1 and 6). Within the cortex layers 5 and 6 (cluster 6), *Tle4*, *Fezf2*, *Foxp2* and *Ntsr1* have been reported in the literature as markers. However, we only found *Tle4* and *Fezf2* expression to spatially coincide with the cluster. Conversely,

**Fig. 3 | SpatialGlue dissects spatial epigenome–transcriptome mouse brain samples at higher resolution. a**, Annotated reference of the mouse brain coronal section from the Allen Mouse Brain Atlas. **b**, Spatial plots of the RNA-seq and ATAC-seq data with unimodal clustering (left) and clustering results (right) from single-cell and spatial multi-omics integration methods—Seurat, MultiVI, MOFA+, scMM, StabMap and SpatialGlue. The annotated labels correspond to SpatialGlue's results and the clustering colors do not necessarily match to the same structures for the other methods. ctx, cerebral cortex; cp, caudoputamen; vl, lateral ventricle; lpo, lateral preoptic area; aca, anterior cingulate area; ls, lateral septal nucleus; aco, anterior commissure (olfactory limb); acb, nucleus accumbens; cc, corpus callosum. **c**, Box plots of Moran's *I* score of the six methods. In the box plot, the center line denotes the median, box limits denote the upper and lower quartiles, and whiskers denote 1.5 times the interquartile range. *n* = 18 clusters. **d**, Comparison of Jaccard similarity scores of the six

methods. **e**, Modality weights of different modalities, denoting their importance to the integrated output of SpatialGlue. **f**, Spatial plots of the RNA-seq and CUT&Tag-seq (H3K27ac) data with unimodal clustering (left), and clustering results (right) from single-cell and spatial multi-omics integration methods—Seurat, MultiVI, MOFA+, scMM, StabMap and SpatialGlue. The annotated labels correspond to SpatialGlue's results and the clustering colors do not necessarily match to the same structures for the other methods. **g**, Box plots of Moran's *I* score of the six methods. In the box plot, the center line denotes the median, box limits denote the upper and lower quartiles, and whiskers denote 1.5 times the interquartile range. *n* = 16 clusters. **h**, Comparison of Jaccard similarity scores of the six methods. **i**, Modality weights of different modalities, denoting their importance to the integrated output of SpatialGlue. **j**, Heat map of DEGs for each cluster. **k**, Heat map of differentially expressed peaks for each cluster. **l**, Heat map of peak-to-gene links.

*Ntsr1*'s gene activity score inferred from the histone marks matched the cluster spatially (Extended Data Fig. 7). This illustrated the different information within each modality that SpatialGlue can leverage to better demarcate different spatial domains.

## Accurately resolving mouse thymus structures and spleen macrophage subsets with SpatialGlue

Lastly, we demonstrated that SpatialGlue is broadly applicable to a wide spectrum of technology platforms by further applying it to

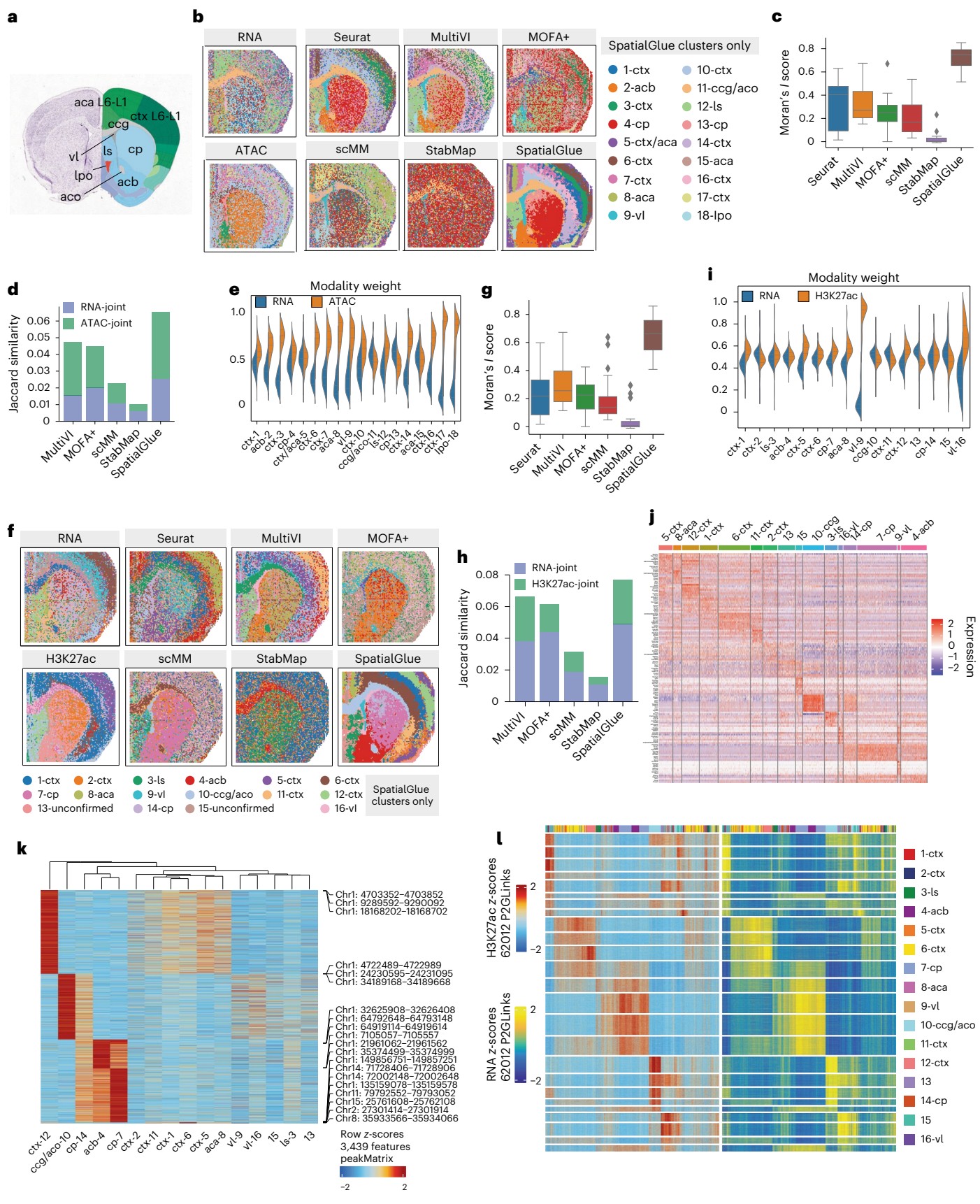

Stereo-CITE-seq and SPOTS-acquired data. The Stereo-CITE-seq[6] was used to analyze a mouse thymus section, capturing mRNA and protein at subcellular resolution (Fig. 4a). The thymus is a small gland surrounded by a capsule of fibers and collagen. It is divided into two lobes connected by a connective isthmus with each lobe being broadly divided into a central medulla surrounded by an outer cortex layer. In each data modality, broad outlines of the medulla regions and the surrounding cortex could be seen (Fig. 4a). We tested eight methods—Seurat, totalVI, MultiVI, MOFA+, MEFISTO, scMM, StabMap and SpatialGlue. MultiVI and StabMap were unable to find coherent clusters that resembled the medulla and cortex regions within the thymus. This was clearly reflected in the Moran's *I* score and Jaccard similarity with these two methods scoring the lowest (Fig. 4b,c). For MultiVI, its poor performance on protein + RNA data (Figs. 2 and 4) was probably due to it being optimized for RNA + ATAC data. Seurat, totalVI, scMM and SpatialGlue were more successful in capturing the internal structures by separating the medulla from the cortex, with SpatialGlue and scMM better demarcating the corticomedullary junction and the inner, middle and outer cortex (clusters 2–5; Fig. 4a and Extended Data Fig. 8e). Overall, SpatialGlue scored the highest in Jaccard similarity and second in Moran's *I* score. This superior performance was further replicated with three additional mouse thymus sections (Supplementary Figs. 6–8). For most clusters, the RNA modality made greater contributions than the protein (Fig. 4d), except for the middle cortex (cluster 4), where the protein modality contributed more.

In our final example, we benchmarked SpatialGlue's capabilities with mouse spleen spatial profiling data consisting of protein and transcript measurements[4]. The spleen is an important organ within the lymphatic system with functions including B cell maturation in germinal centers formed within B cell follicles (Fig. 4e). These are complex structures with an array of immune cells present. The data were generated with SPOTS, which uses the 10x Genomics Visium technology to capture whole transcriptomes and extracellular proteins via polyadenylated antibody-derived tag (ADT)-conjugated antibodies. The protein detection panel used for this experiment was designed to detect the surface markers of B cells, T cells and macrophages, which are well represented in the spleen. After preprocessing, we performed clustering of each data modality and plotted the clusters on the tissue slide to examine their correspondence between modalities (Fig. 4f). The clusters clearly did not align, indicating that each modality possessed different information content (Extended Data Fig. 9f,g). Using the protein markers and DEGs, clusters of spots enriched with B cells, T cells and macrophage subsets were annotated[22–24]. Specifically, we identified macrophage subsets (RpMΦ, MZMΦ, MMMΦ) that were not annotated in the original study. We then tested Seurat, totalVI, MultiVI, MOFA+, MEFISTO, scMM, StabMap and SpatialGlue (Fig. 4f). MultiVI and StabMap again did not capture coherent clusters. This was also reflected in their Moran's *I* scores and Jaccard similarity (Fig. 4g,h).

The remaining methods captured clusters with similar Moran's *I* scores but SpatialGlue scored the highest in terms of Jaccard similarity. We then examined SpatialGlue's learned modality weights (Fig. 4i). The protein modality made the bigger contribution to the MMMΦ cluster, which was mainly found in the protein modality plot. Conversely, SpatialGlue relied more on the RNA modality to capture the T cell cluster. To verify SpatialGlue's performance, we used another SPOTS-acquired dataset of a murine spleen section as a replicate. Here, SpatialGlue achieved comparable or greater Moran's *I* scores than baseline methods and scored the highest in terms of Jaccard similarity (Extended Data Fig. 10e,f).

To annotate the clusters found with SpatialGlue, we visualized the cell types' protein markers (Fig. 4j,k) and RNA expression of selected markers (Fig. 4l). Within the white pulp zone, the T cell spots were known to concentrate in small clusters known as T cell zones, while the B cell-enriched spots were mainly found in areas adjacent to the T cell clusters. The RpMΦ spots in the red pulp zone were easily identified by the strong expression of markers like F4_80 and CD163. To differentiate MZMΦ and MMMΦ, the RNA expression of *Cd209a* (MZMΦ) and *Siglec1* (MMMΦ) and protein expression of CD169 (MMMΦ) were used to guide the annotation.

From the cluster and marker visualization, we observed cell types that were spatially adjoining. Thus, we quantified the spatial relationships by computing neighborhood enrichment (Extended Data Fig. 9d) and co-occurrence probabilities with respect to the T cell and B cell clusters (Extended Data Fig. 9e). In general, we observed neighborhood enrichments that matched known biology such as the high correlation between the B cells and T cells, indicating that they were most likely to be found together at the closest distance. This was followed by MMMΦ, which surrounded T cell and B cell clusters in the white zone. These correlations reflected the layers of cell types that form the follicles and their surroundings. Between the macrophages, we observed a positive correlation between RpMΦ and MZMΦ, and MZMΦ and MMMΦ. This is a result of the red pulp (RpMΦ) forming the spleen's outer layer and the MZMΦ being positioned within the marginal zone surrounding white pulp, which in turn was enriched with MMMΦ.

## Discussion

SpatialGlue is a new deep learning model incorporating graph neural networks with a dual-attention mechanism that enables the integration of multi-omics data in a spatially aware manner. With the presented examples, we demonstrated SpatialGlue's ability to effectively integrate multiple data modalities with their respective spatial context to reveal histologically relevant structures of tissue samples. Furthermore, our quantitative benchmarking demonstrated that SpatialGlue exhibits superior performance to 10 state-of-the-art unimodal and non-spatial methods on 5 simulated datasets and 12 real datasets, highlighting the importance of spatial information and cross-omics

**Fig. 4 | SpatialGlue accurately integrates multimodal data from the mouse thymus (RNA and protein acquired with Stereo-CITE-seq) and mouse spleen (RNA and protein acquired using SPOTS). a**, Spatial plots of RNA and protein data (mouse thymus acquired with Stereo-CITE-seq) with unimodal clustering (left), and comparison of clustering results (right) from single-cell and spatial multi-omics integration methods—Seurat, totalVI, MultiVI, MOFA+, MEFISTO, scMM, StabMap and SpatialGlue. The annotated labels correspond to SpatialGlue's results and the clustering colors do not necessarily match to the same structures for the other methods. **b**, Box plots of Moran's *I* score of the eight methods. In the box plot, the center line denotes the median, box limits denote the upper and lower quartiles, and whiskers denote 1.5 times the interquartile range. *n* = 8 clusters. **c**, Comparison of Jaccard similarity scores of the eight methods. **d**, Modality weights of different modalities, denoting their importance to the integrated output of SpatialGlue. **e**, Histology image of the mouse spleen replicate 1 sample. **f**, Spatial plots RNA and protein data (mouse spleen acquired

using SPOTS) with unimodal clustering (left), and clustering results (right) from single-cell and spatial multi-omics integration methods—Seurat, totalVI, MultiVI, MOFA+, MEFISTO, scMM, StabMap and SpatialGlue. RpMΦ, MMMΦ and MZMΦ are red pulp macro, CD169 + MMM and CD209a+ MZM, respectively. The annotated labels correspond to SpatialGlue's results and the clustering colors do not necessarily match to the same structures for the other methods. **g**, Box plots of Moran's *I* score of the eight methods. In the box plot, the center line denotes the median, box limits denote the upper and lower quartiles, and whiskers denote 1.5 times the interquartile range. *n* = 5 clusters. **h**, Comparison of Jaccard similarity scores of the eight methods. **i**, Modality weights of different modalities, denoting their importance to the integrated output of SpatialGlue. **j**, Heat map of differentially expressed ADTs for each cluster. **k**, Normalized ADT levels of key surface markers for T cells (CD3, CD4, CD8), B cells (IgD, B220, CD19) and RpMΦ (F4_80, CD68, CD163). **l**, Violin plots of two marker genes in the MMMΦ, MZMΦ and RpMΦ clusters.

integration. We also demonstrated SpatialGlue's ability to resolve finer-grained tissue structures, which can facilitate new biological findings in future studies. For example, its application to a set of mouse

brain epigenome–transcriptome data revealed finer cortical layers compared to the original study, which can allow further investigation of gene regulation at a higher spatial resolution. Our examples also

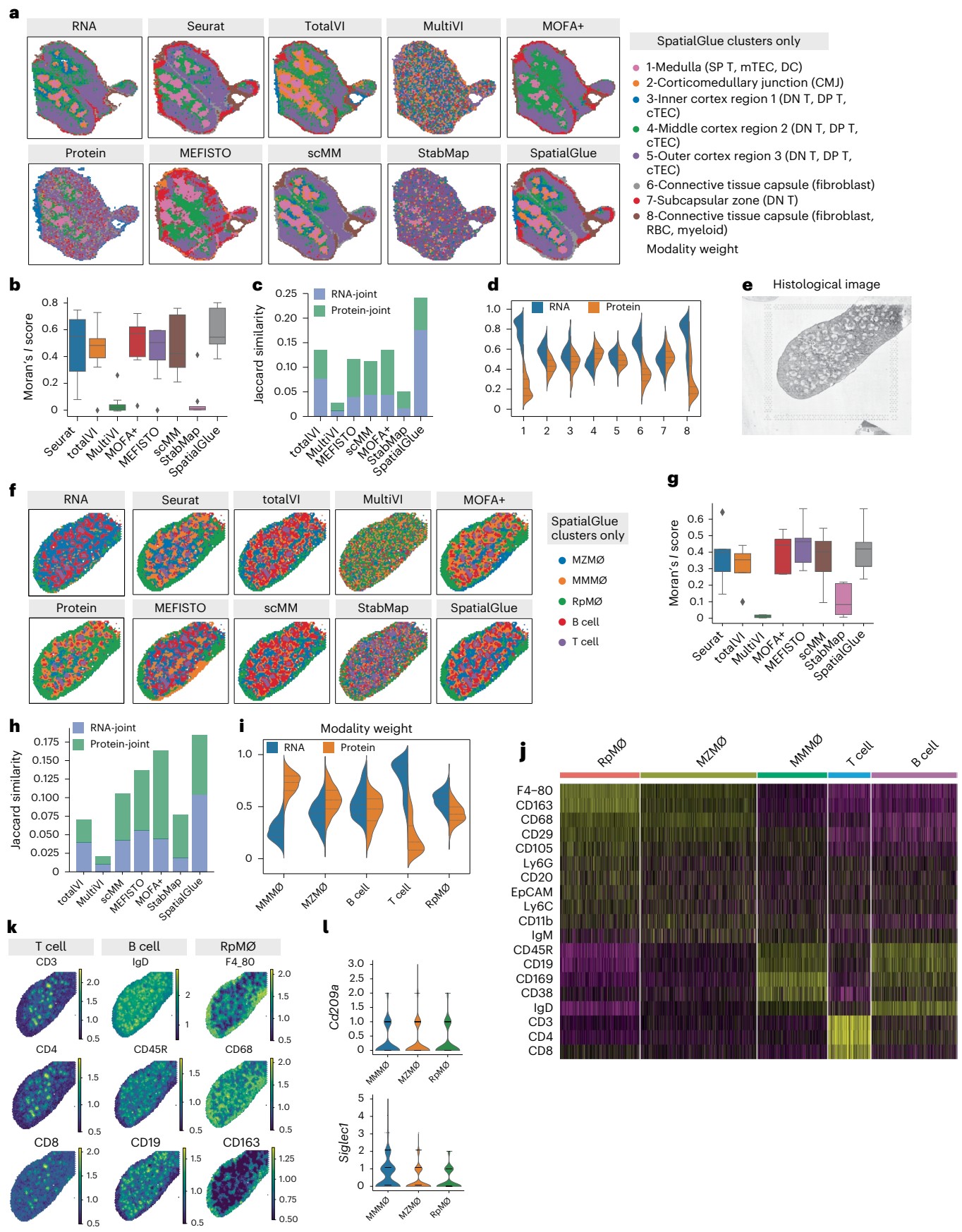

spanned four different tissue types and four technology platforms to show its broad applicability. Despite having demonstrated its application only on sequencing-based spatial omics data, its design allows seamless extension to image-based omics data from technology platforms like 10x Genomics Xenium and Nanostring CosMx, exhibiting a technology-agnostic nature.

As a GNN-based method, SpatialGlue bears such similarity to other GNN methods such as GraphST and SpaGCN that were designed for spatial unimodal omic. Naturally, as a method tailored for spatial multi-omics, SpatialGlue is different as it is explicitly designed to take in multiple data modalities as input and use attention to integrate data, as opposed to concatenation at the data preprocessing step. Unlike other existing multimodal methods such as Seurat, totalVI, MultiVI, MOFA+, scMM and StabMap, our model is spatially informed and it employs attention mechanisms to adaptively learn the relative importance between omics modalities, and between spatial location and omics features within each modality. Currently, all input modalities to SpatialGlue are preprocessed and dimensionally reduced to the same number of dimensions. When one data modality has substantially fewer dimensions than the rest, the resulting input dimension restriction on all modalities may result in some information loss. At present, the protein modality is most likely to be the most restrictive owing to acquisition technology limitations such as in surface protein detection. For example, SPOTS can simultaneously capture up to 32 surface proteins. While a larger number of proteins and consequently dimensions is preferred, 32 is not overly restrictive in our opinion. We believe that the number of proteins captured will increase with technological advances and consequently eliminate this restriction. Moreover, we plan to upgrade our model to work with full feature sets and a modality-specific number of latent dimensions to reflect the respective data modality complexity.

We designed SpatialGlue to be computation resource efficient and thus ensure its relevance as data sizes increase. The largest dataset tested contains 9,752 spots (spatial epigenome–transcriptome mouse brain), and it required about 5 min of wall-clock time on a server equipped with an Intel Core i7-8665U CPU and NVIDIA RTX A6000 GPU. Our testing showed that it scales well with the number of features and cells/spots (Extended Data Fig. 1g,h). Therefore, we believe SpatialGlue will be an invaluable analysis tool for present and future spatial multi-omics data. There are also multiple avenues of possible extensions for SpatialGlue. One is to include images as a modality. Most technologies can produce accompanying imaging data such as H&E, which contains essential information of the cell and tissue morphology. Thus, we plan to extend SpatialGlue to incorporate image data at either the intra-modality or inter-modality attention aggregation layer. We also aim to extend SpatialGlue's functionality with integration of multi-omics data acquired from serial tissue sections. Such multi-section integration can involve sections with the same data modalities or even mosaic integration of different modalities captured by different sections.

## Online content

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

[1]Institute of Molecular and Cell Biology (IMCB), Agency for Science, Technology and Research (A*STAR), Singapore, Singapore. [2]Binformatics Institute (BII), Agency for Science, Technology and Research (A*STAR), Singapore, Singapore. [3]BGI-Shenzhen, Shenzhen, China. [4]BGI Research-Southwest, BGI, Chongqing, China. [5]The Ken & Ruth Davee Department of Neurology, Northwestern University Feinberg School of Medicine, Chicago, IL, USA. [6]Department of Biomedical Engineering, Yale University, New Haven, CT, USA. [7]Singapore Immunology Network (SIgN), Agency for Science, Technology and Research (A*STAR), Singapore, Singapore. [8]Immunology Translational Research Programme, Yong Loo Lin School of Medicine, National University of Singapore, Singapore, Singapore. [9]Department of Microbiology and Immunology, Yong Loo Lin School of Medicine, National University of Singapore, Singapore, Singapore. [10]BGI Research Asia-Pacific, BGI, Singapore, Singapore. [11]Shanghai Immune Therapy Institute, Shanghai Jiao Tong University School of Medicine Affiliated Renji Hospital, Shanghai, China. [12]JFL-BGI STOmics Center, Jinfeng Laboratory, Chongqing, China. [13]Cancer Translational Research Programme, Yong Loo Lin School of Medicine, National University of Singapore, Singapore, Singapore. [14]Center for Computational Biology and Program in Cancer and Stem Cell Biology, Duke-NUS Medical School, Singapore, Singapore. ✉e-mail: chen_jinmiao@bii.a-star.edu.sg

## Methods

### Data

**Human lymph node dataset.** For spatial transcriptomics analysis of human tissues, two sequential sections of 5-μm thickness were utilized from formalin-fixed, paraffin-embedded (FFPE) lymph nodes. The sections underwent spatial transcriptomic library construction using CytAssist Visium platform (10x Genomics). Initially, sections were stained with H&E following the protocol outlined in the Visium CytAssist guide for FFPE samples, which includes steps for deparaffinization, staining, imaging and decrosslinking (CG000658, 10x Genomics). Imaging was performed with a ×20 objective on an EVOS M7000 microscope (Thermo).

Following imaging, spatial gene expression libraries were prepared utilizing probe-based methods, along with spatial protein expression libraries per the guidelines provided in the Visium CytAssist Reagent Kits manual (CG000494, 10x Genomics). We used the Visium Human Transcriptome Probe Set version 2.0 for RNA transcript detection, along with the Human FFPE Immune Profiling Panel, which includes a 35-plex CytAssist Panel of antibodies, both intracellular and extracellular, sourced from BioLegend and Abcam for protein detection. This panel also comprises four isotype controls. Antibody signals were normalized to isotype controls.

Libraries were sequenced on an Illumina NovaSeq S2 PE50 platform, allocating 2,000 million reads per lane at the NUSeq Core Facility, Northwestern University. The resultant FASTQ files were processed using Space Ranger (v2.1.0) software, referencing the GRCh38 human genome (GENCODE v32/Ensembl 98). For precise anatomical context, we conducted manual annotation of the lymph node structures, utilizing the high-resolution images captured by the EVOS M7000 microscope within the Loupe Browser software (10x Genomics).

**Spatial epigenome–transcriptome mouse brain dataset.** Brain tissue sections from a juvenile (P22) mouse were analyzed for the epigenome and transcriptome using spatial ATAC–RNA-seq and CUT&Tag-RNA-seq by Zhang et al.[3]. Microfluidic barcoding was used to capture spatial location and combined with in situ Tn5 transposition chemistry to capture chromatin accessibility. We used four datasets, one spatial ATAC–RNA-seq dataset and three spatial CUT&Tag-RNA-seq datasets. The number of pixels ranged from 9,215 to 9,752, the number of genes ranged from 22,731 to 25,881 and the number of peaks ranged from 35,270 to 121,068.

To preprocess the transcriptomic data, pixels expressing fewer than 200 genes and genes expressing fewer than 200 pixels were filtered out. Next, the gene expression counts were log-transformed and normalized by library size via the SCANPY package[25]. The top 3,000 highly variable genes (HVGs) were selected and used as input to PCA for dimensionality reduction. For consistency with the chromatin peak data, the first 50 principal components were retained and used as input to the encoder. For the chromatin peak data, we used latent semantic indexing to reduce the raw chromatin peak counts data to 50 dimensions.

**Stereo-CITE-seq mouse thymus dataset.** Murine thymus tissue samples were investigated with Stereo-CITE-seq for spatial multi-omics by Liao et al.[6]. For our study, we used data from four sections. The number of bins ranged from 4,228 to 4,697, the number of genes ranged from 23,221 to 23,960 and the sample included 19 or 51 proteins. For the transcriptomic data, we first filtered out genes expressed in fewer than 10 bins and bins with fewer than 80 genes expressed. The filtered gene expression counts were next log-transformed and normalized by library size via the SCANPY package[25]. Finally, to reduce the dimensionality of the data, the top 3,000 HVGs were selected and used as input for PCA. To ensure a consistent input dimension with the ADT data, the first 22 principal components were retained and used as the input of the encoder. For the ADT data, we first filtered out proteins expressed in fewer than 50 bins, resulting in 22 proteins being retained. The protein expression counts were then normalized using centered log ratio across each bin. PCA was then performed on the normalized data, and all 22 principal components were used as the input of the encoder.

**SPOTS mouse spleen dataset.** Ben-Chetrit et al.[4] processed fresh frozen mouse spleen tissue samples and analyzed them using the 10x Genomics Visium system supplemented with DNA-barcoded antibody staining. The antibodies (ADTs) enabled protein measurement alongside the transcriptome profiling using 10x Genomics Visium. The panel of 21 ADTs was designed to capture the markers of immune cells found in the spleen, including B cells, T cells and macrophages. We used two datasets (replicates 1 and 2) from the original study. The data contained 2,568 and 2,768 spots for replicates 1 and 2, respectively, with 32,285 genes captured per spot. For data preprocessing, we first filtered out genes expressed in fewer than 10 spots. The filtered gene expression counts were then log-transformed and normalized by library size using the SCANPY package[25]. Finally, the top 3,000 HVGs were selected and used as input for PCA. We used the first 21 principal components as the input of the encoder to ensure a consistent input dimension with the ADT data. For the ADT data, we applied centered log ratio normalization to the raw protein expression counts. PCA was then performed on the normalized data and the top 21 principal components were used as input to the encoder.

### The SpatialGlue framework

SpatialGlue is a new graph-based model with a dual-attention mechanism that aims to learn a unified representation by fully exploiting the spatial location information and expression data from different omics modalities. Within each modality, SpatialGlue first learns a modality-specific representation using both spatial and omics data. Subsequently, it synthesizes an integrated cross-modality representation by aggregating these modality-specific representations. Compared to cross-omics integration first followed by spatial integration, our approach allows us to capture modality-specific spatial correlations between spots and integrate the spatial information in a modality-specific manner.

We first consider a spatial multi-omics dataset with two different omics modalities, each with a distinct feature set $X_1 \in R^{N \times d_1}$ and $X_2 \in R^{N \times d_2}$. $N$ denotes the number of spots in the tissue. $d_1$ and $d_2$ are the numbers of features for two omics modalities, respectively. For example, in spatial epigenome–transcriptome, $X_1$ and $X_2$ refer to the sets of genes and chromatin regions respectively, while in Stereo-CITE-seq, $X_1$ and $X_2$ refer to the sets of genes and proteins, respectively. The primary objective of spatial multi-omics data integration is to learn a mapping function that can project the original individual modality data into a uniform latent space and then integrate the resulting representations. As shown in Fig. 1a, the SpatialGlue framework consists of four major modules: (1) modality-specific graph convolution network (GCN) encoder, (2) within-modality attention aggregation layer, (3) between-modality attention aggregation layer, and (4) modality-specific GCN decoder. The details of each module are described next. Notably, here we demonstrate the SpatialGlue framework with two modalities. Benefiting from the modular design, SpatialGlue readily extends to spatial multi-omics data with more than two modalities.

### Construction of neighbor graph

Assuming spots that are spatially adjacent in a tissue usually have similar cell types or cell states, we convert the spatial information to an undirected neighbor graph $G_s = (V, E)$ with $V$ denoting the set of $N$ spots and $E$ denoting the set of connected edges between spots. Let $A_s \in R^{N \times N}$ be the adjacent matrix of graph $G_s$, where $A_s(i,j) = 1$ if and only if the Euclidean distance between spots $i$ and $j$ is less than specific neighbor number $r$, otherwise 0. In our examples, we selected the top $r = 3$ nearest spots as neighbors of a given spot for all datasets according to experimental results.

In a complex tissue sample, it is possible for spots with the same cell types/states to be spatially nonadjacent to each other, or even far away. To capture the proximity of such spots in a latent space, we explicitly model the relationship between them using a feature graph. Specifically, we apply the KNN algorithm on the PCA embeddings and construct the feature graph $G_f^m = (V^m, E^m)$, where $V^m$ and $E^m$ denote the sets of $N$ spots and connected edges between spots in the $m \in \{1, 2\}$-th modality, respectively. For a given spot, we choose the top $k$-nearest spots as its neighbors. By default, we set $k$ to 20 for all datasets. We use $A_f^m \in R^{N \times N}$ to denote the adjacency matrix of the feature graph $G_f^m$. If spot $j \in V^m$ is the neighbor of spot $i \in V^m$, then $A_f^m(i,j) = 1$, otherwise 0.

## Graph convolutional encoder for individual modality

Each modality (for example, mRNA or protein) contains a unique feature distribution. To encode each modality in a low-dimensional embedding space, we use the GCN[26], an unsupervised deep graph network, as the encoder of our framework. The main advantage of the GCN is that it can capture the cell expression patterns and neighborhood microenvironment while preserving the high-level global patterns. For each modality, using the preprocessed features as inputs, we separately implement a GCN encoder on the spatial adjacency graph $G_s$ and the feature graph $G_f$ to learn graph-specific representations $H$. These two neighbor graphs reflect distinct topological semantic relationships between spots. The semantic information in the spatial graph denotes the physical proximity between spots, while that in the feature graph denotes the phenotypic proximity of spots that have the same cell types/states but are spatially nonadjacent to each other. This enables the encoder to capture different local patterns and dependencies of each spot by iteratively aggregating the representations from its neighbors. Specifically, the $l$-th ($l \in \{1, 2, \ldots, L-1, L\}$) layer representation in the encoder are formulated according to equations (1–4):

$$H_{s1}^l = \sigma\left(\widetilde{A}_s H_{s1}^{l-1} W_{e1}^{l-1} + b_{e1}^{l-1}\right) \tag{1}$$

$$H_{f1}^l = \sigma\left(\widetilde{A}_f^1 H_{f1}^{l-1} W_{e1}^{l-1} + b_{e1}^{l-1}\right) \tag{2}$$

$$H_{s2}^l = \sigma\left(\widetilde{A}_s H_{s2}^{l-1} W_{e2}^{l-1} + b_{e2}^{l-1}\right) \tag{3}$$

$$H_{f2}^l = \sigma\left(\widetilde{A}_f^2 H_{f2}^{l-1} W_{e2}^{l-1} + b_{e2}^{l-1}\right) \tag{4}$$

where $\widetilde{A} = D^{-\frac{1}{2}} A D^{-\frac{1}{2}}$ represents the normalized adjacency matrix of specific graph and $D$ is a diagonal matrix with diagonal elements being $D_{ii} = \sum_{j=1}^N A_{ij}$. In particular, $\widetilde{A}_s$, $\widetilde{A}_f^1$, and $\widetilde{A}_f^2$ are the corresponding normalized adjacency matrices of the spatial graph, the feature graphs of modalities 1 and 2, respectively. $W_{e\cdot}$, and $b_{e\cdot}$ denote a trainable weight matrix and a bias vector, respectively. $\sigma(\cdot)$ is a nonlinear activation function such as the rectified linear unit. $H_\cdot^l$ denotes the $l$-th layer output representation, and $H_{s1}^0 = H_f^0$ and $H_{s2}^0 = H_{f2}^0$ are set as the input PCA embeddings of the original features $X_1$ and $X_2$, respectively. We also specify $H_\cdot^L \in R^{d_3}$, the output at the $L$-th layer, as the final latent representation of the encoder with $d_3$ as the hidden dimension. $H_{sm}$ and $H_{fm}$ represent the latent representations derived from the spatial and feature graphs within modality $m$, respectively.

## Within-modality attention aggregation layer

For an individual modality, taking its preprocessed features and two graphs (that is, spatial and feature graphs) as input, we can derive two graph-specific spot representations via the graph convolutional encoder, such as $H_{s1}$ and $H_{f1}$. To integrate the graph-specific representations, we design a within-modality attention aggregation layer following the encoder such that its output representation preserves expression similarity and spatial proximity. Given that different neighbor graphs can provide unique semantic information for each spot (as

mentioned above), the aggregation layer is designed to integrate graph-specific representations in an adaptive manner by capturing the importance of each graph. As a result, we derive a modality-specific representation for each modality. Specifically, for a given spot $i$, we first subject its graph-specific representation $h_i^t$ to a linear transformation (that is, a fully connected neural network), and then evaluate the importance of each graph by the similarity of the transformed representation and a trainable weight vector $q$. Formally, the attention coefficient $e_i^t$, representing the importance of graph $t$ to the spot $i$, is calculated by equation (5):

$$e_i^t = q^T \cdot \tanh\left(W_i^{\text{intra}} h_i^t + b_i^{\text{intra}}\right) \tag{5}$$

where $W_i^{\text{intra}}$ and $b_i^{\text{intra}}$ are the trainable weight matrix and bias vector, respectively. To reduce the number of parameters in the model, all the trainable parameters are shared by the different graph-specific representations within each modality. To make the attention coefficient comparable across different graphs, a softmax function is applied to the attention coefficient to derive attention score $\alpha_i^t$, according to equation (6):

$$\alpha_i^t = \frac{\exp\left(e_i^t\right)}{\sum_{t=1}^T \exp\left(e_i^t\right)} \tag{6}$$

where $T$ denotes the number of neighbor graphs (set to 2 here). $\alpha_i^t$ represents the semantic contribution of the $t$-th neighbor graph to the representation of spot $i$. A higher value of $\alpha_i^t$ means greater contribution.

Subsequently, the final representation $Y^m$ in the $m$-th modality can be generated by aggregating graph-specific representations according to their attention scores according to equation (7):

$$y_i^m = \sum_{t=1}^T \alpha_i^t \cdot h_i^t \tag{7}$$

such that $y_i^m \in R^{d_3}$ preserves the raw spot expressions, spot expression similarity and spatial proximity within modality $m$.

## Between-modality attention aggregation layer

Each individual omics modality provides a partial view of a complex tissue sample, thus requiring an integrated analysis to obtain a comprehensive picture. These views can contain both complementary and contradictory elements, and thus different importance should be assigned to each modality to achieve coherent data integration. Here we use a between-modality attention aggregation layer to adaptively integrate the different data modalities. This attention aggregation layer will focus on the more important omics modality by assigning greater weight values to the corresponding representation. Like the within-modality layer, we first learn the importance of modality $m$ by calculating the following coefficient $g_i^m$ according to equation (8):

$$g_i^m = v^T \cdot \tanh\left(W_i^{\text{inter}} y_i^m + b_i^{\text{inter}}\right) \tag{8}$$

where $g_i^m$ is the attention coefficient that represents the importance of the modality $m$ to the representation of spot $i$. $W^{\text{inter}}$, $b^{\text{inter}}$, and $v$ are learnable weight and bias variables, respectively. Similarly, we further normalize the attention coefficients using the softmax function according to equation (9):

$$\beta_i^m = \frac{\exp\left(g_i^m\right)}{\sum_{m=1}^M \exp\left(g_i^m\right)} \tag{9}$$

where $\beta_i^m$ is the normalized attention score that represents the contribution of the modality $m$ to the representation of spot $i$. $M$ is the number of modalities.

Finally, we derive the final representation matrix $Z$ by aggregating each modality-specific representation according to attention score $\beta$ given by equation (10):

$$z_i = \sum_{m=1}^{M} \beta_i^m \cdot y_i^m \tag{10}$$

After model training, the latent representation $z_i \in R^{d_3}$ can be used in various downstream analyses, including clustering, visualization and DEG detection.

## Model training of SpatialGlue

The resulting model is trained jointly with two different loss functions, that is, reconstruction loss and correspondence loss. Each loss function is described as follows.

**Reconstruction loss.** To enforce the learned latent representation to preserve the expression profiles from different modalities, we design an individual decoder for each modality to reverse the integrated representation $Z$ back into the normalized expression space. Specifically, by taking output $Z$ from the between-modality attention aggregation layer as input, the reconstructed representations $\hat{H}_1^l$ and $\hat{H}_2^l$ from the decoder at the $l$-th ($l \in \{1, 2, \ldots, L-1, L\}$) layer are formulated as follows according to equations (11) and (12):

$$\hat{H}_1^l = \sigma\left(\tilde{A}_s Z_1^{l-1} W_{d1}^{l-1} + b_{d1}^{l-1}\right) \tag{11}$$

$$\hat{H}_2^l = \sigma\left(\tilde{A}_s Z_1^{l-1} W_{d2}^{l-1} + b_{d2}^{l-1}\right) \tag{12}$$

where $W_{d1}$, $W_{d2}$, $b_{d1}$ and $b_{d2}$ are trainable weight matrices and bias vectors, respectively. $\hat{H}_1^l$ and $\hat{H}_2^l$ represent the reconstructed expression matrices for the omics modalities 1 and 2, respectively.

SpatialGlue's objective function to minimize the expression reconstruction loss is given by equation (13):

$$\mathcal{L}_{recon} = \gamma_1 \sum_{i=1}^{N} \left\| x_i^1 - \hat{h}_i^1 \right\|_F^2 + \gamma_2 \sum_{i=1}^{N} \left\| x_i^2 - \hat{h}_i^2 \right\|_F^2 \tag{13}$$

where $x^1$ and $x^2$ represent the original features of the modalities 1 and 2, respectively. $\gamma_1$ and $\gamma_2$ are weight factors that are utilized to balance the contribution of different modalities. Owing to the differences of sequencing technologies and molecular types, the feature distributions of different omics assays can vary substantially. As such, the weight factors also vary between different spatial multi-omics technologies but are fixed for datasets obtained using the same omics technology.

**Correspondence loss.** While reconstruction loss can enforce the learned latent representation to simultaneously capture the expression information of different modality data, it does not guarantee that the representation manifolds are fully aligned across modalities. To deal with the issue, we add a correspondence loss function. Correspondence loss aims to force consistency between a modality-specific representation $Y$ and its corresponding representation $\hat{Y}$ obtained through the decoder–encoder of another modality. Mathematically, the correspondence loss is defined in equations (14–16):

$$\mathcal{L}_{corr} = \gamma_3 \sum_{i=1}^{N} \left\| y_i^1 - \hat{y}_i^1 \right\|_F^2 + \gamma_4 \sum_{i=1}^{N} \left\| y_i^2 - \hat{y}_i^2 \right\|_F^2 \tag{14}$$

$$\hat{Y}_1^l = \sigma\left(\tilde{A}_s \left(\sigma\left(\tilde{A}_s Y_1^{l-1} W_{d2}^{l-1} + b_{d2}^{l-1}\right)\right) W_{e2}^{l-1} + b_{e2}^{l-1}\right) \tag{15}$$

$$\hat{Y}_2^l = \sigma\left(\tilde{A}_s \left(\sigma\left(\tilde{A}_s Y_2^{l-1} W_{d1}^{l-1} + b_{d1}^{l-1}\right)\right) W_{e1}^{l-1} + b_{e1}^{l-1}\right) \tag{16}$$

where $\gamma_3$ and $\gamma_4$ are hyper-parameters, controlling the influences of different modality data. We set $\hat{Y}_1^0 = Y_1$ and $\hat{Y}_2^0 = Y_2$. $\sigma(\cdot)$ which is a nonlinear activation function, that is, rectified linear unit.

Therefore, the overall loss function used for model training is defined according to equation (17):

$$\mathcal{L}_{total} = \mathcal{L}_{recon} + \mathcal{L}_{corr} \tag{17}$$

## Implementation details of SpatialGlue

For all datasets, a learning rate of 0.0001 was used. To account for differences in feature distribution across the datasets, a tailored group of weight factors $[\gamma_1, \gamma_2, \gamma_3, \gamma_4]$ was empirically assigned to each one. The weight factors were [1, 5] for the SPOTS mouse spleen dataset, [1, 5, 10] for the 10x Genomics Visium human lymph node dataset, [1, 10] for the Stereo-CITE-seq mouse thymus dataset, [1, 5] for the spatial epigenome–transcriptome mouse brain dataset. We also provided a default parameter set that would work for most users on most data types. The training epochs used for the SPOTS mouse spleen, 10x Genomics Visium human lymph node, Stereo-CITE-seq mouse thymus and spatial epigenome–transcriptome mouse brain datasets were 600, 200, 1,500 and 1,600, respectively.

## Data and detailed methods

Details on the datasets, downstream analyses, competing methods and metrics used are available in the Supplementary Information.

## Reporting summary

Further information on research design is available in the Nature Portfolio Reporting Summary linked to this article.

## Data availability

The SPOTS mouse spleen data were obtained from the Gene Expression Omnibus (GEO) repository (accession no. GSE198353)[4], the Stereo-CITE-seq mouse thymus data from BGI and the spatial epigenome–transcriptome mouse brain data from AtlasXplore (https://web.atlasxomics.com/visualization/Fan/)[3]. GRCh38.p13 human genome data were obtained from the GENCODE repository (accession no. GENCODE v32/Ensembl 98, https://www.gencodegenes.org/human/release_32.html). The 10x Visium human lymph node data were obtained from the GEO (accession no. GSE263617). The details of all datasets used are available in the Methods. The data used as input to the methods tested in this study, inclusive of the Stereo-CITE-seq and the in-house human lymph node data, have been uploaded to Zenodo and are freely available at https://doi.org/10.5281/zenodo.7879713 (ref. 27).

## Code availability

An open-source Python implementation of the SpatialGlue toolkit is accessible at https://github.com/JinmiaoChenLab/SpatialGlue/. A separate Python implementation of the SpatialGlue_3M toolkit designed for triple omics integration is accessible at https://github.com/JinmiaoChenLab/SpatialGlue_3M/. The Jupyter notebooks for reproducing the results in this paper are available at https://github.com/JinmiaoChenLab/SpatialGlue_notebook/.

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

## Acknowledgements

We thank Y. Tan for assistance in interpreting mouse thymus data, M. Wu for comments on the model and T. Watson for assistance

in submitting in-house data to the Gene Expression Omnibus (GEO) database. The research was supported by: A*STAR under its BMRC Central Research Fund (CRF, UIBR) Award; AI, Analytics and Informatics (AI3) Horizontal Technology Programme Office (HTPO) seed grant (Spatial transcriptomics ST in conjunction with graph neural networks for cell–cell interaction C211118015) from A*STAR, Singapore; Open Fund Individual Research Grant (mapping hematopoietic lineages of healthy patients and high-risk patients with acute myeloid leukemia with FLT3-ITD mutations using single-cell omics no. OFIRG18nov-0103) from the Ministry of Health, Singapore; National Research Foundation (NRF), award no. NRF-CRP26-2021-0001; the National Research Foundation, Singapore, and Singapore Ministry of Health's National Medical Research Council under its Open Fund-Large Collaborative Grant ('OF-LCG') (MOH-OFLCG18May-0003); and Singapore National Medical Research Council (NMRC/OFLCG/003/2018). L.G.N. was supported by the National Natural Science Foundation of China (grant 32270956) and Shanghai Science and Technology Commission (grant 20JC1410100).

## Author contributions

J.C. conceptualized and supervised the project. Y.L. designed the model. Y.L. developed the SpatialGlue software. Y.L., K.S.A., and J.C. wrote the manuscript. Y.L., J.C., R.F., D.Z., R.S., S.C.Y., C.Z., H.X. and K.S.A. performed the data analysis. C.Z. and R.S. ran the Seurat WNN algorithm. Y.L. prepared the figures. J.C., N.R.J.G., L.G.N. and N.H. annotated and interpreted the mouse thymus dataset. L.V.O. and I.K. annotated the human lymph node datasets. D.G. and L.V.O. generated the human lymph node dataset. S.L., Y.H., M.J., A.C. and X.X. generated the mouse thymus dataset.

## Competing interests

The authors declare no competing interests.

## Additional information

**Extended data** is available for this paper at https://doi.org/10.1038/s41592-024-02316-4.

**Correspondence and requests for materials** should be addressed to Jinmiao Chen.

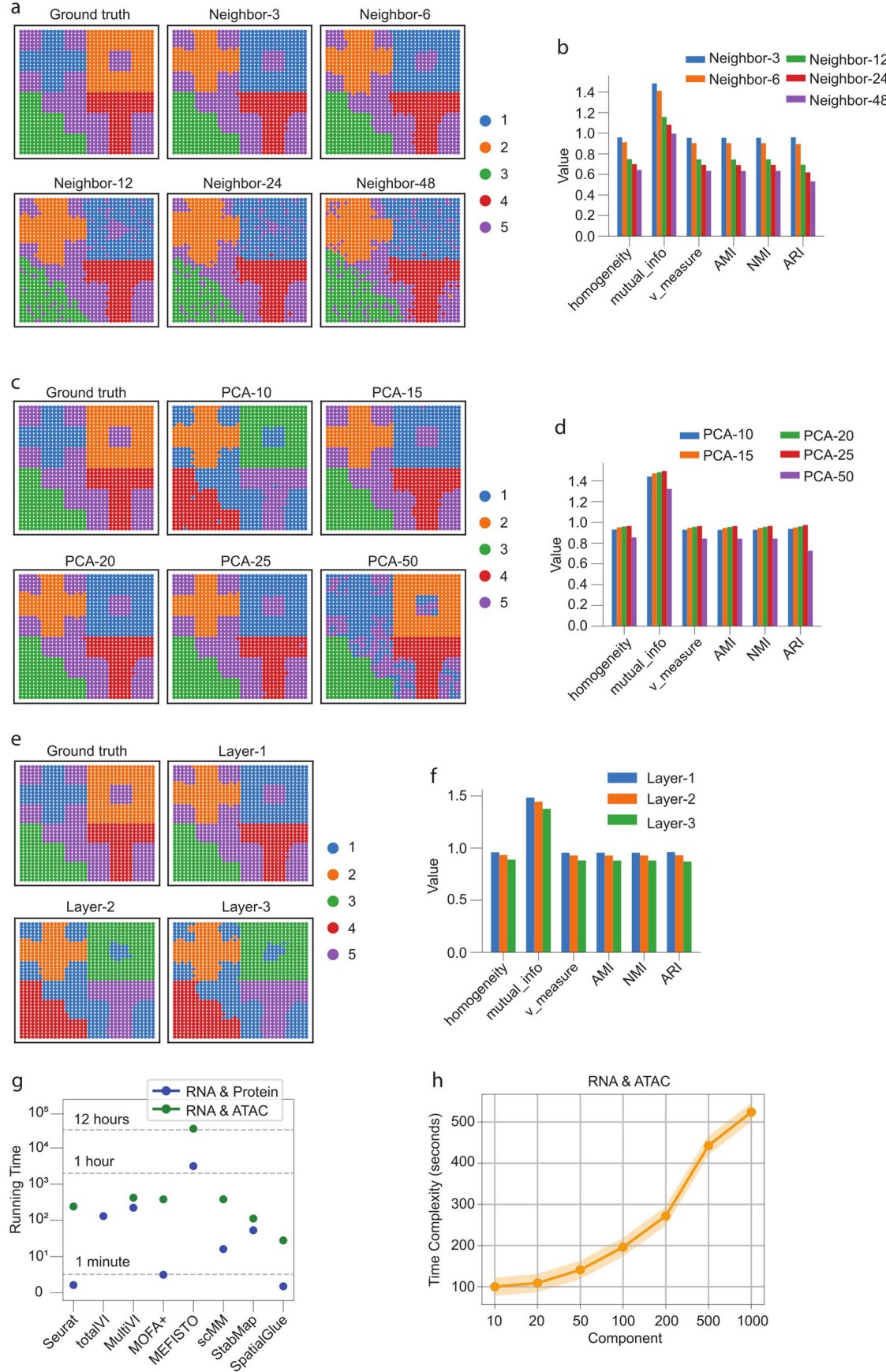

**Extended Data Fig. 1 | See next page for caption.**

**Extended Data Fig. 1 | Analysis of parameter sensitivity and time complexity with simulated and real data.** (**a**) Comparison of clustering results with different numbers of neighbors *k* to illustrate SpatialGlue's sensitivity to parameters. (**b**) Supervised metrics on the clustering results. (**c**) Comparison of clustering results with different number of PCs to illustrate SpatialGlue's sensitivity to input dimensionality. (**d**) Supervised metrics on the clustering results. (**e**) Comparison of clustering results with different numbers of GNN layers to illustrate SpatialGlue's sensitivity to input dimensionality. (**f**) Supervised metrics on the clustering results. (**g**) Time complexity of SpatialGlue and competing methods (Seurat, totalVI, MultiVI, MOFA+, MEFISTO, scMM, StabMap). In the experiment, we used murine spleen dataset with 4,697 spots for RNA & Protein data, and mouse brain RNA ATAC P22 with 9,215 cells for RNA & ATAC data. (**h**) Time complexity of SpatialGlue with various principal components as inputs on mouse brain RNA ATAC P22.

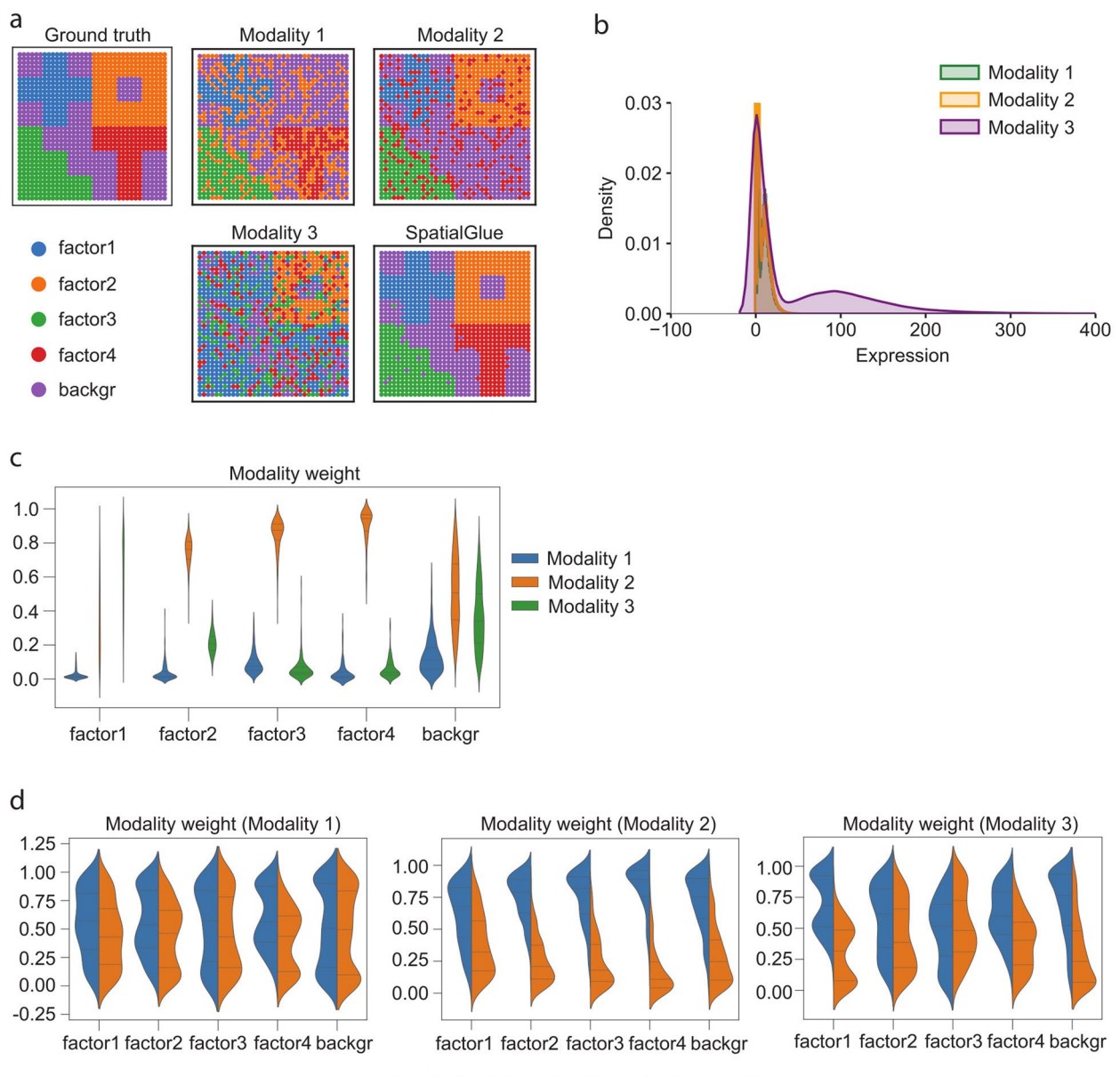

**Extended Data Fig. 2 | Evaluation of SpatialGlue on simulated triple omics data.** (**a**) ground truth and spatial plots of modalities 1, 2, 3, and SpatialGlue. (**b**) Density distribution of simulated data modalities. (**c**) SpatialGlue's between modality weights explaining the importance of each modality to each cluster. (**d**) Within-modality weights for the importance of spatial and feature graphs.

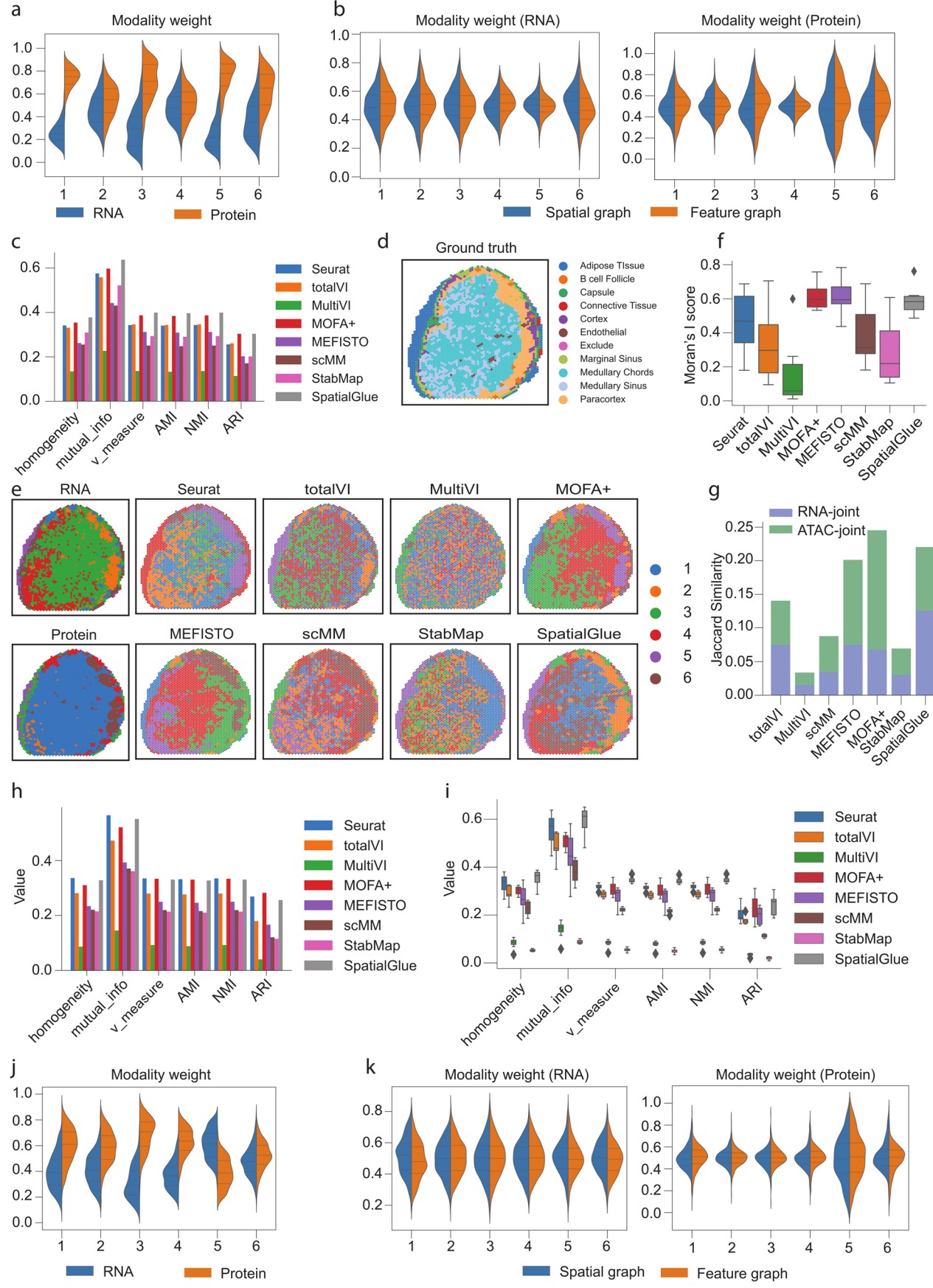

**Extended Data Fig. 3 | See next page for caption.**

**Extended Data Fig. 3 | Results for lymph node samples A1 (a-c) and D1(d-k).**
(**a**) SpatialGlue's between-modality weight explaining the importance of each
modality to each cluster for the lymph node A1 sample. (**b**) Within-modality
weights for the RNA and protein modalities explaining the contributions of
the spatial and feature graphs to each cluster. (**c**) Quantitative evaluation of
SpatialGlue and competing methods. (**d**) Ground truth for the lymph node D1
sample. (**e**) Spatial plots of RNA and protein data (left), and clustering results of
single-cell and spatial multi-omics methods, Seurat, totalVI, MultiVI, MOFA+,
MEFISTO, scMM, StabMap, and SpatialGlue. (**f**) Comparison of Moran's *I* score. In
the box plot, the center line denotes the median, box limits denote the upper and
lower quartiles, and whiskers denote the 1.5× interquartile range. n = 6 clusters.
(**g**) Comparison of Jaccard Similarity scores. (**h**) Quantitative evaluation with
six supervised metrics. (**i**) Box plots of six supervised metrics for clustering
results with number of clusters ranging from 4 to 11. In the box plot, the center
line denotes the median, box limits denote the upper and lower quartiles,
and whiskers denote the 1.5× interquartile range. n = 8 clustering results with
different resolutions for each method. (**j**) Between-modality weights explaining
the importance of each modality to each cluster. (**k**) Within-modality weights for
the RNA and protein modalities explaining the contributions of the spatial and
feature graphs to each cluster.

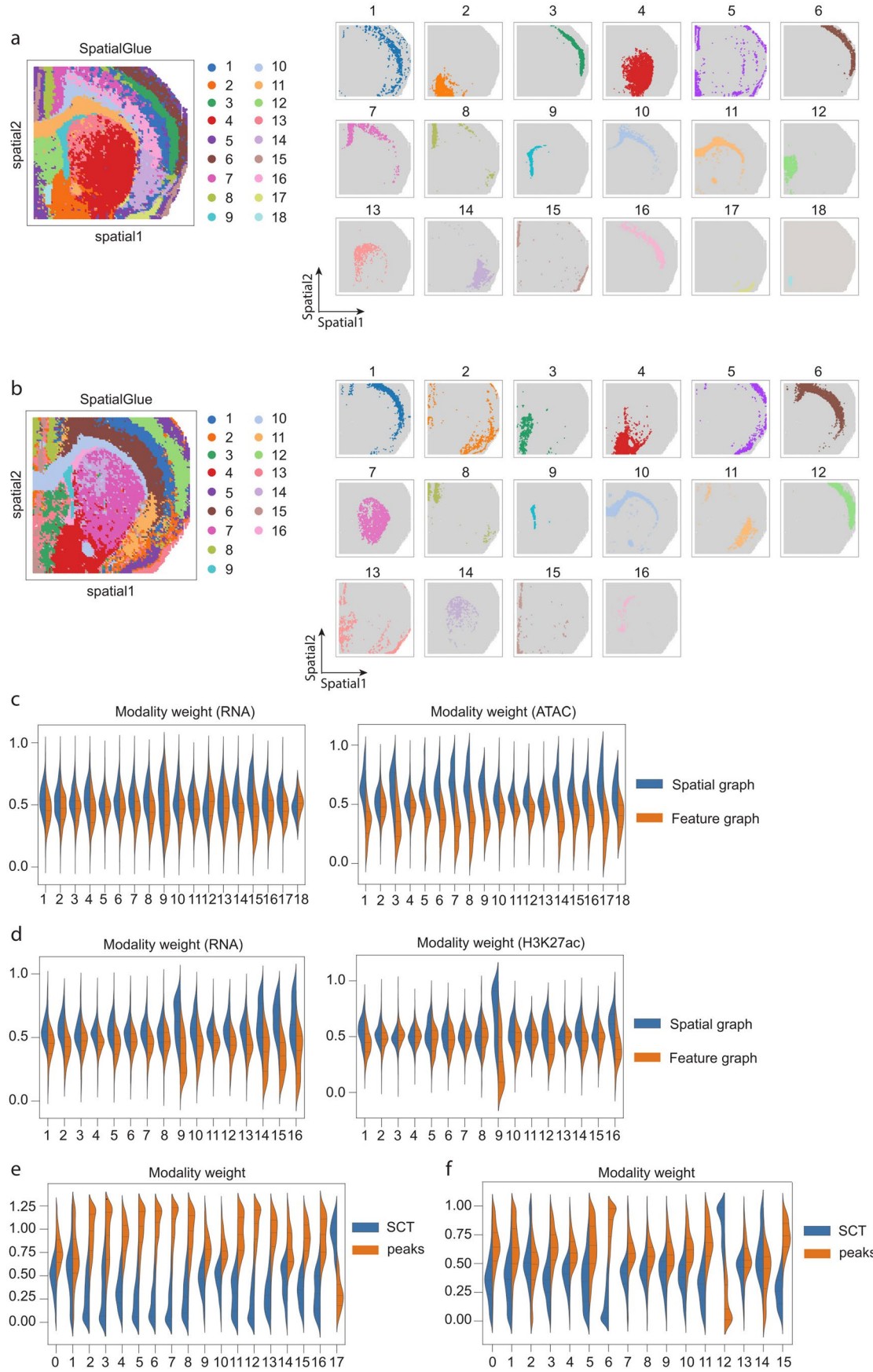

**Extended Data Fig. 4 | See next page for caption.**

**Extended Data Fig. 4 | Additional results of the mouse brain P22 samples (spatial-ATAC-RNA-seq and spatial-CUT&Tag-RNA-seq, H3K27ac). (a)** Separate spatial plots of all clusters identified by SpatialGlue in the mouse brain P22 sample (spatial-ATAC-RNA-seq). **(b)** Separate spatial plots of all clusters identified by SpatialGlue in the mouse brain P22 sample (spatial-CUT&Tag-RNA-seq, H3K27ac). **(c)** Within-modality weights for the RNA and ATAC modalities explaining the importance of the spatial and feature graphs to each cluster. **(d)** Within-modality weights for the RNA and CUT&Tag (H3K27ac) modalities explaining the importance of the spatial and feature graphs to each cluster. **(e)** Modality weights of Seurat when applied to the spatial-ATAC-RNA-seq P22 sample. **(f)** Modality weights of Seurat when applied to the spatial-CUT&Tag-RNA-seq (H3K27ac) sample.

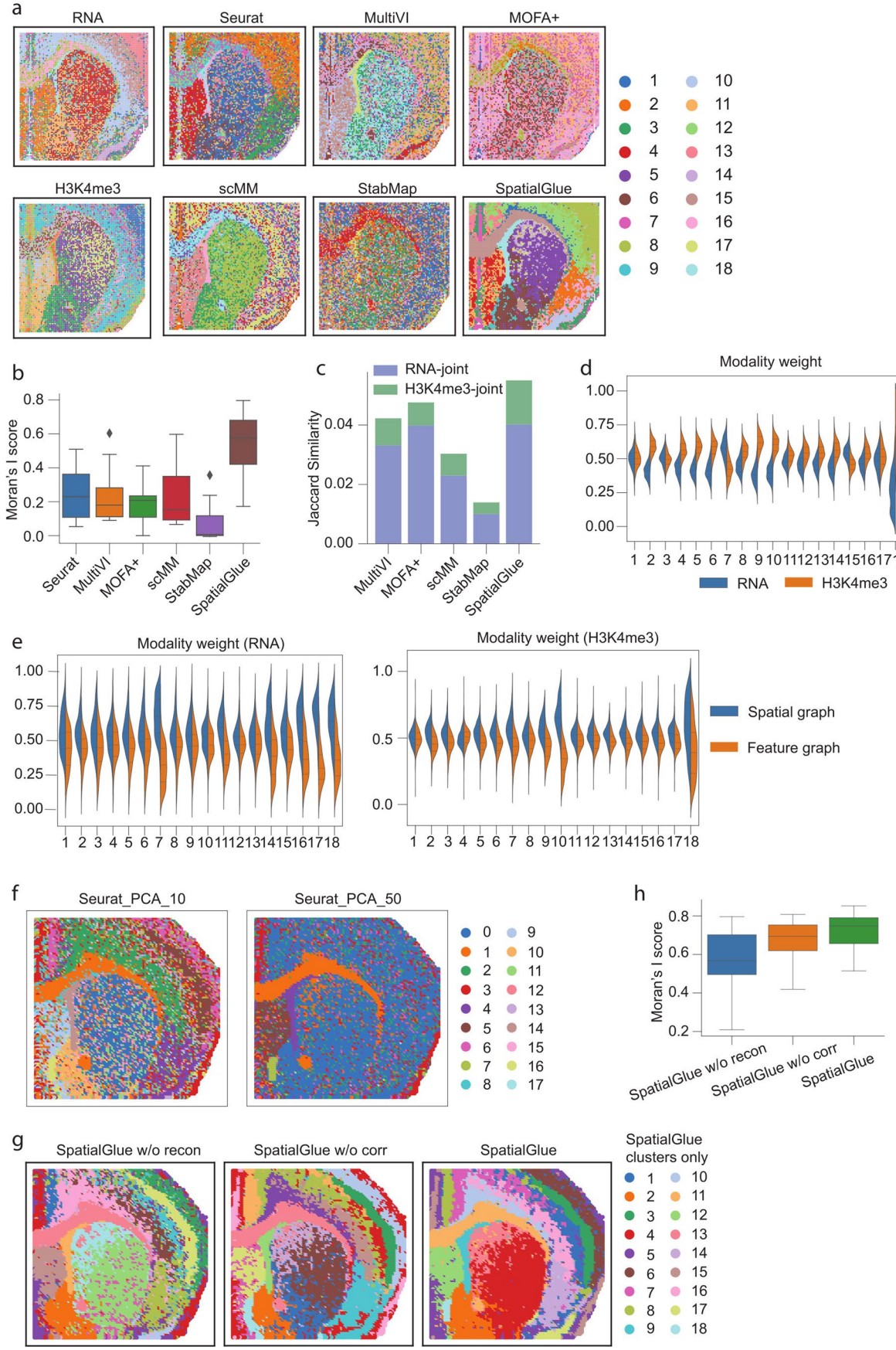

**Extended Data Fig. 5 | See next page for caption.**

**Extended Data Fig. 5 | Results of the mouse brain P22 sample acquired with RNA-seq and CUT&Tag-seq (H3K4me3).** (**a**) Spatial plots of data modalities with unimodal clustering (left), and clustering results (right) from single-cell and spatial multi-omics integration methods, Seurat, MultiVI, MOFA+, scMM, StabMap, and SpatialGlue. (**b**) Comparison of Moran's *I* score. In the box plot, the center line denotes the median, box limits denote the upper and lower quartiles, and whiskers denote the 1.5× interquartile range. n = 18 clusters. (**c**) Comparison of Jaccard Similarity scores. (**d**) Between-modality weights explaining the importance of each modality to each cluster. (**e**) Within-modality weights explaining the contributions of the spatial and feature graphs to each cluster for each modality. (**f**) Comparison of spatial clustering using Seurat with 10 and 50 PC dimensions in the mouse brain P22 spatial-ATAC-RNA-seq sample. (**g**) Comparison of SpatialGlue and its variants, that is, SpatialGlue without reconstruction loss ('SpatialGlue w/o recon') and SpatialGlue without correspondence loss ('SpatialGlue w/o corr'), in the mouse brain P22 spatial-ATAC-RNA-seq sample. (**h**) Comparison of Moran's I score of SpatialGlue and its two variants. In the box plot, the center line denotes the median, box limits denote the upper and lower quartiles, and whiskers denote the 1.5× interquartile range. n = 18 clusters.

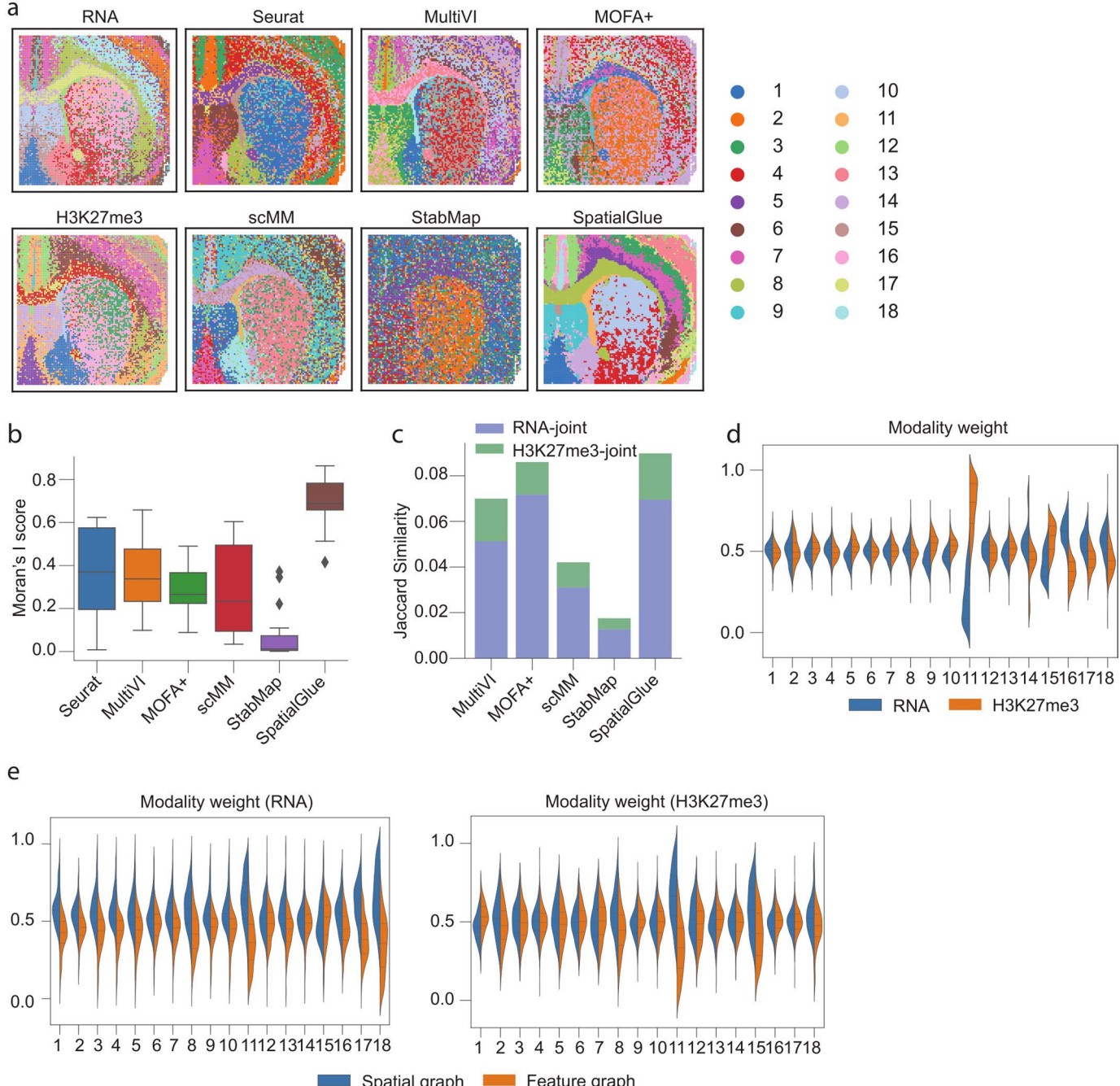

**Extended Data Fig. 6 | Results of the mouse brain P22 sample acquired with RNA-seq and CUT&Tag-seq (H3K27me3).** (**a**) Spatial plots of data modalities with unimodal clustering (left), and clustering results (right) from single-cell and spatial multi-omics integration methods, Seurat, MultiVI, MOFA+, scMM, StabMap, and SpatialGlue. (**b**) Comparison of Moran's *I* score. In the box plot, the center line denotes the median, box limits denote the upper and lower quartiles, and whiskers denote the 1.5× interquartile range. n = 18 clusters. (**c**) Comparison of Jaccard Similarity scores. (**d**) Between-modality weights explaining the importance of each modality to each cluster. (**e**) Within-modality weights explaining the contributions of the spatial and feature graphs to each cluster for each modality.

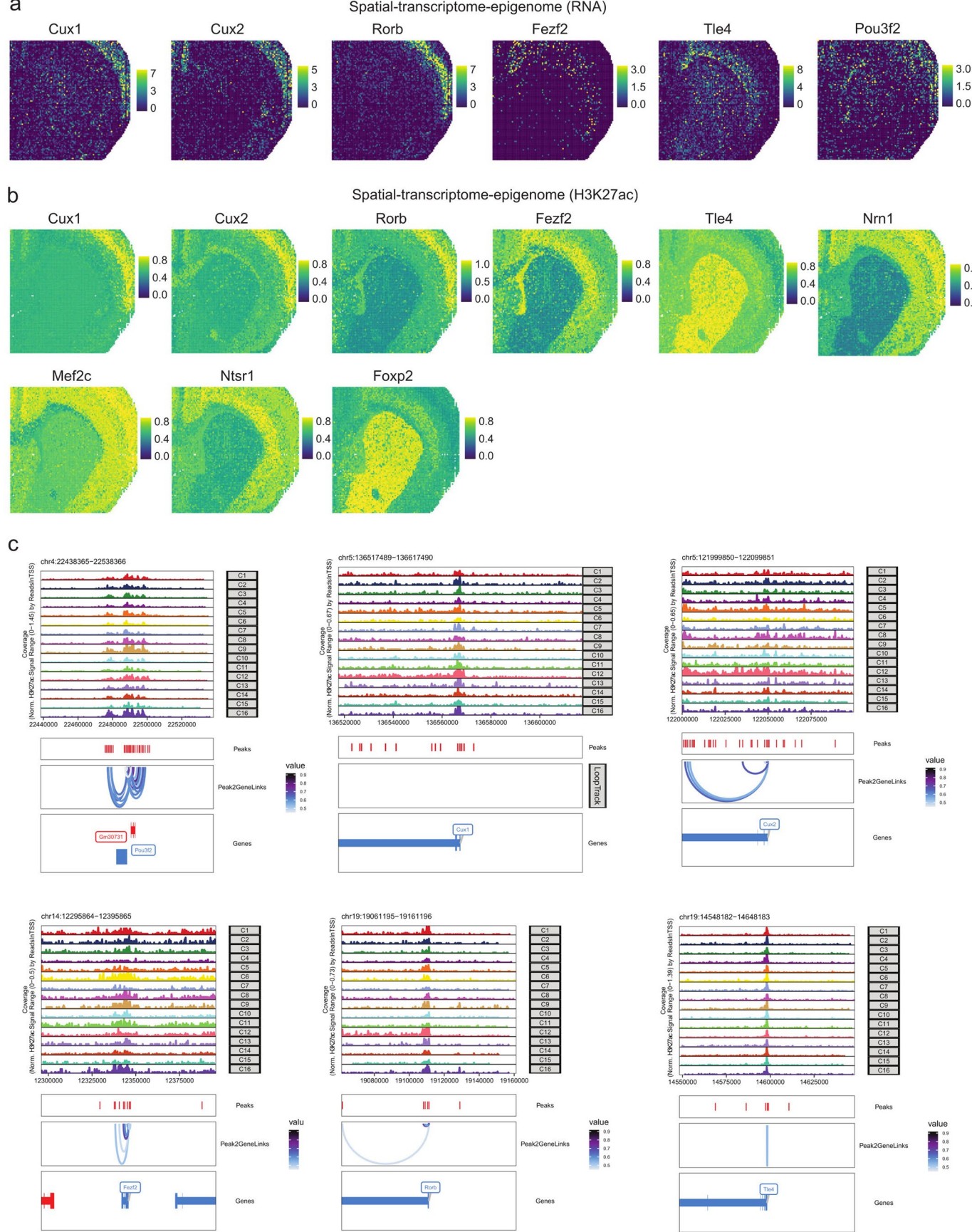

**Extended Data Fig. 7 | Additional results for the mouse brain P22 sample (spatial-CUT&Tag-RNA-seq, H3K27ac).** (**a**) Intensity plots of marker genes. (**b**) Normalized gene activity scores from Zhang et al. (**c**) Peak-to-gene links plots.

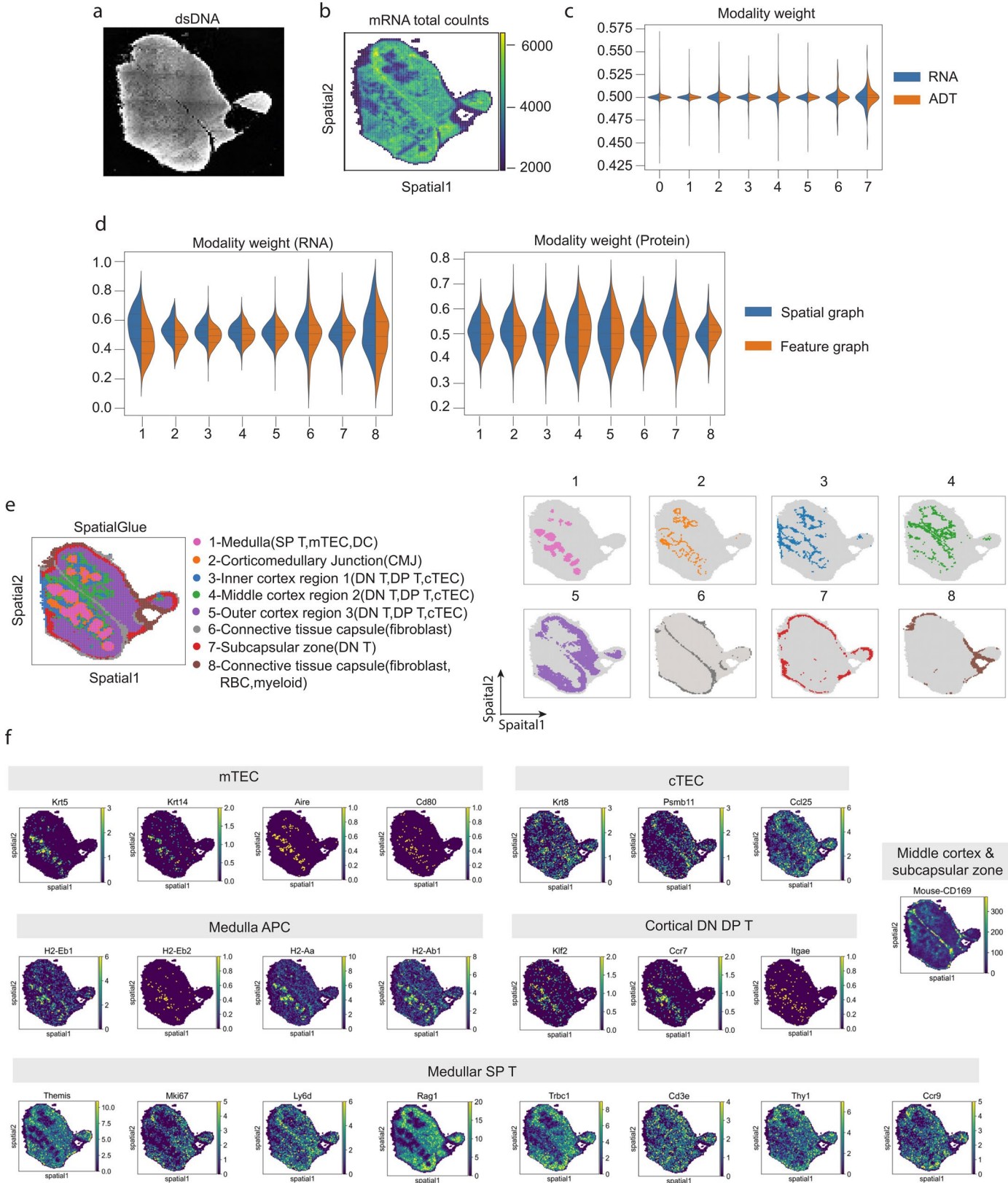

**Extended Data Fig. 8 | Additional results for the mouse thymus 1 sample.**
(**a**) dsDNA image. (**b**) Total mRNA counts. (**c**) Modality weight from Seurat when applied to the sample. (**d**) Within-modality weights of SpatialGlue explaining the contributions of the spatial and feature graphs to each cluster for each modality. (**e**) Separate spatial plots of all clusters identified by SpatialGlue. (**f**) Expression of marker genes and proteins for each cell type.

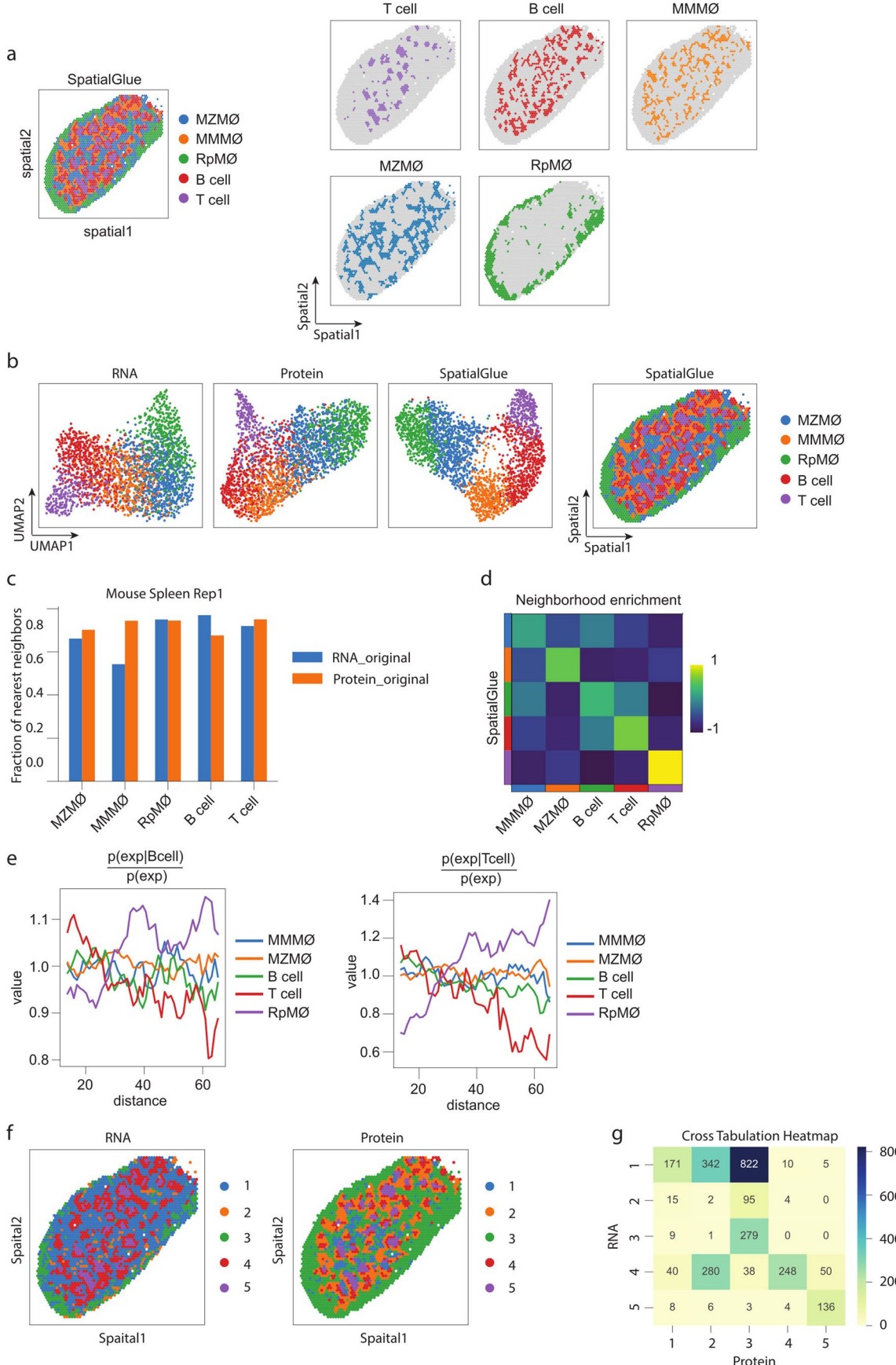

**Extended Data Fig. 9 | See next page for caption.**

**Extended Data Fig. 9 | Results for the mouse spleen replicate 1 sample.**
(**a**) Spatial plots of SpatialGlue's clusters together (left) and separate (right).
(**b**) UMAP plots of the RNA and protein data modalities (left), and spatial plot of
SpatialGlue's clusters (right). (**c**) Comparison of fraction of nearest neighbors
metric for each annotated cluster calculated by the different modalities (original
RNA and protein expression). (**d**) Neighborhood enrichment of cell type pairs.
(**e**) Cluster co-occurrence scores for each cluster at increasing distances.
(**f**) Spatial plots of the RNA and protein data modalities. (**g**) Cross tabulation
heatmap of the clustering labels between the RNA and protein data.

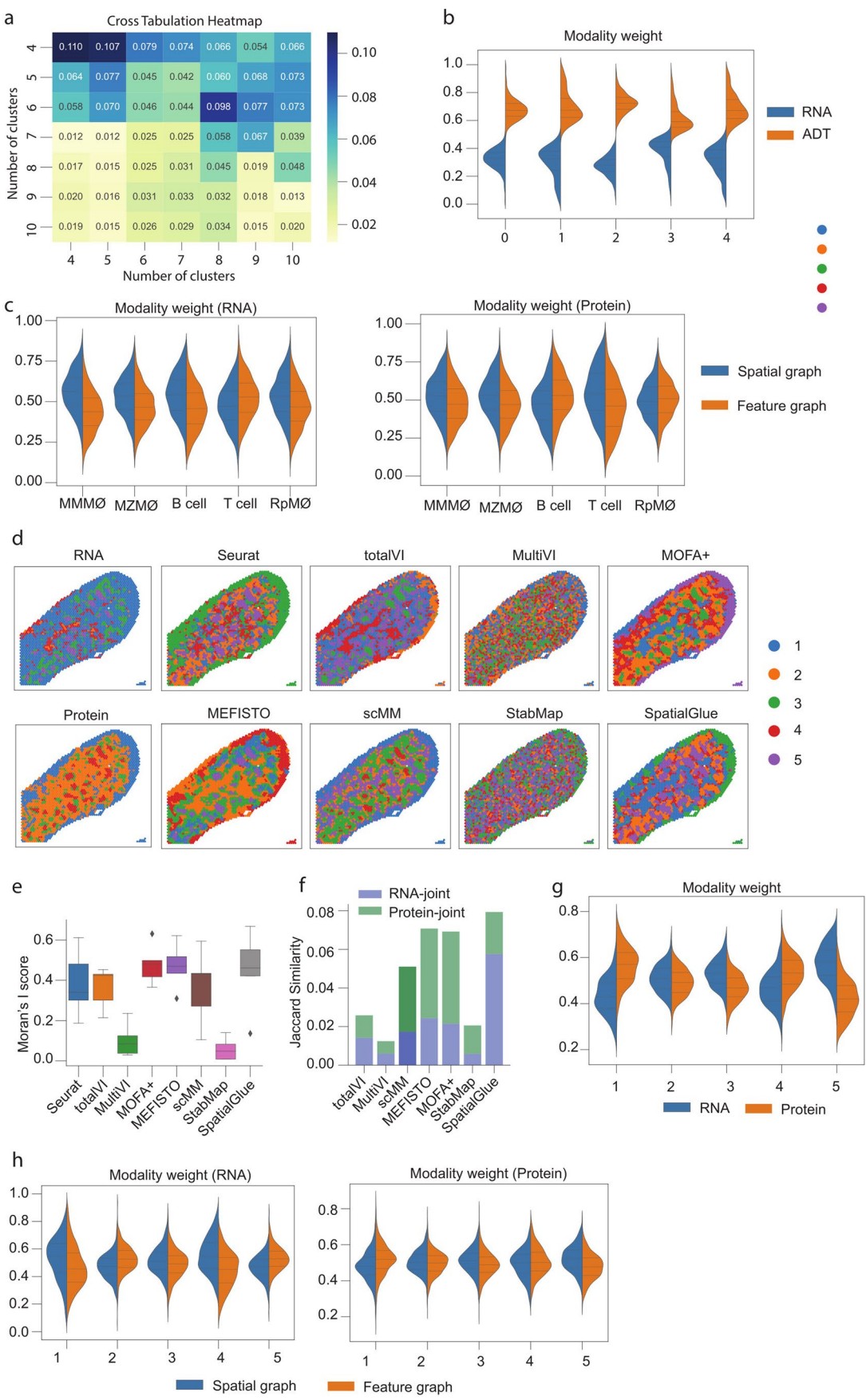

**Extended Data Fig. 10 | See next page for caption.**

**Extended Data Fig. 10 | Results for the mouse spleen replicate 1(a-c) and 2(d-h) samples.** (**a**) Cross tabulation heatmap for the number of clusters between the RNA and protein data. (**b**) Modality weights from Seurat. (**c**) Within-modality weights of SpatialGlue explaining the contributions of the spatial and feature graphs to each cluster for each modality. (**d**) Spatial plots of data modalities with unimodal clustering (left), and clustering results (right) from single-cell and spatial multi-omics integration methods, Seurat, totalVI, MultiVI, MOFA+, MEFISTO, scMM, StabMap, and SpatialGlue. (**e**) Comparison of Moran's *I* score. In the box plot, the center line denotes the median, box limits denote the upper and lower quartiles, and whiskers denote the 1.5× interquartile range. n = 5 clusters. (**f**) Comparison of Jaccard Similarity scores. (**g**) Between-modality weight explaining the importance of each modality to each cluster. (**h**) Within-modality weights explaining the contributions of the spatial and feature graphs to each cluster for each modality.

# Reporting Summary

## Statistics

For all statistical analyses, confirm that the following items are present in the figure legend, table legend, main text, or Methods section.

| n/a | Confirmed | |
|---|---|---|
| ☐ | ☒ | The exact sample size (*n*) for each experimental group/condition, given as a discrete number and unit of measurement |
| ☐ | ☒ | A statement on whether measurements were taken from distinct samples or whether the same sample was measured repeatedly |
| ☒ | ☐ | The statistical test(s) used AND whether they are one- or two-sided<br>*Only common tests should be described solely by name; describe more complex techniques in the Methods section.* |
| ☐ | ☒ | A description of all covariates tested |
| ☐ | ☒ | A description of any assumptions or corrections, such as tests of normality and adjustment for multiple comparisons |
| ☐ | ☒ | A full description of the statistical parameters including central tendency (e.g. means) or other basic estimates (e.g. regression coefficient) AND variation (e.g. standard deviation) or associated estimates of uncertainty (e.g. confidence intervals) |
| ☒ | ☐ | For null hypothesis testing, the test statistic (e.g. *F*, *t*, *r*) with confidence intervals, effect sizes, degrees of freedom and *P* value noted<br>*Give P values as exact values whenever suitable.* |
| ☒ | ☐ | For Bayesian analysis, information on the choice of priors and Markov chain Monte Carlo settings |
| ☐ | ☒ | For hierarchical and complex designs, identification of the appropriate level for tests and full reporting of outcomes |
| ☒ | ☐ | Estimates of effect sizes (e.g. Cohen's *d*, Pearson's *r*), indicating how they were calculated |

*Our web collection on statistics for biologists contains articles on many of the points above.*

## Software and code

Policy information about availability of computer code

| Data collection | No software was used for data collection. |
|---|---|
| Data analysis | Seurat v4.0.0 (https://github.com/satijalab/seurat), scvi-tools v1.0.2 (https://github.com/scverse/scvi-tools), MOFA+ v1.9.2 (https://github.com/bioFAM/MOFA2), MEFISTO v1.13.0 (https://github.com/bioFAM/MEFISTO_analyses), scMM (version unavailable) (https://github.com/kodaim1115/scMM), StabMap (version unavailable) (https://github.com/MarioniLab/StabMap), and SpatialGlue v1.1.5 (https://github.com/JinmiaoChenLab/SpatialGlue) were used for integrating spatial multi-omics data. Scanpy v1.9.1 was used for data pre-processing and result visualization. Signac v1.12.0 (https://github.com/stuart-lab/signac) and ArchR v1.0.2 (https://github.com/GreenleafLab/ArchR) were used for downstream analysis for spatial-epigenome-transcriptome data. Space Ranger v2.1.0 (https://www.10xgenomics.com/support/software/space-ranger/latest), MACS2 v2.2.6 (https://github.com/hbctraining/Intro-to-ChIPseq), and Loupe Brower v8.0 (https://www.10xgenomics.com/support/software/loupe-browser/latest) were used for generating in-house data. |

For manuscripts utilizing custom algorithms or software that are central to the research but not yet described in published literature, software must be made available to editors and reviewers. We strongly encourage code deposition in a community repository (e.g. GitHub). See the Nature Portfolio guidelines for submitting code & software for further information.

## Data

Policy information about availability of data

All manuscripts must include a data availability statement. This statement should provide the following information, where applicable:
- Accession codes, unique identifiers, or web links for publicly available datasets
- A description of any restrictions on data availability
- For clinical datasets or third party data, please ensure that the statement adheres to our policy

We analyzed 12 spatial multi-omics datasets across different data types and technology platforms, including 2 mouse spleen datasets acquired with SPOTS (Ben-Chetrit et al., 2023), 4 mouse thymus datasets from Stereo-CITE-seq (unpublished), and 4 mouse brain spatial-epigenome-transcriptome datasets (Zhang et al. 2023), and the 2 in-house human lymph node datasets acquired with 10x Visium CytAssist.

The SPOTS mouse spleen data was obtained from the GEO repository (accession no. GSE198353, https://www.ncbi.nlm.nih.gov/geo/query/acc.cgi?acc=GSE198353), the Stereo-CITE-seq mouse thymus data from BGI and  the spatial-epigenome-transcriptome mouse brain data from AtlasXplore (https://web.atlasxomics.com/visualization/Fan). GRCh38.p13 human genome was obtained from the GENCODE repository (accession no. GENCODE v32/Ensembl 98, https://www.gencodegenes.org/human/release_32.html). The details of all datasets used are available in the Methods section. The data used as input to the methods tested in this study, inclusive of the Stereo-CITE-seq and the in-house human lymph node data have been uploaded to Zenodo and is freely available at https://zenodo.org/record/7879713#.ZE3aOnZByUk.

We have added a 'data availability' section in the manuscript.

## Human research participants

Policy information about studies involving human research participants and Sex and Gender in Research.

| | |
|---|---|
| Reporting on sex and gender | N.A. |
| Population characteristics | N.A. |
| Recruitment | N.A. |
| Ethics oversight | N.A. |

Note that full information on the approval of the study protocol must also be provided in the manuscript.

# Field-specific reporting

Please select the one below that is the best fit for your research. If you are not sure, read the appropriate sections before making your selection.

☒ Life sciences        ☐ Behavioural & social sciences        ☐ Ecological, evolutionary & environmental sciences

For a reference copy of the document with all sections, see nature.com/documents/nr-reporting-summary-flat.pdf

# Life sciences study design

All studies must disclose on these points even when the disclosure is negative.

| | |
|---|---|
| Sample size | We used 10 publicly available data and 2 in-house data in the manuscript. For spatial transcriptomics analysis of human lymph node tissues, two sequential sections of 5 μm thickness were utilized from formalin-fixed, paraffin-embedded (FFPE) lymph node. |
| Data exclusions | We did not remove any spots from the spatial transcriptomic datasets. We also did not remove specific genes other than applying standard proccedures in data preprocessing. |
| Replication | Not applicable. Our experiments did not aim to uncover any mechanistic or intervention effect. Instead, we benchmarked our proposed methodology against competing methods with different datasets acquired using different technologies. |
| Randomization | Not applicable. Our experiments did not aim to uncover any mechanistic or intervention effect and hence did not require any controls. |
| Blinding | Not applicable. Our experiments did not involve human participants and their responses. |

# Reporting for specific materials, systems and methods

We require information from authors about some types of materials, experimental systems and methods used in many studies. Here, indicate whether each material, system or method listed is relevant to your study. If you are not sure if a list item applies to your research, read the appropriate section before selecting a response.

## Materials & experimental systems

| n/a | Involved in the study |
|-----|----------------------|
| ☒ ☐ | Antibodies |
| ☒ ☐ | Eukaryotic cell lines |
| ☒ ☐ | Palaeontology and archaeology |
| ☐ ☒ | Animals and other organisms |
| ☒ ☐ | Clinical data |
| ☒ ☐ | Dual use research of concern |

## Methods

| n/a | Involved in the study |
|-----|----------------------|
| ☒ ☐ | ChIP-seq |
| ☒ ☐ | Flow cytometry |
| ☒ ☐ | MRI-based neuroimaging |

# Animals and other research organisms

Policy information about [studies involving animals](); [ARRIVE guidelines]() recommended for reporting animal research, and [Sex and Gender in Research]()

| | |
|---|---|
| Laboratory animals | N.A. |
| Wild animals | N.A. |
| Reporting on sex | N.A |
| Field-collected samples | N.A |
| Ethics oversight | N.A |

Note that full information on the approval of the study protocol must also be provided in the manuscript.

