## [Peer Review File · Nature Methods]

Peer Review Information

Manuscript Title: Deciphering spatial domains from spatial multi-omics with SpatialGlue

Corresponding author name(s): Jin Miao Chen

Editorial Notes: None

Reviewer Comments & Decisions:

Decision Letter, initial version:

Dear Jinmiao,

Your Brief Communication, "Deciphering spatial domains from spatial multi-omics with SpatialGlue", has now been seen by three reviewers. As you will see from their comments below, although the reviewers find your work of considerable potential interest, they have raised a number of concerns. We are interested in the possibility of publishing your paper in Nature Methods, but would like to consider your response to these concerns before we reach a final decision on publication.

We therefore invite you to revise your manuscript to address these concerns. We think the referee comments are overall quite constructive. We ask that you focus your efforts on quantitative validation and benchmarking. We will need clear evidence that SpatialGlue outperforms existing methods and/or enables new analyses to send the paper back to referees.

- * include a point-by-point response to the reviewers and to any editorial suggestions
- * please underline/highlight any additions to the text or areas with other significant changes to facilitate review of the revised manuscript
- * address the points listed described below to conform to our open science requirements
- * ensure it complies with our general format requirements as set out in our guide to authors at

www.nature.com/naturemethods

* resubmit all the necessary files electronically by using the link below to access your home page

[Redacted]

We hope to receive your revised paper within three months. If you cannot send it within this time, please let us know. In this event, we will still be happy to reconsider your paper at a later date so long as nothing similar has been accepted for publication at Nature Methods or published elsewhere.

OPEN SCIENCE REQUIREMENTS

REPORTING SUMMARY AND EDITORIAL POLICY CHECKLISTS

DATA AVAILABILITY

We strongly encourage you to deposit all new data associated with the paper in a persistent repository where they can be freely and enduringly accessed. We recommend submitting the data to discipline-specific and community-recognized repositories; a list of repositories is provided here:

<http://www.nature.com/sdata/policies/repositories>

All novel DNA and RNA sequencing data, protein sequences, genetic polymorphisms, linked genotype

and phenotype data, gene expression data, macromolecular structures, and proteomics data must be deposited in a publicly accessible database, and accession codes and associated hyperlinks must be provided in the "Data Availability" section.

CODE AVAILABILITY

Please include a "Code Availability" subsection in the Online Methods which details how your custom code is made available. Only in rare cases (where code is not central to the main conclusions of the paper) is the statement "available upon request" allowed (and reasons should be specified).

For more information on our code sharing policy and requirements, please see: <https://www.nature.com/nature-research/editorial-policies/reporting-standards#availability-of-computer-code>

MATERIALS AVAILABILITY

Authors reporting new chemical compounds must provide chemical structure, synthesis and

characterization details. Authors reporting mutant strains and cell lines are strongly encouraged to use established public repositories.

ORCID

Nature Methods is committed to improving transparency in authorship. As part of our efforts in this direction, we are now requesting that all authors identified as 'corresponding author' on published papers create and link their Open Researcher and Contributor Identifier (ORCID) with their account on the Manuscript Tracking System (MTS), prior to acceptance. This applies to primary research papers only. ORCID helps the scientific community achieve unambiguous attribution of all scholarly contributions. You can create and link your ORCID from the home page of the MTS by clicking on 'Modify my Springer Nature account'. For more information please visit please visit www.springernature.com/orcid.

Sincerely,
Rita

Rita Strack, Ph.D.
Senior Editor
Nature Methods

Reviewers' Comments:

Reviewer #1:
Remarks to the Author:
Dear authors,

First of all, congratulations on designing a model that tackles one of the key challenges in the fast-paced field of spatial omics and on crafting this manuscript. Both the model and the demonstration of its utility in diverse biological contexts seem to have a potential to be of interest to a broad audience and, together with further improvements, to advance the field and to become an integration tool of choice for many datasets and researchers. I hope you will find my comments below useful in order to improve and refine the presentation of the model and the manuscript.

Summary

The manuscript introduces a graph neural network-based model for spatial multi-omics data analysis.

One of the key features of this new model is the combination of measured features (e.g. gene expression or protein abundance) and spatial information related to those measurements. The main output of the trained model is an integrated representation of the dataset learned on both feature measurements and their spatial locations, which can be used for a range of downstream analysis tasks. Importantly, the model is modality-agnostic, and its applications are demonstrated on multiple datasets that measure protein abundance or chromatin accessibility together with gene expression in space.

Strengths

As spatial locations of the measurements can provide additional information about the local spatial context, combining them with the measurements themselves is a well reasoned, and maybe even desired, approach. Moreover, the model uses a multi-step procedure to capture spatial proximity both in the latent space using a feature graph and in the physical space with a spatial graph. Those representations are combined in each modality first before learning a joint cross-modality representation, which allows to achieve a well-structured model. Because of the chosen approach, the model is modality-agnostic, and has been applied in different biological contexts (murine spleen, thymus, brain) for different sequencing-based techniques (SPOTS, Stereo-CITE-seq, spatial ATAC-RNA-seq). Moreover, the framework should be directly applicable to more than two modalities when such datasets become available.

The model addresses the integration challenge by learning an integrated latent space. As such representations are instrumental for many downstream tasks including clustering and visualisation, this is a solid approach that could integrate well with existing analytical workflows.

The framework also enables the evaluation of the contribution of individual modalities to spatial domains using the modality weights, which appears to be a valuable tool for model interpretation.

Major Concerns

The major concerns of the current version of the study revolve around model evaluation including comparisons to existing methods.

1. In particular, a single multi-omics integration method (Seurat WNN) is used for comparisons with the presented model. This might limit the proper evaluation of the performance and utility of the proposed model since (1) spatial information has seemingly not been used as input for Seurat WNN; (2) there are other multi-omics integration methods, which might prove to be more (or less) effective for the described datasets and applications, and including them in the comparison could alter the performance ranking of different integration methods on the discussed datasets; (3) there have been other attempts to combine spatial and omics information such as mentioned below, and it is unclear how the proposed model compares to them, e.g. in terms of its performance to recover biological insights.

For instance a recent review «Methods and applications for single-cell and spatial multi-omics» (<https://www.nature.com/articles/s41576-023-00580-2>) mentions some of the current approaches to spatial multi-omics. [1] One of them presented in Ravi et al., 2022 (<https://doi.org/10.1016/j.jccell.2022.05.009>) describes an approach to first infer spatial regions based on transcriptomics data and then to complement them with other modalities. [2] The approach

published in Velten et al., 2022 (<https://doi.org/10.1038/s41592-021-01343-9>) seems to be directly applicable for integration spatial multi-omics datasets, including spot-based ones such as Visium, as it is described to identify "spatial patterns of variation from multimodal data". While the paper doesn't include multimodal spatial assays such as the ones described in the discussed manuscript, this seems to be a tool at least applicable to, if not "designed specifically for" (as mentioned in the first paragraph of the discussed manuscript), spatial multi-omics. [3] Published in Bao et al., 2022 (<https://www.nature.com/articles/s41587-022-01251-z>), there is another method described previously for "integrative spatial analysis". While it might have a narrower initial scope, it was described to successfully identify anatomical structures and spatial regions as well as to provide a joint multimodal representation of the data.

To better represent the current state of the field, it might be worth mentioning there are other approaches of incorporating spatial information. [1] For instance, while the model is described to combine spatial information with feature matrices, it doesn't use imaging data, which is frequently available and contains more spatial information to take use of (e.g. as in Bergenstråhle et al., 2021, <https://www.nature.com/articles/s41587-021-01075-3>). [2] Another, seemingly more generic approach for combining information including imaging and sequencing data is published in Dai Yang, Belyaeva, et al., 2021 (<https://www.nature.com/articles/s41467-020-20249-2>). The latter study seems to be particularly relevant to the discussed work as it also learns a joint representation of the data using multiple modalities.

Please note that this is merely a list of pointers to a few related studies, and it might be neither exhaustive nor representative of existing methods. I expect the readers who are familiar with some of the currently published methods will be interested in learning if and how SpatialGlue is more generic, efficient, or performant for certain analysis tasks.

2. Following the question of model evaluation, the major results of applying the model to different datasets seem to be presented in the current version of the manuscript in a qualitative rather than in a quantitative manner. A more rigorous definition of the term "spatial domain" and a more rigorous spot enrichment / separation analysis seems to be necessary for better understanding the quality of the results and consequently its potential utility.

For clarity, an example of such qualitative evaluation is the following sentence in the context of describing the model performance on the spleen dataset. "Conversely, the red pulp macrophage enriched spots were better separated in the RNA UMAP..." Observations like that could seem subjective to the reader and make it hard to evaluate and compare models' outputs. More issues with such interpretations can be also found in literature, for instance in the manuscript by Chari and Pachter, 2022 (<https://www.biorxiv.org/content/10.1101/2021.08.25.457696v4>).

3. Following the concerns regarding the choice of baseline models and model evaluation, the current way to present results may not make it clear to the reader how different the conclusions are from the annotations and descriptions of the original papers (that first described those datasets) when the outputs of SpatialGlue for individual datasets are interpreted. E.g. are there novel biological insights, and are they comparable to what's currently been done, even on unimodal data, e.g. using integrative analysis of spatial ATAC-seq data (Deng et al., 2022, <https://www.nature.com/articles/s41586-022-05094-1>)?

4. Clustering seems to be one of the key final steps in order to infer spatial domains using the

integrated representation, however the details of how it was performed in each of the use cases seem to be missing. Related, the term “spatial domains” features in the title, in the abstract, and in the text, however it might not be transparent for the reader if this term refers to the clusters in the joint representation learned by the model and how it relates to the physical space at various scales (local and global structures in tissues).

Minor Concerns

1. The conceptual motivation behind some choices of the model is provided, which also makes the methods section more easily accessible for a wider audience. However it remains unclear if the authors also attempted any alternative ways to combine those information layers (e.g. to integrate the modalities first and then to integrate the joint embedding/graph with the spatial information) and concluded their approach to be more performant in comparison — or if this is rather the sole model that they focused on. At the same time, echoing the concerns listed above, there’s no comparison to methods that use spatial information for learning a latent space so the benefits of using a certain way of incorporating spatial information over the other are currently not reflected in this manuscript.

2. Different categories of spatial technologies referenced in the manuscript (sequencing- and imaging-based) give justice to the diversity of spatial datasets. However, as all the applications focus on a subset of sequencing-based techniques, it is unclear if the current version of the model is also applicable to the imaging-based spatial datasets. Are there any additional challenges there?

3. Would the utilised approach to combine spatial information with feature matrices be useful for unimodal datasets as well? If so, and if the method is "scalable", readers might expect this to be demonstrated by being applied to a larger spatial dataset. This might also aid with having a richer choice of the baseline method(s) to compare SpatialGlue with.

Related, SpaGCN and GraphST are only referenced but no comparison to them or related discussion has been presented. Considering the fact that multimodal methods typically stand on the shoulders of unimodal methods, it might be useful to mention what approaches SpatialGlue borrows from GraphST, SpaGCN, STAGATE, or other methods and what new things SpatialGlue brings to the table.

4. It might be beneficial for the manuscript to reiterate the specifics of 10x Visium-derived and other technologies used for the described datasets (e.g. spatial resolution, imaged area size, where applicable – spot size, distance between spots, etc.).

5. It is mentioned that the outputs of the model can be used “in various downstream analysis, including clustering, visualisation, and DGE detection”. Readers might appreciate a slightly more expanded description of how these tasks can be performed and/or references to the relevant methods.

6. In the main text:

6.1. In paragraph 1, “... feature counts that can vary enormously” — might benefit from changing to more quantitative and precise terms as it is not clear what exactly this refers to.

6.2. In paragraph 1, “... there is no tool designed specifically for spatial multi-omics” — as mentioned above, there are tools and approaches that can handle spatial multi-omics datasets though they might have a different scope compared to this model.

6.3. In paragraph 1, “existing tools <...> target spatial single omics integrated analysis, while GLUE and Seurat WNN perform multi-omics data integration...” refers to a few methods however there are

- more spatial (e.g. benchmarked in <https://www.nature.com/articles/s41592-022-01480-9>) and multi-omics (e.g. reviewed in <https://www.nature.com/articles/s12276-020-0420-2>, benchmarked for cancer studies in <https://www.nature.com/articles/s41467-020-20430-7>) integration methods. Moreover, it might be worth noting that GLUE is originally tailored to unpaired multi-omics data.
- 6.4. In paragraph 2, the term "spot" is mentioned without any accompanying context ("Similarly, the different data modalities can have distinct and complementary contributions to each spot"). This, together with Figure 1a labels, also already suggests spot-based sequencing methods to be central for this work; however both sequencing- and imaging-based assays are referenced in the previous paragraph.
- 6.5. In paragraph 2, in "... than the common approaches of summation or concatenation", it is unclear what approaches are referred to here. As this model is referred to as the first tool for spatial multi-omics in the previous paragraph, it might be worth expanding (via text and/or citations) what common approaches are implied here.
- 6.6. Panel 1b doesn't seem to provide much information in its current state. Moreover, it is unclear why H&E staining appears black-and-white.
- 6.7. In paragraph 3, "... clusters clearly did not align..." provides an observation that is not further corroborated by any quantifications.
- 6.8. Related, Figure 1c displays 6 clusters in each modality. There are no details on the clustering procedure. Assuming clustering was performed on each modality individually, colour coding is different for different panels of Figure 1c, which might be confusing for the reader (i.e. cluster 1 spots for the RNA UMAP are not the same as cluster 1 spots for the Protein UMAP, etc.).
- 6.9. In paragraph 3, "... plotted the clusters obtained from the integrated analysis onto the individual modalities' UMAPs..." the language could be improved: effectively, spots in the individual modalities' UMAP spaces were plotted and colours by their cluster identity.
- 6.10. Spot enrichment / separation statements at the end of the paragraph 3 as well as analogous statements in the following text are not supported by any quantifications. Performing and displaying those would greatly improve the understanding of the performance of the model and consequently its appeal.
- 6.11. In paragraph 5, neighbourhood enrichment analysis is mentioned however the details of the analysis seem to be missing from the methods section. The respective Figure 1i could be potentially improved by adjusting the labels for the colour scale as the two-digit precision doesn't seem to be meaningful.
- 6.12. Similarly, the details of the co-occurrence score calculation could be useful for the readers.
- 6.13. For comparisons of the modality weights for SpatialGlue and Seurat WNN (e.g. Figure 1e vs Supplementary Figure S1d), it is unclear if there's a systematic difference in terms of how these two methods learn modality weights, which in turn influences how these weights should be interpreted and if they are directly comparable.
- 6.14. Figure 2c denotes clusters as numbers while tissue structures are named in the corresponding text (paragraph 6), which makes it hard to relate those two.
- 6.15. In paragraph 6, "... the clusters showed greater fragmentation, especially in the cortex" provides an observation that is not supported with any quantifications.
- 6.16. In paragraph 7, "... grainy without clear boundaries between regions" provides an observation that is not supported with any quantifications.
- 6.17. In paragraph 8, when speaking about computational efficiency, it might be beneficial to also demonstrate how SpatialGlue scales depending on the number of spots, features or principal components, modalities as well as to provide similar scalability properties for the baseline method(s).

7. In the methods section:

- 7.1. It is unclear why the choice of 50 principal components for the gene expression modality of the SPOTS dataset as the input of the encoder is consistent with the input dimension with the ADT data, which has 21 measurements.
- 7.2. For the Stereo-CITE-seq dataset, "in fewer than 10 cells" and "expressed in fewer than 50 cells" should probably say "in fewer than 10 spots" and "expressed in fewer than 50 spots", respectively.
- 7.3. For the Spatial-ATAC-RNA-seq dataset, "2,2914" probably means "22,914" and "fewer than 200 cells" probably means "fewer than 200 spots".
- 7.4. For model training, it might be worth expanding how exactly the weight factors "vary between different spatial multi-omics technologies". Related, it is unclear how impactful the chosen weight factors are for the final model. Individual values are mentioned in the implementation details however it is unclear how the user should choose or adjust those.
- 7.5. For the correspondence loss, do the terms $Y_{\{1\}^{\{l-1\}}}$ $W_{\{d2\}^{\{l-1\}}}$ and $Y_{\{2\}^{\{l-1\}}}$ $W_{\{d1\}^{\{l-1\}}}$ imply that all modalities should have the same number of features when provided as input to the encoder? If yes, does this limit the utility of the method in scenarios when modalities have vastly different numbers of features?
- 7.6. For the adversarial loss, it is unclear from the text how it should be extended to three or more modalities though the framework is described to be extensible to more than two modalities.
- 7.7. For the analysis of the Spatial-ATAC-RNA dataset, it is stated that 50 principal components were used for SpatialGlue while only 10 were used for Seurat WNN. Together with other differences in data processing for the two methods that are then directly compared, this raises the question of ensuring that the presented discrepancies between the two methods are not influenced or even fully explained by the data processing choices.

For the code availability, it currently reads "GraphST", which should probably be replaced by "SpatialGlue".

Conclusion

With the growing scale and complexity of experimental designs, spatial omics datasets pose a lot of analytical challenges with data integration in its centre. This reiterates the relevance of the discussed work, and I expect it to be of interest to those readers who are interested in spatial omics as well as a broader audience thanks to the attempts to derive biological insights from different biological contexts (murine spleen, thymus, brain profiled with different technologies). Presenting the model as part of the complex landscape of the currently available spatial integration methods by providing rigorous model evaluations as well as quantifying clustering performance and differences to other baseline methods could improve the understanding of the model's merits and help other researchers to choose SpatialGlue for spatial omics integration when it's appropriate.

Finally, thank you for the opportunity to review this work!

Sincerely,
Reviewer

Reviewer #2:
Remarks to the Author:

In this manuscript Long et al. present SpatialGlue, a novel method for spatial multimodal data integration using graph neural networks and attention mechanisms. The manuscript is well written and with the current availability of spatial multi-omics technologies is highly relevant to the field. The proposed approach seems promising, however I do have some concerns/remarks. My main concern is due to the lack of real quantification of the performance of the approach and on how this will affect end-users, as most of the analyses presented in the manuscript seem to be more qualitative based on the anecdotal examples presented by the authors:

Major:

- The authors state that: "We believe this approach enables more accurate integration than the common approaches of summation or concatenation." This statement should be properly backed up with quantitative evidence. Not only should the authors compare to simple integration strategies, such as concatenation, the manuscript should consider and compare to state-of-the-art methods such as, MOFA+ by Arguelaguet et al. (2020, Genome Biology), MEFISTO by Velten et al. (2022, Nature Methods), multiVI by Ashuach et al. (2023, Nature Methods), scMM by Minoura et al. (2021, Cell Reports). If the integration of the spatial component in SpatialGlue, which many of these methods lack, delivers a more meaningful integration, its contribution should be clearly and quantitatively demonstrated. For example, what are the attention weights for the spatial neighbourhood graph vs. the feature graph? The reader should be convinced of the added value of 1) the integration of the different modalities, and 2) the incorporation of spatial information to each multi-modal data point.
- Although it's nice that the authors illustrate their method with 3 different datasets using different technologies, the performance of the approach still seems very anecdotal as only 1 tissue slice seems to be analysed for each technology. Is this the best performing slice? Was the performance consistent across slices? Are the learned representations stable across slices of the same tissue?
- Much of the analysis seems to rely on the accurate representation of known anatomy and on how this is captured by unsupervised clustering. When the authors state that "The clusters clearly did not align..." between modalities, how did the authors choose the appropriate clustering resolution to make that comparison? Choosing the resolution with the same number of clusters seems to be overly simplistic, as it's difficult to predict if at a lower/higher resolution a more optimal alignment might be achieved (even if more/less clusters are called). These sort of statements should also be quantitatively backed by known measures of cluster overlap/separation (e.g. Jaccard index, cluster overlap, silhouette scores etc.). The same holds true for the comparisons made with Seurat WNN and other tools.
- The authors state several times the contribution of both modalities to the integrated space, however in the methods section it is stated that to account "for differences in feature distributions across the datasets", weight factors were assigned. What's the relationship between the modality weights and these hyperparameters?
- The authors use an interesting deep learning conceptual framework, however they seem to limit the input dimensionality drastically by performing linear dimensionality reduction prior to applying their approach. Isn't the strength of deep learning approaches that they don't have to deal with prior feature engineering, and can tackle nonlinearities often found in highly complex biological datasets? What happens if the GNN is given the complete input vector instead of the PCA reduced one? What is the effect of different methods of dimensionality reduction methods?

Minor:

- The abstract is rather vague, what is meant by "SpatialGlue can accurately aggregate cell types into spatial domains at a higher resolution across different tissue types"?
- "These constructed graphs can possess unique semantic information that should be integrated."
What kind of unique semantic information is referred to here?

Reviewer #3:

Remarks to the Author:

Review - Deciphering spatial domains from spatial multi-omics with SpatialGlue

The paper proposes a way to perform multi-modal integration in the context of spatial data. Graph neural networks are employed to represent data across multiple modalities, in feature and space dependent manners, respectively. Attention mechanisms are used to first weigh the contribution of feature specific and spatial specific representations within modalities, and then to weigh the contribution of modalities themselves.

Related approaches have been proposed in a number of different applications, see for example Multi-Modal Graph Neural Network for Joint Reasoning on Vision and Scene Text, Gao et al. and Multimodal learning with graphs, Ektefaie et al. Most similar is perhaps Graph Neural Networks for Multimodal Single-Cell Data Integration, Wen et al. The latter is particularly interesting as it does not employ an attention mechanism to solve related tasks.

While the approach is elegant, my feeling is that the paper relies too much on state-of-the-art concepts in machine learning (attention, GNN, adversarial training) without properly evaluating the need for each of the building blocks used. The study lacks benchmarking, there is no simulation scenario to help build intuition on when the different components are needed, and a quantitative evaluation section with metrics beyond visualization is lacking. Below I provide some ideas on how to improve along these dimensions, however I am aware that this would require significant rewriting.

Major points.

Benchmarking. The paper currently lacks comparison with other simpler and related methods (see above, other ML communities). Within the spatial transcriptomics field, a paper I particularly like is Stabilized mosaic single-cell data integration using unshared features, by Ghazanfar et al. This paper would be a fantastic point of comparison! Comparing spatialGlue with StabMap on the StabMap datasets would offer insight into the advantages of spatialGlue. I would also be interested in understanding if removing the attention mechanism and concatenating the feature centric and spatial representations as representations would lead to a much worse performance.

Simulations. A thought on generating spatial data from multiple modalities in order to have some notion of ground truth for evaluation is to use the generative model in Nonnegative spatial factorization applied to spatial genomics by Townes et al. To generate different modalities, one could sample different subsets of columns (intersecting or not) like in StabMap, and transform them (add

noise, nonlinearities etc).

Metrics. Quantitative metrics such as predicting cell types from representations, predicting spatial domains, preservation of distances in the joint space should be considered and reported for benchmarks and variants of the algorithm.

Algorithm variants and robustness. Many parameters could have a strong impact on learning (given the appropriate metrics, see above). These include: the number (50) of PCA neighbors, the number of neighboring spots considered (6), number of GNN layers (l).

Optimization details. Not enough details are provided regarding the adversarial training and the corresponding loss. How are the hyperparameters optimized? Which latent representation is used to compute the probabilities p_i ? What is the performance of the algorithm in the absence of an adversarial loss? This line of reasoning applies to the other losses in equation 18 as well. Addressing these questions would significantly improve the quality of the paper!

Minor points. While, the paper reads well generally, there are a few typos:

In equations 6, 7 α and a are used interchangeably

$W_{\{Wi\}}$ is confusing, perhaps another letter could be used

Eq 7, should be Y_i^m and not Y_i^t

Be consistent with capitalization (I take Y is a matrix, and y is a vector). These are sometimes interchanged.

In (11) and (12): would using A_s^{-1} instead of A_s have an impact? I am worried that multiplying by A_s too many times is equivalent to dividing by the $(\text{node_degree})^l$ of a node i , with l the number of layers, which can lead to values of zero very fast.

What is the impact on overall performance of using two nonlinear activation functions in 15 and 16?

How affected is the performance of the algorithm by choosing different activation functions?

Author Rebuttal to Initial comments
--

Response to comments for paper NMETH-BC52548 “Deciphering spatial domains from spatial multi-omics with SpatialGlue”

Editor's comment: “We think the referee comments are overall quite constructive. We ask that you focus your efforts on quantitative validation and benchmarking. We will need clear evidence that SpatialGlue outperforms existing methods and/or enables new analyses to send the paper back to referees.”

Responses to editor's comment: We have perused the comments from the reviewers, and we agree that the comments are valuable in helping us improve the manuscript. The following is a summary of the changes that we have made.

Summary of revision

Comprehensive benchmarking. We agree that a comprehensive quantitative benchmarking is very important. In the revised manuscript, we have included simulation data, new experimentally acquired data, additional publicly available data, quantitative evaluation, and compared with more competing methods.

1. **Generating new datasets:** To perform quantitative evaluation, ground truth labeling of cell types or anatomical structures are usually required. For the datasets used in our initial manuscript, ground truth labels are not provided in the original studies, and the resolution of the associated images is not high enough for a domain expert to generate manual labeling. As such, we have acquired two new datasets from human lymph node tissues using the latest 10x Visium CytAssist Spatial Gene and Protein Expression technology. We also obtained high resolution H&E images with which our clinician collaborator manually annotated the tissue structures. We used the manual annotations as ground truth to quantitatively evaluate the clusters produced by different algorithms.
2. **Including additional publicly available datasets:** We also downloaded more datasets from public data repositories. We had 4 datasets in our original submission. Now, we have 12 different datasets (Table R1), including 2 mouse spleen SPOTS, 4 mouse thymus Stereo-CITE-seq, 4 mouse brain spatial-epigenome-transcriptome, and 2 human lymph node 10x Visium datasets.
3. **Simulation data:** We adopted the approach outlined by Townes et al. to simulate spatial multi-omics data. The simulation recapitulated the ZINB and NB distribution of spatial transcriptomics and proteomics respectively and matched the cells from the two modalities. To better mimic real-world scenarios, we added Gaussian distributed noise. To increase statistical analysis power, we generated 5 simulation datasets with different parameters.
4. **Quantitative metrics:** We introduced multiple quantitative metrics for evaluation in the revised version of manuscript. For datasets that come with H&E images (dataset 11-12 human lymph node), our domain expert collaborators have used H&E to generate manual annotation to provide a ground truth reference. For such datasets with ground truth labels and simulation data, we have introduced quantitative metrics such as AMI, NMI, ARI, Homogeneity, Mutual information, and V measure to assess the accuracy of predicting cell types or spatial domains from joint representations. For experimental datasets for which we cannot obtain ground truth labelling, we used the “unsupervised” metrics such as Jaccard Similarity to evaluate the preservation of distance in the joint space, and Moran's I score to evaluate the spatial autocorrelation of clusters produced with different algorithms.
5. **Compare to more methods:** We now compared SpatialGlue to 7 competing methods, namely Seurat, MOFA+, MEFISTO, totalVI, MultiVI, scMM, and StabMap. All methods can perform cross omics integration. Only MEFISTO and SpatialGlue can also integrate the spatial information, meanwhile, integrate across omics.
6. **Ablation studies:** We added ablation studies to justify our design choices for the structure of SpatialGlue and assess the robustness of SpatialGlue.

7. Time and memory efficiency: We have also evaluated the time usage of each tested method.

Further analysis of the mouse brain Spatial-epigenome-transcriptome data: We have also used SpatialGlue outputs to perform further analysis to derive new biological insights that were not available in the original study and can't be obtained with other algorithms. Specifically, we identified four cortical layers from the mouse brain spatial-epigenome-transcriptome data while the original study only identified two layers, inner and outer layers. We performed further analyses to identify differential expressions and peaks to dissect gene regulation in the different cortical layers.

Table R1. Experimental datasets used in the revised manuscript.

Dataset	Name	Platform	Size (spots x genes/proteins/peaks)	Figure
Dataset1	Mouse spleen replicate1	SPOTS (RNA-protein)	2,568x32,285 2,568x21	Figure 3e-l, Figure S24, Figure S25a-c
Dataset2	Mouse spleen replicate2	SPOTS (RNA-protein)	2,768x32,285 2,768x21	Figure S25d-h
Dataset3	Mouse Thymus1	Stereo-CITE-seq (RNA-protein)	4,697x23,622 4,697x51	Figure 3a-d, Figure S16, Figure S20a, Figure S21
Dataset4	Mouse Thymus2	Stereo-CITE-seq (RNA-protein)	4,253x23,529 4,253x19	Figure S17, Figure S20b, Figure S22
Dataset5	Mouse Thymus3	Stereo-CITE-seq (RNA-protein)	4,646x23,960 4,646x19	Figure S18, Figure S20c, Figure S23a
Dataset6	Mouse Thymus4	Stereo-CITE-seq (RNA-protein)	4,228x23,221 4,228x19	Figure S19, Figure S20d, Figure S23b
Dataset7	Mouse Brain RNA ATAC P22	Spatial-epigenome-transcriptome	9,215x22,914 9,215x121,068	Figure 2a-e, Figure S12a,c,e
Dataset8	Mouse Brain RNA H3K4me3	Spatial-epigenome-transcriptome	9,548x22,731 9,548x35,270	Figure S14a-e
Dataset9	Mouse Brain RNA H3K27ac	Spatial-epigenome-transcriptome	9,370x23,415 9,370x104,162	Figure 2f-l, Figure S12b,d,f, Figure S13
Dataset10	Mouse Brain RNA H3K27me3	Spatial-epigenome-transcriptome	9,752x25,881 9,752x70,470	Figure S15
Dataset11	Human Lymph Node A1	10x Visium (RNA-protein)	3,484x18,085 3,484x31	Figure 1g-k, Figure S8a-c, Figure S9a, Figure S10
Dataset12	Human Lymph Node D1	10x Visium (RNA-protein)	3,359x18,085 3,359x31	Figure S8d-k, Figure S9b, Figure S11,
Dataset13	Simulation 1	NSF (Townes et al., 2022)	1,296x1,000 1,296x100	Figure 1b-f, Figure S1a
Dataset14	Simulation 2	NSF (Townes et al., 2022)	1,296x1,000 1,296x100	Figure S1b-f
Dataset15	Simulation 3	NSF (Townes et al., 2022)	1,296x1,000 1,296x100	Figure S2

Dataset16	Simulation 4	NSF (Townes et al., 2022)	1,296x1,000 1,296x100	Figure S3a-e
Dataset17	Simulation 5	NSF (Townes et al., 2022)	1,296x1,000 1,296x100	Figure S3f-j
Dataset18	Simulation 6 (triplet omics)	NSF (Townes et al., 2022)	1,296x1,000 1,296x100	Figure S4

Reviewer #1 (Remarks to the Author):

First of all, congratulations on designing a model that tackles one of the key challenges in the fast-paced field of spatial omics and on crafting this manuscript. Both the model and the demonstration of its utility in diverse biological contexts seem to have a potential to be of interest to a broad audience and, together with further improvements, to advance the field and to become an integration tool of choice for many datasets and researchers. I hope you will find my comments below useful in order to improve and refine the presentation of the model and the manuscript.

Summary: *The manuscript introduces a graph neural network-based model for spatial multi-omics data analysis. One of the key features of this new model is the combination of measured features (e.g. gene expression or protein abundance) and spatial information related to those measurements. The main output of the trained model is an integrated representation of the dataset learned on both feature measurements and their spatial locations, which can be used for a range of downstream analysis tasks. Importantly, the model is modality-agnostic, and its applications are demonstrated on multiple datasets that measure protein abundance or chromatin accessibility together with gene expression in space.*

Strengths: *As spatial locations of the measurements can provide additional information about the local spatial context, combining them with the measurements themselves is a well reasoned, and maybe even desired, approach. Moreover, the model uses a multi-step procedure to capture spatial proximity both in the latent space using a feature graph and in the physical space with a spatial graph. Those representations are combined in each modality first before learning a joint cross-modality representation, which allows to achieve a well-structured model. Because of the chosen approach, the model is modality-agnostic, and has been applied in different biological contexts (murine spleen, thymus, brain) for different sequencing-based techniques (SPOTS, Stereo-CITE-seq, spatial ATAC-RNA-seq). Moreover, the framework should be directly applicable to more than two modalities when such datasets become available.*

The model addresses the integration challenge by learning an integrated latent space. As such representations are instrumental for many downstream tasks including clustering and visualization, this is a solid approach that could integrate well with existing analytical workflows.

The framework also enables the evaluation of the contribution of individual modalities to spatial domains using the modality weights, which appears to be a valuable tool for model interpretation.

Response: *We would like to thank the reviewer for the many insightful comments and valuable corrections to help improve our work. We have carefully gone through all the comments and suggestions to improve SpatialGlue. Based on the reviews received, we redesigned and performed benchmarking studies to assess the performance of SpatialGlue. In particular, we have added various relevant competing methods to test alongside SpatialGlue. To ensure a comprehensive benchmarking, different metrics were added for both annotated and unannotated data. To reflect the changes, we also extensively modified the manuscript with the new results and made clarifications to improve the manuscript's readability.*

Major Concerns: *The major concerns of the current version of the study revolve around model evaluation including comparisons to existing methods.*

Comment 1.1: *In particular, a single multi-omics integration method (Seurat WNN) is used for comparisons with the presented model. This might limit the proper evaluation of the performance and*

utility of the proposed model since (1) spatial information has seemingly not been used as input for Seurat WNN; (2) there are other multi-omics integration methods, which might prove to be more (or less) effective for the described datasets and applications, and including them in the comparison could alter the performance ranking of different integration methods on the discussed datasets; (3) there have been other attempts to combine spatial and omics information such as mentioned below, and it is unclear how the proposed model compares to them, e.g. in terms of its performance to recover biological insights.

For instance a recent review «Methods and applications for single-cell and spatial multi-omics» (<https://www.nature.com/articles/s41576-023-00580-2>) mentions some of the current approaches to spatial multi-omics. [1] One of them presented in Ravi et al., 2022 (<https://doi.org/10.1016/j.ccell.2022.05.009>) describes an approach to first infer spatial regions based on transcriptomics data and then to complement them with other modalities. [2] The approach published in Velten et al., 2022 (<https://doi.org/10.1038/s41592-021-01343-9>) seems to be directly applicable for integration spatial multi-omics datasets, including spot-based ones such as Visium, as it is described to identify "spatial patterns of variation from multimodal data". While the paper doesn't include multimodal spatial assays such as the ones described in the discussed manuscript, this seems to be a tool at least applicable to, if not "designed specifically for" (as mentioned in the first paragraph of the discussed manuscript), spatial multi-omics. [3] Published in Bao et al., 2022 (<https://www.nature.com/articles/s41587-022-01251-z>), there is another method described previously for "integrative spatial analysis". While it might have a narrower initial scope, it was described to successfully identify anatomical structures and spatial regions as well as to provide a joint multimodal representation of the data.

To better represent the current state of the field, it might be worth mentioning there are other approaches of incorporating spatial information. [1] For instance, while the model is described to combine spatial information with feature matrices, it doesn't use imaging data, which is frequently available and contains more spatial information to take use of (e.g. as in Bergenstr hle et al., 2021, <https://www.nature.com/articles/s41587-021-01075-3>). [2] Another, seemingly more generic approach for combining information including imaging and sequencing data is published in Dai Yang, Belyaeva, et al., 2021 (<https://www.nature.com/articles/s41467-020-20249-2>). The latter study seems to be particularly relevant to the discussed work as it also learns a joint representation of the data using multiple modalities.

Please note that this is merely a list of pointers to a few related studies, and it might be neither exhaustive nor representative of existing methods. I expect the readers who are familiar with some of the currently published methods will be interested in learning if and how SpatialGlue is more generic, efficient, or performant for certain analysis tasks.

Response 1.1: Thank you for highlighting some of latest development in the field. We have perused the suggested articles and conducted further searches in the literature. In the revised manuscript, we expanded our coverage of methods compared to Seurat WNN, MOFA+, totalVI, MultiVI, scMM, StabMap, and MEFISTO. The former six are state-of-the-art integration methods for single-cell multi-omics data and do not make use of spatial information. MEFISTO, presented by Velten et al., 2022 (<https://doi.org/10.1038/s41592-021-01343-9>), is a generic factor analysis method for multimodal data that can capture spatial or temporal dependencies between data points. MEFISTO indeed can be applied to spatial multi-omics data, although it was not specifically designed for that purpose nor its usage on such data was demonstrated.

The study by Ravi et al., 2022 (<https://doi.org/10.1016/j.ccell.2022.05.009>) did use spatial multi-omics including transcriptomics, metabolomics, and proteomics. However, the various omics were obtained from adjacent tissue slides instead of the same slide. Handling spatial multi-omics data acquired from different slides is out of the scope of SpatialGlue that was designed for multi-omics simultaneously obtained from the same tissue slide. To process the different omics data acquired from serial sections, Ravi et al. developed the SPATA (SPAtial Transcriptomic Analysis) framework. With SPATA, the author first used SURF, an image registration algorithm, to align the H&E images of the metabolomic and transcriptomic slides, and then integrated the metabolomic and transcriptomic data using WNN from the Seurat package. In the WNN integration, spatial information is not utilized such that the spatial correlations between spots are not taken into account. As we have benchmarked SpatialGlue against Seurat WNN, we do not compare SpatialGlue with SPATA.

Different from SPATA, SpatialGlue is designed for spatial multi-omics that simultaneously measures different omics of the same tissue slide, and it integrates the different omics in a spatially informed manner. It achieves that with a dual-attention mechanism. The first attention integrates the omics data with spatial information to produce a spatially aware representation of omics and subsequently the second attention integrate across omics. We have clarified this in the revised introduction. Lately, more and more studies have presented spatial multi-omics data from adjacent sections. That has urged us to develop another novel method called SpatialFusion to handle such datasets. We will present SpatialFusion in another manuscript.

Bao et al., 2022 (<https://www.nature.com/articles/s41587-022-01251-z>) presented MUSE, which was designed for integrating spatial transcriptomics and imaging (hematoxylin and eosin or fluorescence microscopy) modalities. Bergenstr hle et al., 2021 presented XFUSE that integrates spatial gene expression with histological image data from the same tissue section to infer higher-resolution expression maps. Yang et al., 2021 used a cross-modal autoencoder model to integrate and translate between single-cell imaging and sequencing data. We agree with reviewer that the current version of SpatialGlue does not integrate imaging information. Therefore, we do not compare SpatialGlue with XFUSE, MUSE, and other similar methods that employ the image modality. However, we plan to incorporate imaging modality into SpatialGlue version 2 in the near future.

Comment 1.2: *Following the question of model evaluation, the major results of applying the model to different datasets seem to be presented in the current version of the manuscript in a qualitative rather than in a quantitative manner. A more rigorous definition of the term "spatial domain" and a more rigorous spot enrichment / separation analysis seems to be necessary for better understanding the quality of the results and consequently its potential utility.*

For clarity, an example of such qualitative evaluation is the following sentence in the context of describing the model performance on the spleen dataset. "Conversely, the red pulp macrophage enriched spots were better separated in the RNA UMAP..." Observations like that could seem subjective to the reader and make it hard to evaluate and compare models' outputs. More issues with such interpretations can be also found in literature, for instance in the manuscript by Chari and Pachter, 2022 (<https://www.biorxiv.org/content/10.1101/2021.08.25.457696v4>).

Response 1.2: We agree with the reviewer that term "spatial domain" is not defined clearly in the original manuscript. We define spatial domain to be spatial regions with similar measured features in the different omics. We have elaborated this in the revised manuscript.

We also agree that UMAP visualization can be misleading and the sentence "Conversely, the red pulp macrophage enriched spots were better separated in the RNA UMAP..." sound subjective. We have added a quantitative metric, fraction of nearest neighbours (Chari, Tara, and Lior Pachter. 2023), to support this statement. This metric calculates the fraction of nearest neighbors for each spot with same label as spot itself. We applied this metric to the mouse spleen SPOTS data and found that the RNA original expression showed a higher fraction of nearest neighbors than the protein original expression for the red pulp macrophage enriched spots (Figure R1.1 (b)), suggesting that the red pulp macrophage enriched spots were better separated in the RNA modality. To pursue a more quantitative benchmarking focus on the clustering results, we have removed the qualitative discussions on the UMAPs from the main text.

Moreover, we also observed high concordance between the fraction of nearest neighbor metric and the modality weights derived by SpatialGlue for all five cell types including T cell, B cell, RpM Φ , MZM Φ , and MMM Φ (Figure R1.1 (b) and (c)). This indicates that SpatialGlue's attention weights do capture the importance of each modality correctly. For example, the SpatialGlue derived modality weights (Figure R1.1 (c)) suggested that the MMM Φ enriched spots are better separated in the protein modality. The greater contribution of protein modality can be also validated by its higher fraction of nearest neighbors than the RNA modality (Figure R1.1 (b)).

Figure R1.1. (a) Comparison of UMAP plots of RNA and protein modalities and SpatialGlue integrated output, all colored by annotated clusters obtained from the integrated output. (b) Comparison of fraction of nearest neighbors metric calculated by different feature spaces, i.e., original RNA and protein expressions, for each annotated clusters. (c) Modality weight to see the contribution of each modality to the integrated representation.

SpatialGlue learns a within-modality spatially aware latent representation for each modality using its first level of attention integration. Here, we included another two quantitative metrics, Jaccard distance (Chari, Tara, and Lior Pachter. 2023) and number of nearest cells metric (Shila, et al., 2023), to estimate the concordance between the RNA and protein modality in the original and latent space. Higher concordance was observed between the two modalities in the latent representation than the original expression (Figure R1.2 (a) and (b)), suggesting that SpatialGlue allows the modality-specific latent representation to capture some information from the other modality and thus increases the cross-modality concordance.

Figure R1.2. (a) Density curves displaying the cumulative distribution of Jaccard distance (dissimilarity) between RNA and protein features in the original or latent spaces. Lower value means better performance. (b) Bar plot displaying the cumulative number of RNA-resolved spots, grouped by the number of unmatched protein-resolved cells found to be nearer than the matched protein-resolved cell.

Lastly but most importantly, we added multiple quantitative metrics for evaluation in the revised version of manuscript. For datasets that come with H&E images (dataset 11-12 human lymph node), our domain expert collaborators have used the H&E images to generate manual annotation to provide a ground truth reference. For such datasets with ground truth labels and simulation data, we have introduced quantitative metrics such as AMI, NMI, ARI, Homogeneity, Mutual information, and V measure to assess

the accuracy of predicting cell types or spatial domains from joint representations. For experimental datasets for which we cannot obtain ground truth labelling, we used the “unsupervised” metrics such as Jaccard Similarity to evaluate the preservation of distance in the joint space, and Moran’s I score to evaluate the spatial autocorrelation of clusters produced with different algorithms. We have also provided detailed descriptions of these metrics in the revised manuscript.

Reference

- [1]. Chari, Tara, and Lior Pachter. "The specious art of single-cell genomics." PLOS Computational Biology 19.8 (2023): e1011288.
- [2]. Ghazanfar, Shila, Carolina Guibentif, and John C. Marioni. "Stabilized mosaic single-cell data integration using unshared features." Nature Biotechnology (2023): 1-9.

Comment 1.3: *Following the concerns regarding the choice of baseline models and model evaluation, the current way to present results may not make it clear to the reader how different the conclusions are from the annotations and descriptions of the original papers (that first described those datasets) when the outputs of SpatialGlue for individual datasets are interpreted. E.g. are there novel biological insights, and are they comparable to what’s currently been done, even on unimodal data, e.g. using integrative analysis of spatial ATAC-seq data (Deng et al., 2022, <https://www.nature.com/articles/s41586-022-05094-1>)?*

Response 1.3: We thank the reviewer for raising this critical issue. We agree with the reviewer that previous results do not make it clear to the reader how different the conclusions are between SpatialGlue and the original paper. In response to the reviewer’s comments, we have demonstrated the novel biological insights revealed by SpatialGlue on two different datasets from two tissue types and distinct technological platforms, i.e., mouse brain H3K27me3 + RNA and mouse spleen protein + RNA. On the mouse brain dataset, the original study identified only two layers in the cortex, inner and outer layers (Figure R1.3a). The clusters identified by SpatialGlue showcase four cortical layers, namely cluster 1, 5, 6, 12, with clear boundaries (Figure R1.3b, right). We also performed downstream analyses to identify differential expressions and peaks to dissect gene regulation in the different cortical layers (Figure 2, S13). On the mouse spleen data, SpatialGlue identified more macrophage subsets compared to the clustering from the original paper (Figure R1.4).

Figure R1.3. Spatial plots of the RNA and CUT&Tag (H3K27me3) data modalities in mouse brain P22 dataset. Original study by Zhang et al. (a) and SpatialGlue’s clustering output (b).

Figure R1.4. (a) Spatial clustering from the original paper in mouse spleen replicate 1 dataset. (b) Spatial clustering from SpatialGlue.

Comment 1.4: *Clustering seems to be one of the key final steps in order to infer spatial domains using the integrated representation, however the details of how it was performed in each of the use cases seem to be missing. Related, the term “spatial domains” features in the title, in the abstract, and in the text, however it might not be transparent for the reader if this term refers to the clusters in the joint representation learned by the model and how it relates to the physical space at various scales (local and global structures in tissues).*

Response 1.4: We thank the reviewer for the comments. Clustering in our model consists of two main steps. We first derive the integrated representation vector from the SpatialGlue model. With the integrated representation vector as input, a well-known clustering algorithm ‘mclust’ is implemented to obtain clusters according to specified number of clusters. In response to the reviewer’s comments, we have provided the details of clustering in the revised version of the manuscript.

We apologize that our wording become a source of confusion regarding our use of the term “spatial domains”. Indeed, the term “spatial domains” in the manuscript refers to the clusters identified by the joint representation. As SpatialGlue is a spatially informed method, the integrated representation includes spatial information. Consequently, the clusters will likewise be spatially informed.

Minor Concerns:

Comment 1.5: *The conceptual motivation behind some choices of the model is provided, which also makes the methods section more easily accessible for a wider audience. However it remains unclear if the authors also attempted any alternative ways to combine those information layers (e.g. to integrate the modalities first and then to integrate the joint embedding/graph with the spatial information) and concluded their approach to be more performant in comparison — or if this is rather the sole model that they focused on. At the same time, echoing the concerns listed above, there’s no comparison to methods that use spatial information for learning a latent space so the benefits of using a certain way of incorporating spatial information over the other are currently not reflected in this manuscript.*

Response 1.5: We thank the reviewer for the constructive comments. We acknowledge that we did not discuss why we first integrates the omics data with spatial information to produce a spatially aware representation of omics and subsequently integrate across omics. We also agree that one alternative way is to integrate the modalities first and then to integrate the joint embedding with the spatial information. Here we would like to discuss the conceptual motivation behind this choice. With the choice we made for SpatialGlue, we can capture modality-specific spatial correlations between spots and integrate the spatial information in a modality-specific manner. While with the alternative option, we are assuming that the two modalities share the same inter-spot spatial correlation patterns. Based on our observations of the example datasets, the spatial correlations between spots are likely to be different in different omics modalities. For instance, as we demonstrated in the response to Comment 1.2, the red pulp macrophage (RpMΦ) enriched spots were better separated in the RNA modality and showed higher fraction of nearest neighbors than in the protein modality. In contract, the MMMΦ enriched spots are better separated in the protein modality. The RpMΦ and MMMΦ are known to occupy different

regions and present distinct spatial distributions in the mouse spleen, which can be better captured by SpatialGlue's integration strategy.

Furthermore, with its current integration strategy and formulation of loss functions, SpatialGlue can learn a within-modality spatially aware latent representation for each modality. As we demonstrated in the response to Comment 1.2, higher concordance was observed between the two modalities in the latent representation than the original expression (Figure R1.2 (a) and (b)), suggesting that SpatialGlue allows the modality-specific latent representation to capture some information from the other modality and thus increases the cross-modality concordance. Subsequently, with the second level of attention, the cross-omics integration is further strengthened.

In the revised manuscript, we have also compared SpatialGlue to alternative methods that use spatial information for learning a latent space, including MEFISTO, STAGATE, SpaGCN, and GraphST. MEFISTO is a generic factor analysis method for multimodal data that can capture spatial or temporal dependencies between data points. MEFISTO indeed can be applied to spatial multi-omics data, although it was not specifically designed for that purpose nor its usage on such data was demonstrated. STAGATE, SpaGCN, and GraphST were developed for spatial single-omics data. We simply concatenated the two omics modalities and fed them to these three methods. We compared their performance on both simulation and experimental datasets (Supplementary Figure S6, revised manuscript). On the simulation data, SpatialGlue and GraphST outperformed SpaGCN and STAGATE in recapitulating the ground truth factors, with GraphST showing better quantitative metrics than SpatialGlue. On the experimental data, SpatialGlue and GraphST produced comparably good results with SpatialGlue showing better Jaccard Similarity and Moran's I score. In contrast, STAGATE's clusters were overly smoothed, while SpaGCN's clusters were highly fragmented with low Moran's I score.

Lastly, we also performed ablation studies to compare SpatialGlue with its variants. In SpatialGlue, we use dual attention mechanisms to integrate the different information layers. The first attention integrates spatial information with omics measurement, and the second integrates across omics. We performed the ablation studies to compare SpatialGlue with its variants wherein either one or both of the two attentions were replaced by concatenation (SpatialGlue-AC, SpatialGlue-CA, SpatialGlue-CC). We evaluated their performance with the simulation data and quantitative metrics to confirm that SpatialGlue performed better than its variants, demonstrating the advantages of attention integration over concatenation (Supplementary Figure S5a-d).

Comment 1.6: *Different categories of spatial technologies referenced in the manuscript (sequencing- and imaging-based) give justice to the diversity of spatial datasets. However, as all the applications focus on a subset of sequencing-based techniques, it is unclear if the current version of the model is also applicable to the imaging-based spatial datasets. Are there any additional challenges there?*

Response 1.6: We thank the reviewer for raising this critical issue. We agree that all the examples presented in the manuscript focus on a subset of sequencing-based techniques. With the current version, our SpatialGlue is also applicable to imaging-based spatial datasets as long as the data are processed into spot/cell by feature matrices. However, to the best of our knowledge, no existing imaging-based spatial technology can simultaneously measure more than one omics from the same tissue section. Therefore, we could not demonstrate SpatialGlue's usage on such data in the manuscript. Some imaging-based technologies, such as MERSCOPE by Vizgen, can co-profile the expression of mRNA and protein. However, its perplexity of the protein modality is low measuring only a few proteins. Another technology, CosMx by Nanostring, can measure the expression of thousands of genes and 64 or 128 proteins, but on different tissue sections. The next version of CosMx that enables co-profiling of high dimensional multi-omics on the same tissue slide will be released soon. Once the technology and data become available, SpatialGlue is ready to be applied to such data.

Comment 1.7: *Would the utilised approach to combine spatial information with feature matrices be useful for unimodal datasets as well? If so, and if the method is "scalable", readers might expect this to be demonstrated by being applied to a larger spatial dataset. This might also aid with having a richer choice of the baseline method(s) to compare SpatialGlue with.*

Related, SpaGCN and GraphST are only referenced but no comparison to them or related discussion has been presented. Considering the fact that multimodal methods typically stand on the shoulders of

unimodal methods, it might be useful to mention what approaches SpatialGlue borrows from GraphST, SpaGCN, STAGATE, or other methods and what new things SpatialGlue brings to the table.

Response 1.7: We thank the reviewer for the great questions. In SpatialGlue, we first use within-modality attention to combine spatial information with feature matrices and then between-modality attention to integrate across modalities. Indeed, the within-modality attention mechanism can be applied to integrate spatial information with feature matrices for unimodal data. However, SpatialGlue's current architecture is specifically designed for spatial multi-omics and cannot be directly applied to unimodal dataset. Besides, testing on unimodal data is also out of scope of our manuscript.

We also acknowledge that we did not compare our SpatialGlue with unimodal methods, such as SpaGCN, STAGATE, and GraphST in the original manuscript. We also agree that including unimodal methods would help extend the choice of baseline methods to compare SpatialGlue with. Motivated by the reviewer's comment, we transformed spatial multi-omics into unimodal data by concatenating the different omics modalities, and then applied unimodal methods. In this way, we compared SpatialGlue with these three methods on simulation and mouse brain P22 data. On the simulation data, SpatialGlue achieved better performance than SpaGCN and STAGATE (Figure R1.5a-c). GraphST possesses comparable performance to SpatialGlue. The quantitative evaluation in Figure R1.5 (c) validates the visual observation. To further assess the performance, we compared SpatialGlue and unimodal methods on the mouse brain P22 dataset. The results in Figure R1.5 (d) shows that SpatialGlue has much better performance than unimodal methods as it identifies anatomical regions with clear boundaries such as cortex layers and genu of corpus callosum, which is also validated by quantitative evaluation using Moran's I score and Jaccard Similarity (Figure R1.5 (e) and (f)). We need to highlight that while STAGATE has higher Moran's I score than SpatialGlue, but obviously, the clusters identified by STAGATE is spatially over-smoothed.

Figure R1.5. Comparison between SpatialGlue and single-modal methods, i.e., SpaGCN, STAGATE, and GraphST. (a) Ground truth for simulation data. (b) Comparison between SpatialGlue and these

three methods in the simulation data. (c) Quantitative evaluation of these methods in the simulation data. (d) Comparison between SpatialGlue and these three methods in the mouse brain P22 dataset. (e) Comparison of Moran's I score of these methods evaluated on the mouse brain dataset. (f) Comparison of Jaccard Similarity scores of these methods evaluated on the mouse brain P22 dataset.

Besides unimodal methods, to fully demonstrate the superiority of SpatialGlue in integrating spatial multi-omics data, we have compared SpatialGlue with 7 other state-of-the-art baseline methods including Seurat, totalVI, MultiVI, MOFA+, scMM, MEFISTO, and StabMap in the revised manuscript (Figure 1, 2, 3). We carried out the comparison on 5 simulated and 12 real datasets.

We also acknowledge the fact that multimodal methods typically stand on the shoulders of unimodal methods. There are some common characteristics between SpatialGlue and GraphST, SpaGCN, and STAGATE. First, SpatialGlue, GraphST, SpaGCN, and STAGATE are all graph neural networks (GNNs)-based models that learn spatially resolved latent representations. Second, all of them incorporate expression data and spatial information for spatial omics data analysis. However, there are major differences between SpatialGlue and GraphST, SpaGCN, and STAGATE. Firstly, the applicable data types are different between SpatialGlue and these three baseline methods. SpatialGlue is tailored for spatial multi-modal data while all of GraphST, SpaGCN, and STAGATE are tailored for unimodal data. Although GraphST, SpaGCN, and STAGATE can be also applied to multi-modal data by taking the concatenation of different modality data as input, the results are often undesirable due to simplistic cross-omics concatenation (Figure S6 in the revised manuscript). Secondly, for each modality, SpatialGlue constructs two graphs, one for feature similarity and one for spatial proximity, and combine them using an attention mechanism to encode a spatially informed representation. This is considerably different from the approaches used by other available unimodal methods. We have discussed this in the revised manuscript.

Comment 1.8: *It might be beneficial for the manuscript to reiterate the specifics of 10x Visium-derived and other technologies used for the described datasets (e.g. spatial resolution, imaged area size, where applicable – spot size, distance between spots, etc.).*

Response 1.8: We agree that such information will be useful to the reader. We have reviewed the literature for the relevant technical specifications and added them to the manuscript's supplementary. Here we also describe the technical specifications. The 10x Visium and associated technologies (SPOTS) employ slides with dimensions of 6.5 x 6.5 mm, capture spots that are 55 μm in diameter, and the distance between spots being 100 μm (center to center). Stereo-seq employs slides with a capture area 200 mm^2 , bin size of 0.22 μm , and the distance between spots being 0.5 μm (Zhang et al. 2023). The Spatial-epigenome-transcriptome (Zhang et al., 2023) technology has a pixel size of 20 μm , with the pixels filling a 50 x 50 or 100 x 100 grid.

Table R2. Summary of technical specifications of different technologies.

Platform	Spatial resolution (μm)	Distance between spots (μm)	Image area size
10x Visium & SPOTS	55	100	6.5 x 6.5 mm
Stereo-seq	0.22	0.5	200 mm^2
Spatial-epigenome-transcriptome	20	-	50x50 or 100x100 grid

Reference

- [1]. Liao, Sha, et al. "Integrated Spatial Transcriptomic and Proteomic Analysis of Fresh Frozen Tissue Based on Stereo-seq." bioRxiv (2023): 2023-04.

[2]. Zhang, D. et al. "Spatial epigenome-transcriptome co-profiling of mammalian tissues". *Nature* 616, 113–122 (2023).

Comment 1.9: *It is mentioned that the outputs of the model can be used "in various downstream analysis, including clustering, visualisation, and DGE detection". Readers might appreciate a slightly more expanded description of how these tasks can be performed and/or references to the relevant methods.*

Response 1.9: We thank the reviewer for the great suggestions. We have added descriptions on the relevant downstream analyses that users can perform with the latent representations generated by SpatialGlue or other relevant methods. For instance, the latent representations can be used as input to various clustering algorithms such as Leiden, Louvain, or mclust to cluster the spots/cells into spatial domains. Subsequently, differential expressed gene (DEG) analysis can be performed on the identified clusters using statistical approaches to identify genes or proteins that are significantly more or less abundant in one cluster compared to the others. Visualization methods such as tSNE and UMAP have been widely used to visualize single-cell omics data. Similarly, tSNE and UMAP plots can be generated using the latent representations to visualize spatial omics data.

Comment 1.10: *In the main text:*

Comment 1.10.1: *In paragraph 1, "... feature counts that can vary enormously" — might benefit from changing to more quantitative and precise terms as it is not clear what exactly this refers to.*

Response 1.10.1: We apologize for the confusion caused. Here we are referring to the number of features measured in the different omics and how they differ across the omics. For example, the protein data modality typically measures 21, 31 and 51 proteins, the RNA-seq modality measures 18,085 to 32,285 genes, and ATAC-seq modality measures 121,068 peaks. Clearly, the number of features measured are very different. We have added feature counts for all datasets to Table S1.

Comment 1.10.2: *In paragraph 1, "... there is no tool designed specifically for spatial multi-omics" — as mentioned above, there are tools and approaches that can handle spatial multi-omics datasets though they might have a different scope compared to this model.*

Response 1.10.2: We agree that there are other multi-omics methods such as Seurat WNN, totalVI, MultiVI, MOFA+, scMM, StabMap, and MEFISTO that can be applied to spatial multi-omics data. We have also expanded our benchmarking to include such methods in the comparison with SpatialGlue. However, we would like to emphasize the difference between most of these methods and SpatialGlue. These competing methods either do not include spatial information as input or do not require the measurements from each modality to be from the same cell/location. On the other hand, SpatialGlue is designed specifically for data with spatial information, and with the different omics acquired from the same cells within a single tissue sample to achieve better performance in such cases. As of right now, the only method that is similar in scope is MEFISTO, which is designed for multi-omics data with either spatial or temporal dependency.

Comment 1.10.3: *In paragraph 1, "existing tools <...> target spatial single omics integrated analysis, while GLUE and Seurat WNN perform multi-omics data integration..." refers to a few methods however there are more spatial (e.g. benchmarked in <https://www.nature.com/articles/s41592-022-01480-9>) and multi-omics (e.g. reviewed in <https://www.nature.com/articles/s12276-020-0420-2>, benchmarked for cancer studies in <https://www.nature.com/articles/s41467-020-20430-7>) integration methods. Moreover, it might be worth noting that GLUE is originally tailored to unpaired multi-omics data.*

Response 1.10.3: Our method SpatialGlue aims to integrate spatial multi-omics data acquired from the same tissue section. In other words, the different omics captured correspond to the same cells or locations. SpatialGlue is also tailored for spatial omics data, requiring in the input of the spatial location of each measurement. The spatial methods mentioned in the benchmarking study by Li et al. (<https://www.nature.com/articles/s41592-022-01480-9>) aim to integrate single-cell RNA-seq data and spatial transcriptomics for deconvolution of the spatial data, which is a different task from what SpatialGlue aims to accomplish. In this case, the datasets are from the same omics (not different omics) and are acquired from different tissue samples/cells. In the review by Lee et al. (<https://www.nature.com/articles/s12276-020-0420-2>), the methods mentioned like MOFA and LIGER

are relevant in terms of multi-omics data analysis, but they are not specifically designed for spatial data or multi-omics from single tissue sample. The same goes for the methods tested in the benchmarking study by Cantini et al. (Benchmarking joint multi-omics dimensionality reduction approaches for the study of cancer).

We agree that GLUE is originally tailored to unpaired multi-omics data. We have replaced GLUE with methods that were designed for paired multi-omics data such as totalVI, MultiVI, MOFA+, scMM, StabMap, and MEFISTO.

Comment 1.10.4: *In paragraph 2, the term "spot" is mentioned without any accompanying context ("Similarly, the different data modalities can have distinct and complementary contributions to each spot"). This, together with Figure 1a labels, also already suggests spot-based sequencing methods to be central for this work; however both sequencing- and imaging-based assays are referenced in the previous paragraph.*

Response 1.10.4: We thank the reviewer for pointing out a potential source of confusion for our usage of terms. There are a number of terms used within the literature for describing the spatial points of capture and/or measurement. The commonly employed 10x Visium technology uses the term "spot", as do many similar technologies. Others may use the terms "pixel", "bin", or "voxel", especially for imaging-based technologies. As SpatialGlue takes in omics expression matrices and their accompanying spatial locations, it accepts preprocessed data from both sequencing and imaging-based methods. For brevity, we used the term "spot" to denote any capture or measurement with a spatial location. We have added an explanation in the manuscript to help clarify this.

Comment 1.10.5: *In paragraph 2, in "... than the common approaches of summation or concatenation", it is unclear what approaches are referred to here. As this model is referred to as the first tool for spatial multi-omics in the previous paragraph, it might be worth expanding (via text and/or citations) what common approaches are implied here.*

Response 1.10.5: We thank the reviewer for raising this critical issue. We agree with the reviewer that it is unclear what approaches are referred to here. We are referring to summing or concatenating the multiple matrices (cell/spot by feature) to generate an integrated matrix that can be analyzed by a unimodal algorithm. We have rephrased this in the manuscript to help clarify this.

Comment 1.10.6: *Panel 1b doesn't seem to provide much information in its current state. Moreover, it is unclear why H&E staining appears black-and-white.*

Response 1.10.6: We thank the reviewer for pointing this out. It is indeed not H&E staining. We have renamed it as "histological image" that shows the structure of cells and tissues. From this image, we can observe tiny circular structures known as germinal centres in the white pulp that serve as the sites of lymphocyte production. In the germinal centre, we expect to see T cells that are surrounded by B cells, white pulp, and red pulp macrophages in an "inside-out" order, which helped us visually confirm SpatialGlue's clusters. Ideally, we should use the image to produce ground truth labelling to quantitatively assess the clustering accuracy of different methods. However, due to the low resolution of the image, our pathologist collaborators were unable to manually produce a reliable reference annotation. Here, we retained and enlarged this image and displayed it in parallel to the clustering results, simply to provide a qualitative comparison of the tissue morphology with the computational clustering.

Comment 1.10.7: *In paragraph 3, "... clusters clearly did not align..." provides an observation that is not further corroborated by any quantifications.*

Response 1.10.7: We thank the reviewer for raising this critical issue. We acknowledge the absence of quantitative evaluation related to the description of "The clusters clearly did not align...". In response to the reviewer's comments, we plotted a cross tabulation heatmap to quantitatively assess the extent of alignment between RNA and protein expression data (Figure R1.5 (b)). We observed that only protein cluster 5 and RNA cluster 5 aligned well to each other. Protein cluster 2 and 3 were each split into two major clusters in the RNA modality, while RNA cluster 1 and 4 were each split into 3 and 2 major clusters respectively in the protein modality.

Figure R 1.6. (a) Spatial plots of the RNA and protein expression data. (b) Cross tabulation heatmap between RNA and protein cluster labels.

Comment 1.10.8: *Related, Figure 1c displays 6 clusters in each modality. There are no details on the clustering procedure. Assuming clustering was performed on each modality individually, colour coding is different for different panels of Figure 1c, which might be confusing for the reader (i.e. cluster 1 spots for the RNA UMAP are not the same as cluster 1 spots for the Protein UMAP, etc.).*

Response 1.10.8: We thank the reviewer for raising this critical issue. Figure 1c (from the original manuscript) shows the UMAP and spatial plots of individual RNA and protein modalities, respectively. The clustering was performed on each modality individually and separated color coding should be used for RNA and protein modalities, respectively. In response to the reviewer's comments, we have changed Figure 1c panel (from the original manuscript) in the revised version of manuscript.

Comment 1.10.9: *In paragraph 3, "... plotted the clusters obtained from the integrated analysis onto the individual modalities' UMAPs..." the language could be improved: effectively, spots in the individual modalities' UMAP spaces were plotted and colours by their cluster identity.*

Response 1.10.9: We agree that this statement can be improved. At present, we have substantially revised the manuscript with a large volume of new quantitative results. As such, the presentation of results has been revised and this discussion is no longer present in the current version of the manuscript.

Comment 1.10.10: *Spot enrichment / separation statements at the end of the paragraph 3 as well as analogous statements in the following text are not supported by any quantifications. Performing and displaying those would greatly improve the understanding of the performance of the model and consequently its appeal.*

Response 1.10.10: We thank the reviewer for this suggestion on improving our manuscript. The data analyzed in this example was based on the 10x Genomics Visium platform, which has spot sizes of 55 μm . Thus, each spot captures the transcriptomes and proteins of a mixture of multiple cells. Therefore, based on the expression of markers, we can only conclude the higher likelihood of specific cell types being present in each spot. We cannot dismiss the possibility of other cell types being present.

Furthermore, to perform quantitative evaluation, ground truth labeling of cell types or anatomical structures are usually required. For this dataset, ground truth labels are not provided in the original study, and the resolution of the associated image is not high enough for a domain expert to generate manual labeling. As such, we have adopted the "unsupervised" metrics such as Jaccard Similarity to evaluate the preservation of distance in the joint space, and Moran's I score to evaluate the spatial autocorrelation of clusters produced with different algorithms as shown in the revised Figure 3g, h.

Comment 1.10.11: *In paragraph 5, neighbourhood enrichment analysis is mentioned however the details of the analysis seem to be missing from the methods section. The respective Figure 1i could be potentially improved by adjusting the labels for the colour scale as the two-digit precision doesn't seem to be meaningful.*

Response 1.10.11: We thank the reviewer for this suggestion on improving our manuscript. We have added the details of neighbourhood enrichment analysis in methods section of the revised manuscript. We have also adjusted the labels for the color scale in Figure S24d in the revised manuscript.

Comment 1.10.12: *Similarly, the details of the co-occurrence score calculation could be useful for the readers.*

Response 1.10.12: We thank the reviewer again for the suggestion. We have added the details of co-occurrence score calculation in the revised manuscript.

Comment 1.10.13: *For comparisons of the modality weights for SpatialGlue and Seurat WNN (e.g. Figure 1e vs Supplementary Figure S1d), it is unclear if there's a systematic difference in terms of how these two methods learn modality weights, which in turn influences how these weights should be interpreted and if they are directly comparable.*

Response 1.10.13: Here we would like to clarify the approaches used by SpatialGlue and Seurat WNN to compute modality weights. There are substantial differences in how SpatialGlue and Seurat compute their respective modality weights. Firstly, for Seurat, the procedure of modality weight calculation consists of three main steps: 1) Constructing independent k-nearest neighbor (KNN) graphs for both modalities. 2) Performing within and across-modality predictions. 3) Calculating the cell-specific modality weights. For SpatialGlue, there are two main steps: 1) Learning modality-specific latent representations for both modalities. 2) Learning cell-specific modality weights by taking into account the importance of each modality to each cell in an adaptive way. During model training, SpatialGlue adaptively captures the difference of the importance of each modality to each cell. This adaptability allows SpatialGlue to potentially provide a more context-sensitive assessment of each modality's contribution to cellular characteristics. Secondly, the inputs used for modality weight calculation differ between Seurat and SpatialGlue. Seurat employs raw expressions as input, which may be susceptible to noise, potentially affecting the accuracy of the computed modality weights. In contrast, SpatialGlue calculates its modality weights using the denoised latent representations as inputs. These denoised representations have been demonstrated to be more informative than the original expression profiles (as shown in Figure R1.2 above). While the procedure of modality weight calculation is different between SpatialGlue and Seurat, these two modality weights are directly comparable, as both modality weights measure the contributions of each modality to the integrated clusters.

Comment 1.10.14: *Figure 2c denotes clusters as numbers while tissue structures are named in the corresponding text (paragraph 6), which makes it hard to relate those two.*

Response 1.10.14: We thank the reviewer for highlighting the lack of clarity on our labeling. The clusters in Figure 2c and Figure 2e correspond in the original figures; the full names of the clusters were removed from Figure 2c due to space constraints. In view of potential confusions, we have updated all figures in the revised manuscript to include the full name of clusters as much as possible. The only exception is Figure 3d due to space constraints. Therefore, we emphasize that the numbers used to label clusters do correspond between sub-figures of the same dataset.

Comment 1.10.15: *In paragraph 6, "... the clusters showed greater fragmentation, especially in the cortex" provides an observation that is not supported with any quantifications.*

Response 1.10.15: We agree that our statement is qualitative, and quantitation will help support our assessments. In the revised manuscript with the present results, we have made our assessment more quantitative with the help of Moran's I score to assess the fragmentation of clusters (e.g., Figure 3b,g).

Comment 1.10.16: *In paragraph 7, "... grainy without clear boundaries between regions" provides an observation that is not supported with any quantifications.*

Response 1.10.16: We agree that our statement is qualitative, and quantitation will help support our assessments. Now we have used Moran's I score to quantitatively assess the performance of Seurat and SpatialGlue in the mouse brain P22 dataset. Figure R1.7 (c) shows that SpatialGlue achieves much higher score than Seurat, confirming our observation of Seurat clusters being grainy without clear boundaries between regions. We have also added the quantitative assessment for all datasets in the revised manuscript.

Figure R1.7 (a) Spatial clustering of Seurat in the mouse brain P22 dataset. (b) Spatial clustering of SpatialGlue in the mouse brain P22 dataset. (c) Quantitative comparison of Moran's I scores between Seurat and SpatialGlue.

Comment 1.10.17: *In paragraph 8, when speaking about computational efficiency, it might be beneficial to also demonstrate how SpatialGlue scales depending on the number of spots, features or principal components, modalities as well as to provide similar scalability properties for the baseline method(s).*

Response 1.10.17: We thank the reviewer for the comments. We have added a figure showing time complexity to compare the scalability of different methods in the revised manuscript (Figure R1.8 below and Figure S7g). Among the tested methods, SpatialGlue consumed the least amount of time on both the RNA + protein and RNA + ATAC datasets.

Figure R1.8 Time complexity of SpatialGlue and competing methods.

Comment 11: *In the methods section:*

Comment 1.11.1: *It is unclear why the choice of 50 principal components for the gene expression modality of the SPOTS dataset as the input of the encoder is consistent with the input dimension with the ADT data, which has 21 measurements.*

Response 1.11.1: We thank the reviewer for spotting the error. It is a typographical error, and we did choose the first 21 principal components for the gene expression modality of the SPOTS dataset as input to the encoder to ensure a consistent input dimension with the ADT data. We have corrected it in the revised version of our manuscript.

Comment 1.11.2: For the Stereo-CITE-seq dataset, “in fewer than 10 cells” and “expressed in fewer than 50 cells” should probably say “in fewer than 10 spots” and “expressed in fewer than 50 spots”, respectively.

Response 1.11.2: We thank the reviewer again for carefully reading our manuscript. We have carefully gone through the manuscript and corrected various errors. As for the Stereo-CITE-seq data, the expression matrix was divided into non-overlapping bins covering an area of 50 X 50 DNA nanoball (DNB), and the transcripts of the same gene were aggregated within each bin. As such we have changed “cells” to “bins”.

Comment 1.11.3: For the Spatial-ATAC-RNA-seq dataset, “2,2914” probably means “22,914” and “fewer than 200 cells” probably means “fewer than 200 spots”.

Response 1.11.3: We thank the reviewer again for carefully reading our manuscript. We have carefully gone through the manuscript and corrected the errors.

Comment 1.11.4: For model training, it might be worth expanding how exactly the weight factors “vary between different spatial multi-omics technologies”. Related, it is unclear how impactful the chosen weight factors are for the final model. Individual values are mentioned in the implementation details however it is unclear how the user should choose or adjust those.

Response 1.11.4: We agree that the choice of weights can have substantial impact on the analysis results, and the choice of weights is influenced by the characteristics of the respective input data. At present, SpatialGlue has assigned weights that we have tested on four different technologies, namely SPOTS, Stereo-CITE-seq, Spatial-epigenome-transcriptome, and 10x Visium. Users with data from these platforms can employ the respective sets of weights to analyze their data. For data acquired with other technologies, we provide a default parameter set that would work for most users on most data types. For advanced users, they are allowed to adjust the weight factors based on the loss curve to ensure each loss has comparable scale. We have also revised the manuscript (and the SpatialGlue Github repository) to guide users on this.

Comment 1.11.5: For the correspondence loss, do the terms $Y_{\{1\}}^{(l-1)} W_{\{d2\}}^{(l-1)}$ and $Y_{\{2\}}^{(l-1)} W_{\{d1\}}^{(l-1)}$ imply that all modalities should have the same number of features when provided as input to the encoder? If yes, does this limit the utility of the method in scenarios when modalities have vastly different numbers of features?

Response 1.11.5: We thank the reviewer for the questions. Indeed, the terms $Y_{1}^{(l-1)} W_{d2}^{(l-1)}$ and $Y_{2}^{(l-1)} W_{d1}^{(l-1)}$ imply that all modalities should have the same number of features when provided as inputs to the encoder. However, before inputting into our model, we reduce the dimensions of both input modalities (e.g., RNA and protein) to the same reduced dimensions via PCA. Therefore, our model is not limited to the difference of number of features between different modalities.

Comment 1.11.6: For the adversarial loss, it is unclear from the text how it should be extended to three or more modalities though the framework is described to be extensible to more than two modalities.

Response 1.11.6: We apologize for confusion regarding the extensibility of our model. We agree that we did not clearly exemplify on how SpatialGlue can be extended to three or more modalities. To improve the extensible ability of our SpatialGlue model, we have removed the adversarial loss in the framework while ensuring superior and stable performance and demonstrated its usage on 3 modality simulation data. Figure R1.9 displays that SpatialGlue achieves desirable performance compared to ground truth.

Figure R1.9 Integration of triplet omics data by SpatialGlue on simulation data.

Comment 1.11.7: For the analysis of the Spatial-ATAC-RNA dataset, it is stated that 50 principal components were used for SpatialGlue while only 10 were used for Seurat WNN. Together with other differences in data processing for the two methods that are then directly compared, this raises the question of ensuring that the presented discrepancies between the two methods are not influenced or even fully explained by the data processing choices.

Response 1.11.7: We agree that this may be a potential source of performance differential when comparing two methods. We have run the Seurat workflow with different number of PCs and found that Seurat performs better with 10 PCs than 50 PCs (Figure R1.10 below). The optimal number of PCs for different methods could be different. Thus, we retained Seurat’s results with 10 PCs and have also added the results with 50 PCs in Supplementary Figure S14f.

Figure R1.10 Comparison of Seurat’s results with 10- and 50-dimension PCA features as input.

Comment 1.11.8: For the code availability, it currently reads “GraphST”, which should probably be replaced by “SpatialGlue”.

Response 1.11.8: Thank you for pointing out the mistake. We have corrected it.

Conclusion

With the growing scale and complexity of experimental designs, spatial omics datasets pose a lot of analytical challenges with data integration in its centre. This reiterates the relevance of the discussed work, and I expect it to be of interest to those readers who are interested in spatial omics as well as a broader audience thanks to the attempts to derive biological insights from different biological contexts (murine spleen, thymus, brain profiled with different technologies). Presenting the model as part of the complex landscape of the currently available spatial integration methods by providing rigorous model evaluations as well as quantifying clustering performance and differences to other baseline methods

could improve the understanding of the model's merits and help other researchers to choose SpatialGlue for spatial omics integration when it's appropriate.

Finally, thank you for the opportunity to review this work!

Reviewer #2 (Remarks to the Author):

In this manuscript Long et al. present SpatialGlue, a novel method for spatial multimodal data integration using graph neural networks and attention mechanisms. The manuscript is well written and with the current availability of spatial multi-omics technologies is highly relevant to the field. The proposed approach seems promising; however I do have some concerns/remarks. My main concern is due to the lack of real quantification of the performance of the approach and on how this will affect end-users, as most of the analyses presented in the manuscript seem to be more qualitative based on the anecdotal examples presented by the authors:

Response: We thank the reviewer for the positive comments. We have carefully addressed all the comments and suggestions when preparing this revision of the manuscript. We acknowledge the concerns regarding the lack of quantitative evaluations, and we have introduced several measurement metrics to quantitatively evaluate the results and added them into the manuscript. These are also presented in the following paragraphs. Please let us know if you have additional comments.

Comment 2.1: *The authors state that: "We believe this approach enables more accurate integration than the common approaches of summation or concatenation." This statement should be properly backed up with quantitative evidence. Not only should the authors compare to simple integration strategies, such as concatenation, the manuscript should consider and compare to state-of-the-art methods such as, MOFA+ by Arguelaguet et al. (2020, Genome Biology), MEFISTO by Velten et al. (2022, Nature Methods), multiVI by Ashuach et al. (2023, Nature Methods), scMM by Minoura et al. (2021, Cell Reports). If the integration of the spatial component in SpatialGlue, which many of these methods lack, delivers a more meaningful integration, its contribution should be clearly and quantitatively demonstrated. For example, what are the attention weights for the spatial neighbourhood graph vs. the feature graph? The reader should be convinced of the added value of 1) the integration of the different modalities, and 2) the incorporation of spatial information to each multi-modal data point.*

Response 2.1: We thank the reviewer for the constructive suggestions. We acknowledge the manuscript can be improved as suggested. We have conducted ablation studies to demonstrate the contribution of 1) the integration of the different modalities and 2) the incorporation of spatial information in our model by comparing SpatialGlue with its variants. Secondly, we expanded our coverage of methods compared to MOFA+, MEFISTO, totalVI, MultiVI, scMM, and StabMap, which are state-of-the-art integration methods for single-cell multi-omics data. Lastly, we also added supervised and unsupervised metrics to perform quantitative evaluation with other state-of-the-art methods. The supervised metrics are AMI, ARI, NMI, homogeneity, mutual information, and V measure, while the unsupervised metrics are Jaccard Similarity, and Moran's I score.

We agree with the reviewer that the statement "We believe this approach enables more accurate integration than the common approaches of summation or concatenation." should be properly backed up with quantitative evidence. In SpatialGlue, we use dual attention mechanisms. The first attention integrates spatial information with omics measurement, and the second integrates across omics. Here we have performed ablation studies to compare SpatialGlue with its variants wherein either one or both two attentions were replaced by concatenation (SpatialGlue-AC, SpatialGlue-CA, SpatialGlue-CC). We evaluated their performance with the simulation data and quantitative metrics (Figure R2.1). Figure R2.1 (c) shows that all of SpatialGlue and its variants could accurately identify four factors compared to ground truth. However, discontinuous spatial factors with noises could be found in the variants 'SpatialGlue-CA' and 'SpatialGlue-CC'. 'SpatialGlue-AC' achieves comparable performance to SpatialGlue. The quantitative results in Figure R2.1 (d) indicates that SpatialGlue consistently outperforms all variants in terms of six metrics. The results demonstrate that intra- and inter-modality attention mechanisms do enable more accurate integration than the common approach of concatenation.

Figure R2.1 Ablation study on simulation data. (a) Ground truth. (b) Spatial plots of the modalities 1 and 2. (c) Spatial plots of SpatialGlue and its variants, SpatialGlue-CA, SpatialGlue-AC, and SpatialGlue-CC. (d) Quantitative comparison between SpatialGlue and its variants in terms of six measurement metrics.

In addition, we also simply concatenated the two omics modalities and fed them to methods developed for spatial single-modal data such as STAGATE and GraphST. We evaluated their performance with simulation and real data. To evaluate the performance, we used supervised metrics for simulation data and unsupervised metrics, i.e., Jaccard Similarity and Moran's I score, for real data. Figure R2.2 (b) shows that in the simulation data, both SpatialGlue and GraphST have much better performance than STAGATE compared to ground truth (Figure R2.2 (a)) while GraphST exhibits performance akin to SpatialGlue, which is also validated by the quantitative evaluation in Figure R2.2 (c). To further assess the performance of SpatialGlue, we compared SpatialGlue with STAGATE and GraphST on a mouse brain dataset. As illustrated in Figure R2.2 (d), both GraphST and SpatialGlue exhibit superior performance over STAGATE. Although GraphST succeeds in identifying certain cortex layers, it fails to identify precise boundaries. In contrast, SpatialGlue excels in accurately identifying cortex layers with distinct boundaries. The quantitative results of Moran's I scores (Figure R2.2 (e)) and Jaccard Similarity (Figure R2.2 (f)) validate the superiority of our SpatialGlue model once again. We need to highlight that while STAGATE achieves a higher Moran's I score than SpatialGlue, obviously, the clusters identified by STAGATE are over-smoothed.

Figure R2.2. (a) Ground truth for simulation data. (b) Comparison of spatial plots between SpatialGlue and single-modal methods STAGATE and GraphST in the simulation data. (c) Quantitative comparison between SpatialGlue and these two single-modal methods. (d) Comparison between SpatialGlue and STAGATE and GraphST in the real mouse brain data. (e) Comparison of Moran's I score of these three methods in the mouse brain data. (f) Comparison of Jaccard Similarity scores of these methods in the mouse brain data.

To validate the contribution of spatial component, we conducted another ablation study by comparing SpatialGlue with its variant, i.e., SpatialGlue without spatial information (SpatialGlue w/o spatial). On the simulated data, while 'SpatialGlue w/o spatial' identifies factors 2, 3, and 4 (Figure R2.3 (c)), some noise can be observed when compared to ground truth (Figure R2.3 (a)). In contrast, SpatialGlue successfully identifies 4 factors with few noises. That highlights the importance of spatial information. Figure R2.3 (d) shows that SpatialGlue achieves much higher scores than the variant in terms of 6 metrics. Therefore, we can conclude that spatial information plays an important role in improving the model performance.

Figure R2.3. Ablation study on simulation data. (a) Ground truth. (b) Spatial plots of the modalities 1 and 2. (c) Spatial plots of SpatialGlue and its variant, i.e., SpatialGlue without spatial information. (d) Quantitative comparison between SpatialGlue and its variant in terms of six metrics.

In addition, we have also added the attention weights for the spatial neighbourhood graph vs. the feature graph to the supplementary figures (e.g., Figure S1a, S2e, Figure S3e, j, Figure S8b, k, and Figure S12c, d) in the revised manuscript. In most of the datasets, the spatial graph was assigned with heavier weights compared to the feature graph, suggesting spatial adjacency plays a more important role than feature similarity. That implies that most cells or spots demonstrate spatial correlation and their distribution in the tissue is spatially restricted. Nevertheless, despite the more important role played by the spatial graph, the feature graph constructed based on feature similarity is also informative. As such, we combine them with attention integration in our model.

Furthermore, we compared the performance of SpatialGlue with 7 state-of-the-art methods with the simulated data and real data in the revised manuscript. Among the tested methods, only SpatialGlue and MEFISTO make use of spatial information. As shown in the Figure R2.4 (b), our SpatialGlue demonstrated superior performance in matching the ground truth (Figure R2.4 (a)) compared to baseline methods (Seurat, MultiVI, totalVI, MOFA+, MEFISTO, scMM, and StabMap). Among the baseline methods, MEFISTO is the only method that can accurately identify two of four factors with minimal noise. The results of both spatial methods, namely SpatialGlue and MEFISTO, highlight the important role of spatial information in spatial multi-omics analysis. Among the non-spatial methods, Seurat achieved relatively better performance as it recovered four factors but with noises. While other baseline methods can uncover some patterns of part of factors, some noises could be obviously found. The quantitative results are reported in Figure R2.4 (c). In terms of 6 metrics, SpatialGlue consistently outperformed all other methods, which validated the superior performance of SpatialGlue. The results on the remaining 4 simulation datasets were presented in the Supplementary Figures S1, S2, and S3.

Figure R2.4 (d) shows the box plot of six measurement metrics of the eight methods on the 5 simulated data, indicating SpatialGlue has much higher median scores. This demonstrated the effectiveness and robustness of SpatialGlue in integrating spatial multi-omics data again.

Figure R2.4. Benchmarking on simulation data. (a) Ground truth. (b) Spatial plots of the simulated modalities 1 and 2, and comparison of spatial plots of integrated methods. (c) Quantitative comparison between SpatialGlue and baseline methods in terms of six metrics on the simulated data. (d) Boxplots of quantitative evaluation of the integrated methods on the simulated data, each containing the results of five simulated datasets.

Comment 2.2: *Although it's nice that the authors illustrate their method with 3 different datasets using different technologies, the performance of the approach still seems very anecdotal as only 1 tissue slice seems to be analysed for each technology. Is this the best performing slice? Was the performance consistent across slices? Are the learned representations stable across slices of the same tissue?*

Response 2.2: We thank the reviewer for raising this critical issue. We acknowledge that one tissue slice for each technology is indeed too few. In the original manuscript, we tested SpatialGlue on three datasets from three different technological platforms, i.e., SPOTS, Stereo-CITE-seq, and Spatial-epigenome-transcriptome. The dataset for each technological platform was randomly selected. In response to the reviewer's comments, we collected more tissues slices from these platforms and further assessed the performance of SpatialGlue on them to demonstrate the stability and generalizability of SpatialGlue. Specifically, we have shown the results for four mouse brain slices (Spatial-epigenome-transcriptome) (Figure 2, Figures S12-S15), four mouse thymus slices (Stereo-CITE-seq) (Figure 3, Figures S17-S19), and two mouse spleen slices (SPOTS) (Figure 3, Figure S24-25) in the revised manuscript. In addition, we produced and added new datasets from two human lymph node slices (Figure 1 and Figure S8) with 10x Visium RNA-protein co-profiling platform. On these datasets, SpatialGlue showed consistent performance across slices, and the learned representations were stable across slices of the same tissue.

Comment 2.3: *Much of the analysis seems to rely on the accurate representation of known anatomy and on how this is captured by unsupervised clustering. When the authors state that "The clusters clearly did not align..." between modalities, how did the authors choose the appropriate clustering resolution to make that comparison? Choosing the resolution with the same number of clusters seems to be overly simplistic, as it's difficult to predict if at a lower/higher resolution a more optimal alignment might be achieved (even if more/less clusters are called). These sorts of statements should also be quantitatively backed by known measures of cluster overlap/separation (e.g. Jaccard index, cluster overlap, silhouette scores etc.). The same holds true for the comparisons made with Seurat WNN and other tools.*

Response 2.3: We thank the reviewer for the insightful comments and suggestions. It is true that much of the analysis does rely on the accurate representation of known anatomy and on how this is captured by unsupervised clustering. In our original manuscript, for each dataset used, we chose the number of clusters based on the expected number of cell types or tissue structures present. In the revised version, we generated different number of clusters to obtain a box plot of clustering accuracy (Figure 1 (f) and (k)).

In addition, we also calculated Jaccard index to quantitatively measure the cluster overlap between the RNA and ADT modalities of the mouse spleen data to support this statement “*The clusters clearly did not align...*” (Figure R2.5). The Jaccard index does vary for different clustering resolution. The highest score is achieved when the number of clusters is set to 4 for both RNA and protein modalities. Nevertheless, despite the well alignment with 4 clusters, when increasing the number of clusters, most of tested methods were able to identify more layers of macrophage cell types. As such, we set the number of clusters as 5 for both RNA and protein modalities in the revised manuscript.

Figure R2.5. Crosstab heat map of Jaccard index calculated by RNA and protein modalities with different number of clusters in the mouse spleen replicate 1 sample.

Furthermore, for datasets with ground truth labeling of cell types or anatomical structures, we used 6 “supervised” metrics including AMI, NMI, ARI, Homogeneity, Mutual information, and V measure to evaluate clustering accuracy at different clustering resolutions. For datasets without ground truth, we used Jaccard Similarity to evaluate the preservation of single-omics data variation after multi-omics integration. We applied such quantitative evaluation to SpatialGlue and all 7 competing methods including Seurat WNN. We have shown the results in the revised manuscript (Figures 1-3 and Figures S1-S25).

Comment 2.4: *The authors state several times the contribution of both modalities to the integrated space, however in the methods section it is stated that to account “for differences in feature distributions across the datasets”, weight factors were assigned. What’s the relationship between the modality weights and these hyperparameters?*

Response 2.4: We thank the reviewer for the questions. Weight factors are used in formula 13 of the manuscript (see also below), which is a reconstruction loss function where the first and second terms represent the feature reconstruction losses of modalities 1 and 2, respectively. Considering the differences of feature distribution between different modalities, we introduce two weight factors γ_1 and γ_2 to balance the contributions of these two modalities. Therefore, these factors will influence the representation learning. A higher value will result in bias towards minimizing the loss for the corresponding modality during representation learning. As a result, that modality will attain higher attention weight values, indicating more contributions to the integrated representation. The weight factors need to be chosen based on the characteristics of the respective input data. Data acquired with different technologies exhibit different characteristics and weight factors chosen for them should be different. For instance, weight factors for spatial RNA+protein and spatial RNA+ATAC should be different. For each of the four technologies (SPOTS, Stereo-CITE-seq, 10x Visium, and Spatial-epigenome-transcriptome), we provide a default parameter set that allows the model to perform well on

most datasets. In cases wherein users wish to further optimize the performance for their dataset, they can adjust these two parameters based on model parameter ('weight factors').

$$\mathcal{L}_{recon} = \gamma_1 \sum_{i=1}^N \|x_i^1 - \hat{h}_i^1\|_p^2 + \gamma_2 \sum_{i=1}^N \|x_i^2 - \hat{h}_i^2\|_p^2, \quad (13)$$

Comment 2.5: *The authors use an interesting deep learning conceptual framework; however they seem to limit the input dimensionality drastically by performing linear dimensionality reduction prior to applying their approach. Isn't the strength of deep learning approaches that they don't have to deal with prior feature engineering, and can tackle nonlinearities often found in highly complex biological datasets? What happens if the GNN is given the complete input vector instead of the PCA reduced one? What is the effect of different methods of dimensionality reduction methods?*

Response 2.5: We agree that this is an important question within machine learning frameworks. The original framework of SpatialGlue does reduce the dimensionalities of both input modalities (i.e., RNA and protein) using PCA. This procedure aims to ensure the dimensionalities of the representations of both modalities are consistent when integrating them in the attention module. We also agree that deep learning approaches do not need prior feature engineering and can tackle nonlinearities found in highly complex biological datasets. In response to the comment, we conducted experiments to test the performance of our SpatialGlue with simulated data by taking the complete input expression data (SpatialGlue-full) as input instead of the PCA-derived latent representation.

Figure R2.6. Ablation study on simulation data. (a) Ground truth. (b) Comparison of spatial plots of SpatialGlue and its variant, i.e., SpatialGlue with full expression as input. (c) Quantitative comparison between SpatialGlue and its variant.

Figure R2.6 (b) shows that both SpatialGlue and its variant 'SpatialGlue-full' are able to accurately identify the four factors compared to ground truth and their performance is visually comparable, which is also validated by the quantitative evaluation (Figure R2.6 (c)). However, with raw data as input, we need to incorporate an additional MLP for dimension reduction. The MLP's first layer has the same number of nodes as the initial dimension of the input data, which is usually in the scale of thousands. This results in a much bigger model with more parameters to train, which consequently increases the time and memory usage. Given that SpatialGlue using PCA or raw values as input produced comparable accuracy, for simplicity, we choose to use PCA instead of raw values.

Minor points:

Comment 2.6: *The abstract is rather vague, what is meant by "SpatialGlue can accurately aggregate cell types into spatial domains at a higher resolution across different tissue types"?*

Response 2.6: We apologize for any confusion regarding the above statement. For spatial multi-omics data, individual modalities often suffer from insufficient resolution. For example, some cell types may not be identifiable or separable in certain modalities due to low signal capture efficiency. SpatialGlue aims to enhance the resolution of such cell types by integrating multi-modal data to increase the strength of signals that separate cell types or states. Here, spatial domains refer to spatial distribution of cell types in the tissue. We have revised the text for greater clarity to aid the reader.

Comment 2.7: "These constructed graphs can possess unique semantic information that should be integrated." What kind of unique semantic information is referred to here?

Response 2.7: We apologize for any confusion regarding the definition of unique semantic information. In SpatialGlue, we construct two neighbor graphs, spatial graph and feature graph, to capture different information. Based on the assumption that spots that are spatially adjacent in a tissue usually are similar cell types or cell states, we construct a spatial graph based on the Euclidean distance computed from the spatial locations. Therefore, the semantic information in the spatial graph refers to the physical proximity between spots, which supervises the model to learn similar representations of spatially adjacent spots. The feature graph aims to model the phenotypic proximity of spots which have the same cell types but are spatially non-adjacent to each other, or even far away. To construct the feature graph, we apply the k -nearest neighbor algorithm (KNN) on the PCA embeddings. For a given spot, we choose the top k nearest spots as its neighbors. Therefore, the semantic information in the feature graph refers to expression similarity of spots. To improve clarity, we have added the definition for the semantic information into the manuscript in the methods section.

Reviewer #3 (Remarks to the Author):

The paper proposes a way to perform multi-modal integration in the context of spatial data. Graph neural networks are employed to represent data across multiple modalities, in feature and space dependent manners, respectively. Attention mechanisms are used to first weigh the contribution of feature specific and spatial specific representations within modalities, and then to weigh the contribution of modalities themselves.

Related approaches have been proposed in a number of different applications, see for example Multi-Modal Graph Neural Network for Joint Reasoning on Vision and Scene Text, Gao et al. and Multimodal learning with graphs, Ektefaie et al. Most similar is perhaps Graph Neural Networks for Multimodal Single-Cell Data Integration, Wen et al. The latter is particularly interesting as it does not employ an attention mechanism to solve related tasks.

While the approach is elegant, my feeling is that the paper relies too much on state-of-the-art concepts in machine learning (attention, GNN, adversarial training) without properly evaluating the need for each of the building blocks used. The study lacks benchmarking, there is no simulation scenario to help build intuition on when the different components are needed, and a quantitative evaluation section with metrics beyond visualization is lacking. Below I provide some ideas on how to improve along these dimensions, however I am aware that this would require significant rewriting.

Response: We thank the reviewer for the very constructive comments to help us improve the manuscript. We have carefully addressed all the comments and suggestions when preparing this revision of the manuscript. Please let us know if you have additional comments.

We agree that the three papers that the reviewer mentioned do have some relation to the work that we are pursuing with SpatialGlue. However, these methods are not directly applied for spatial multi-omics data due to the following reasons. Firstly, the modalities referred to in the paper 'Multi-Modal Graph Neural Network for Joint Reasoning on Vision and Scene Text' are image and scene text that are different from the spatial multi-omics data used in SpatialGlue. Secondly, the paper 'Multimodal learning with graphs' introduced a blueprint for multimodal graph learning. It aimed to provide guidelines to help users design new models. The list of multimodal learning approaches provided are mainly developed for applications in computer vision (CV) and natural language processing (NLP), of which few of them can be directly applied to spatial multi-omics data used in our study. Finally, the paper 'Graph Neural Networks for Multimodal Single-Cell Data Integration' by Wen et al. was developed for multimodal single cell data integration and can be also applied to the integration of RNA&ADT or RNA&ATAC like SpatialGlue. While it does bear some similarity to SpatialGlue in terms of applications, there is a significant difference with it being a supervised model while SpatialGlue is an unsupervised model. Wen et al. adopted a 'pre-trained' setting and used cell type labels to supervise the training. However, cell type labels are not available in most spatial multi-omics data including those used in our manuscript. Moreover, on average, a supervised model in the scenario of correctly labeled data possesses a greater advantage than unsupervised model in representation learning. Therefore, it is unfair and not feasible to compare Wen et al.'s model with SpatialGlue.

We acknowledge that the manuscript can be substantially improved with additional benchmarking studies, addition of simulated data scenarios, and quantitative evaluation. Therefore, we compared SpatialGlue with 7 state-of-the-art methods on 12 real-world datasets and 5 simulation data in the revised version of manuscript. We also introduced several measurement metrics for quantitative evaluation.

To justify our design choices for the structure of SpatialGlue, we performed various ablation studies to evaluate the need for each of the building blocks such as attention, GNN, and loss function. It should be noted that to make our SpatialGlue a general model, we have removed the adversarial learning block in the revised model. Therefore, we focus on validating the contributions of attention and GNN. In the framework of SpatialGlue, dual attention mechanisms, i.e., intra- and inter-modality attentions, are designed to integrate information of different views. Intra-modality attention is employed to integrate the representations obtained based on spatial graph and feature graph. Inter-modality attention is employed to integrate representations across omics. Here we have performed ablation studies to compare SpatialGlue with its variants wherein either one or both of the two attentions were replaced by concatenation (SpatialGlue-AC, SpatialGlue-CA, SpatialGlue-CC). We evaluated their performance with the simulation data and quantitative metrics (Figure R3.1).

Figure R3.1 Ablation study on simulation data. (a) Ground truth. (b) Spatial plots of modalities 1 and 2. (c) Spatial plots of SpatialGlue and its variants, SpatialGlue-CA, SpatialGlue-AC, and SpatialGlue-CC. (d) Quantitative comparison between SpatialGlue and its variants in terms of six measurement metrics.

Figure R3.1 (c) shows that all of SpatialGlue and its variants could accurately identify four factors compared to ground truth. However, discontinuous spatial factors with noises could be found in the variants 'SpatialGlue-CA' and 'SpatialGlue-CC'. 'SpatialGlue-AC' achieves comparable performance to SpatialGlue. The quantitative results in Figure R3.1 (d) indicates that SpatialGlue consistently outperforms all variants in terms of six metrics. The results demonstrate that intra- and inter-modality attention mechanisms do enable more accurate integration than the common approach of concatenation.

In the framework of SpatialGlue, GNN is the base architecture that contributes to learning the latent representation of spots by integrating distinct omics data and spatial information. Considering the challenges in quantitatively evaluating the contribution of GNN, instead, we conducted parameter analysis to assess the influences of different parameters in the GNN block, including the dimension of PCA-reduced features, the number of neighboring spots, and the number of GNN layers, on the representation learning ability of GNN. We would show the results in the following Response 3.4.

Our SpatialGlue model is jointly trained by both reconstruction loss and correspondence loss, each contributing uniquely to the model performance. We would demonstrate it in the following Response 3.5.

Major points.

Comment 3.1: Benchmarking. *The paper currently lacks comparison with other simpler and related methods (see above, other ML communities). Within the spatial transcriptomics field, a paper I particularly like is Stabilized mosaic single-cell data integration using unshared features, by Ghazanfar et al. This paper would be a fantastic point of comparison! Comparing spatialGlue with StabMap on the StabMap datasets would offer insight into the advantages of spatialGlue. I would also be interested in understanding if removing the attention mechanism and concatenating the feature centric and spatial representations as representations would lead to a much worse performance.*

Response 3.1: We thank the reviewer for the suggestions. We acknowledge that the manuscript could be improved with more comparisons with other relevant methods. As discussed earlier, there are methods that have been developed for multi-modal data integration such as "Multi-Modal Graph Neural Network for Joint Reasoning on Vision and Scene Text" by Gao et al.; "Multimodal learning with graphs" by Ektefaie et al.; and "Graph Neural Networks for Multimodal Single-Cell Data Integration" by Wen et al. However, they were not designed for spatial multi-omics and do not utilize the accompanying spatial information. Moreover, some of these methods were designed for non-sequencing data integration, such as image and text. Some are supervised models that need ground truth cell type labels for model training. We also agree that the paper 'Stabilized mosaic single-cell data integration using unshared features' by Ghazanfar et al. does an excellent job in demonstrating their method StabMap in mosaic single-cell data integration. We have revised our comparisons by including StabMap as one of the 7 competing methods for benchmarking. The 6 other methods include Seurat, MOFA+, totalVI, MultiVI, scMM, and MEFISTO. The former five were developed for single-cell multi-omics and can be applied to spatial multi-omics without using the spatial information. MEFISTO is a generic factor analysis method for multimodal data. It can capture spatial dependencies between data points, and thus can be applied to spatial multi-omics.

The datasets used in the StabMap paper do include spatial multi-omics data. Specifically, the authors collected imaging mass cytometry (IMC), CITE-seq, and 10x Genomics Xenium data from breast tumour samples, and performed multi-hop mosaic integration using the CITE-seq data as the "bridge". In this scenario, StabMap does integrate spatial multi-omics (IMC protein + Xenium mRNA). However, the integration requires a single-cell multi-omics (CITE-seq protein + mRNA) dataset to serve as a bridge, which is not required by SpatialGlue. Moreover, the IMC and Xenium data were acquired from different samples, which is also out of scope for SpatialGlue that was designed for multi-omics simultaneously obtained from the same tissue slide. Instead of the StabMap datasets, we compared SpatialGlue with StabMap and other 6 methods on five simulation dataset and 12 real-world datasets in the revised version of manuscript (Table R1 on the 2nd page of this response letter, Figures 1-3 and Supplementary Figures S1, S2, S3, S8, S14, S15, S17-S19, S25).

Attention mechanism is an important component of SpatialGlue. We have conducted ablation study to demonstrate the contribution of attention mechanism to the performance of SpatialGlue by comparing SpatialGlue with its variant, i.e., concatenating feature centric and spatial representations as the integrated representations instead of the attention integration mechanism (SpatialGlue-CA), in the revised version of manuscript (Supplementary Figure S5c) and Figure R3.2 below. Figure R3.2 (b) shows the results with simulation data. Both the original SpatialGlue and its variant were able to recover all four ground truth factors compared to the ground truth. However, some noises can be obviously found in the variant's result. The quantitative assessment (Figure R3.2 (c)) also shows that SpatialGlue consistently outperformed its variant (SpatialGlue-CA) in terms of 6 metrics. Both the qualitative and quantitative results demonstrate that the attention mechanism enables more accurate integration than concatenation.

Figure R3.2. Abalation study on simulated data. (a) Ground truth. (b) Comparison of spatial plots of SpatialGlue and its variant, SpatialGlue with concatenation instead of intra-modality attention integration. (c) Quantitative comparison between SpatialGlue and its variant.

Comment 3.2: Simulations. A thought on generating spatial data from multiple modalities in order to have some notion of ground truth for evaluation is to use the generative model in *Nonnegative spatial factorization applied to spatial genomics* by Townes et al. To generate different modalities, one could sample different subsets of columns (intersecting or not) like in *StabMap*, and transform them (add noise, nonlinearities etc).

Response 3.2: We appreciate the reviewer for the valuable suggestion to incorporate simulation data into our study. Simulation data indeed allows us to assess performance with a ground truth for evaluation. In response to your suggestion, we adopted the approach outlined by Townes et al. to use nonnegative spatial factorization to simulate spatial multi-omics data. We employed the 'ggblocks' model from Townes et al. to generate a spatial gene expression matrix with zero-inflated negative binomial (ZINB) distribution, dimensions of 1,296 cells x 1,000 genes, and featuring four distinct factors. Similarly, we generated a spatial protein expression matrix with negative binomial (NB) distribution, dimensions of 1,296 cells x 100 proteins, and featuring 4 distinct factors. The simulation recapitulates the ZINB and NB distributions of spatial transcriptomics and proteomics respectively and matches the cells from the two modalities. To better mimic real-world scenarios, we added Gaussian distributed noise to both modalities. To increase statistical analysis power, we generated 5 simulation datasets with different parameters (Table R2).

Table R2. Summary of simulation parameters

Dataset	RNA						Protein				
	ZINB			Gaussian		dimension	NB		Gaussian		dimension
	pi	nzprob_nsp	blk_mean	mean	std		nzprob_nsp	blk_mean	mean	std	
Simulation1	0.6	0.2	0.2	2	0.5	1,000	0.25	0.4	2	0.5	100
Simulation2	0.5	0.2	0.3	2	0.5	1,000	0.25	0.5	2	0.5	100
Simulation3	0.6	0.2	0.4	2	0.5	1,000	0.25	0.6	2	0.5	100
Simulation4	0.5	0.2	0.5	2	0.5	1,000	0.25	0.7	2	0.5	100
Simulation5	0.5	0.2	0.6	2	0.5	1,000	0.25	0.8	2	0.5	100

We compared the performance of SpatialGlue with 7 state-of-the-art methods with the simulated data. Among the tested methods, only SpatialGlue and MEFISTO make use of spatial information. As shown in the Figure R3.3 (b), our SpatialGlue demonstrated superior performance in matching the ground truth (Figure R3.3 (a)) compared to baseline methods (Seurat, totalVI, MultiVI, MOFA+, MEFISTO, scMM, and StabMap). Among the baseline methods, MEFISTO is one of two methods that can accurately identify two of four factors with minimal noise. The results of both spatial methods, SpatialGlue and MEFISTO, highlight the important role of spatial information in spatial multi-omics analysis. Among the non-spatial methods, Seurat achieved relatively better performance as it recovered four factors but with noises. While other baseline methods can uncover some patterns of part of factors, some noises could be obviously found. The quantitative results are reported in Figure R3.3 (c). In terms of 6 metrics,

SpatialGlue consistently outperformed all other methods, which validated the superior performance of SpatialGlue. The results on the remaining 4 simulation datasets were presented in the Supplementary Figures S1-S3. Figure R3.3(d) shows the box plot of six measurement metrics of the eight methods on the 5 simulated data, indicating SpatialGlue has much higher median scores. This demonstrated the powerful ability of SpatialGlue in integrating spatial multi-omics data again.

Figure R3.3. Comparison between SpatialGlue and baseline methods on simulation data. (a) Ground truth. (b) Spatial plots of the modalities 1 and 2, and comparison of spatial plots of SpatialGlue and baseline methods, Seurat, totalVI, MultiVI, MOFA+, MEFISTO, scMM, and StabMap. (c) Quantitative comparison of SpatialGlue and baseline methods in terms of six metrics. (d) Boxplots of six metrics of the eight methods applied to the simulated data, each plotted by the values of 5 simulated datasets.

Comment 3.3: *Metrics. Quantitative metrics such as predicting cell types from representations, predicting spatial domains, preservation of distances in the joint space should be considered and reported for benchmarks and variants of the algorithm.*

Response 3.3: We thank the reviewer for the valuable suggestion. In response to the reviewer's suggestion, we have introduced multiple quantitative metrics for evaluation in the revised version of manuscript. For datasets that have ground truth labels of cell types or spatial domains, we used the "supervised" metrics including AMI, NMI, ARI, Homogeneity, Mutual information, and V measure to assess the accuracy of predicting cell types or spatial domains from representations. For datasets without ground truth, we used the "unsupervised" metric, Jaccard Similarity (Shila et al.) between the joint space and the original individual spaces such RNA or protein, to evaluate the preservation of distances in the joint space. To calculate the Jaccard Similarity, we first identify the k nearest neighbors of each cell within the joint space and individual spaces respectively and then calculate the overlap of the neighbor sets between the joint and individual spaces (formula given below). Using these metrics, we compared SpatialGlue with other 7 methods with simulation and real-world datasets (Figures 1-3 and Supplementary Figures S1, S2, S3, S8, S14, S15, S17-S19 and S25 in the revised manuscript). Our comparison showed that SpatialGlue better predicted cell type or spatial domains and showed better preservation of distances in the joint space than competing methods (Figure R3.4-3.7 below).

Jaccard Similarity: For cell i in embedding S we have m positions for the m omics levels (for example, RNA and protein). We extract the sets of size k (default 100) containing the nearest cells of the same omics layer, that is, $N_{im} = \{\text{set of neighbors of omics layer } m \text{ s.t. } \text{rank}(D(S_{im}, S_{jm}) \leq k \text{ where } D(a, b) \text{ is the Euclidean distance of vectors } a \text{ and } b. \text{ The Jaccard Similarity is thus}$

$$J_i = \text{Jaccard}(N_{i1}, N_{i2}) = \frac{|N_{i1} \cap N_{i2}|}{|N_{i1} \cup N_{i2}|}$$

Larger values of J_i correspond to larger overlap of neighbors between the two omics layers and are thus desired.

Figure R3.4. Comparison between SpatialGlue and baseline methods on simulation data. (a) Ground truth. (b) Spatial plots of the modalities 1 and 2, and comparison of spatial plots of SpatialGlue and baseline methods, Seurat, totalVI, MultiVI, MOFA+, MEFISTO, scMM, and StabMap. (c) Quantitative comparison of SpatialGlue and baseline methods in terms of six metrics. (d) Boxplots of six metrics of the eight methods applied to the simulated datasets, each plotted by the values of 5 simulated datasets.

Figure R3.5. Comparison between SpatialGlue and baseline methods on human lymph node A1 sample. (a) Manual annotation. (b) Comparison of spatial clustering of SpatialGlue and 7 state-of-the-art

methods. (c) Quantitative comparison of SpatialGlue and baseline methods in terms of six metrics. (d) Boxplots of six metrics of the eight methods applied to the human lymph node A1 sample, each plotted by the clustering results with number of clusters ranging from 4 to 11.

Figure R3.6. Comparison between SpatialGlue and baseline methods on mouse brain P22 sample. (a) Comparison of spatial clustering of SpatialGlue and 5 baseline methods. (b) Comparison of Jaccard Similarity scores of the 6 methods.

Figure R3.7. Comparison between SpatialGlue and baseline methods on mouse thymus sample. (a) Comparison of spatial clustering of SpatialGlue and 7 baseline methods. (b) Comparison of Jaccard Similarity scores of the 8 methods.

Reference

Ghazanfar, Shila, Carolina Guibentif, and John C. Marioni. "Stabilized mosaic single-cell data integration using unshared features." Nature Biotechnology (2023): 1-9.

Comment 3.4: Algorithm variants and robustness. Many parameters could have a strong impact on learning (given the appropriate metrics, see above). These include: the number (50) of PCA neighbors, the number of neighboring spots considered (6), number of GNN layers (l).

Response 3.4: We thank the reviewer for the comments. We agree that many parameters could have a strong impact on the representation learning. In response to the reviewer's comments, we conducted parameter analysis to validate the robustness of our SpatialGlue model against some main parameters, including the dimension of PCA-reduced features, the number of neighboring spots, and the number of GNN layers.

First, we measured our model performance with different dimensions of PCA-reduced features. We chose the values from {10, 15, 20, 25, 50}. Figure R3.8 (a) shows that SpatialGlue's performance is relatively stable when the PCA dimension is less than 50. Some noises could be obviously found when

the PCA dimension is equal to 50. The quantitative evaluation in Figure 3.8 (b) is consistent with the visual results. This indicates that our model is relatively robust against PCA dimensions when the value is less than 50. In our manuscript, we set the PCA dimension to 25 for the RNA and protein datasets by default.

Second, the number of neighboring spots controls how much feature information the model can learn. To verify the influence of the number of neighboring spots, we tested the performance of our SpatialGlue model with the value set to 3, 6, 12, 24, and 48, respectively. Figure 3.8 (c) and (d) shows that as the number of neighboring spots decreases, the performance degrades, and the best performance is achieved when the value is 3. By default, we set the number of neighboring spots to 3 for all datasets in our experiments.

Third, the number of GNN layers is a crucial hyper-parameter that can influence the model's ability to capture features in the graph. We evaluate the performance of our SpatialGlue model with the number of GNN layers ranging from 1 to 3. The results in Figure 3.8 (e) and (f) indicate that our SpatialGlue mode achieves the best performance when the number of GNN layers is set to 1. We use this value as default one in our model.

Figure R3.8. Parameter analysis on simulation data.

Comment 3.5: Optimization details. Not enough details are provided regarding the adversarial training and the corresponding loss. How are the hyperparameters optimized? Which latent representation is

used to compute the probabilities p_i ? What is the performance of the algorithm in the absence of an adversarial loss? This line of reasoning applies to the other losses in equation 18 as well. Addressing these questions would significantly improve the quality of the paper!

Response 3.5: We thank the reviewer for the constructive comments. We indeed do not provide enough details regarding the adversarial training and the corresponding loss in the original manuscript. In the previous version, the model is jointly trained by reconstruction loss, correspondence loss, and adversarial loss. The hyperparameters of $\{\gamma_1, \gamma_2, \gamma_3, \gamma_4\}$ are empirically optimized. Specifically, for datasets from different technological platforms, we use different hyper-parameter sets considering the differences between their feature distributions. For datasets from the same technological platforms, our model can generate stable output with the same hyper-parameter set. In the revised version of SpatialGlue, we provided a default hyper-parameter set that allows the model to perform well on most datasets. In cases wherein users wish to further optimize the performance for their dataset, they can adjust the parameters ('weight factors'). In the extended version of our SpatialGlue, we plan to introduce a self-adaptive mechanism to learn the hyper-parameters.

Indeed, the detail to calculate the probability p_i is not clear. In our previous version of model, we used modality-specific representation Y_1 and Y_2 of the modalities 1 and 2 to calculate their probabilities p_i and $(1 - p_i)$. It is achieved using a multi-layer neural network. As we have excluded adversarial training loss in the revised model to enhance generalizability, the detail is no longer present in the current version of the manuscript.

The adversarial loss within our model aims at constraining representations from different modalities within a uniform latent space. This constraint would ensure better alignment between a spot's representations across different modalities. Owing to its limited impact on the model's performance, as mentioned above, we have excluded adversarial training loss in the revised version to enhance generalizability while maintaining superior and stable model performance. Also, we have added more details regarding the optimization details in the revised manuscript.

In the revised model, the overall loss function consists of reconstruction loss and correspondence loss, each contributing uniquely to model training. The reconstruction loss ensures preservation of expression profiles in the integrated representation Z across modalities, while the correspondence loss enforces consistency between modality-specific representation Y and its corresponding representation \hat{Y} obtained through the decoder-encoder of another modality. In response to the reviewer's comment, we conducted ablation study to assess the impact of each loss function on the model's performance of using mouse brain P22 dataset. To do that, we obtain two variants of SpatialGlue, i.e., SpatialGlue without reconstruction loss ('SpatialGlue w/o recon') and SpatialGlue without correspondence loss ('SpatialGlue w/o corr'). Figure R 3.9 (a) shows that all of SpatialGlue and its variants achieve desirable performance as they identify some major regions, such as cortex layers. Yet more noises can be observed in 'SpatialGlue w/o recon' and 'SpatialGlue w/o corr' than SpatialGlue. Figure R3.9 (b) reports Moran's I score of SpatialGlue and its variants, which indicates SpatialGlue scores the highest. Therefore, we can make a conclusion that both reconstruction and correspondence losses contribute to the performance improvement.

Figure R3.9. Ablation study in mouse brain P22 spatial-ATAC-RNA-seq dataset. (a) Comparison of SpatialGlue with its variants, i.e., SpatialGlue without reconstruction loss ('SpatialGlue w/o recon') and

SpatialGlue without correspondence loss ('SpatialGlue w/o corr'). (b) Comparison of Moran's I score of SpatialGlue and its variants.

Minor points:

Comment 3.6: *While, the paper reads well generally, there are a few typos:*

- In equations 6, 7 α and β are used interchangeably $W_{\{W_i\}}$ is confusing, perhaps another letter could be used Eq 7, should be Y_i^m and not Y_i^t

- Be consistent with capitalization (I take Y is a matrix, and y is a vector). These are sometimes interchanged.

- In (11) and (12): would using A_s^{-1} instead of A_s have an impact? I am worried that multiplying by A_s too many times is equivalent to dividing by the $(\text{node_degree})^l$ of a node i , with l the number of layers, which can lead to values of zero very fast.

Response 3.6: We thank the reviewer for spotting these errors and raising the insightful question. We agree that some typos exist in the previous manuscript. We have carefully gone through the manuscript and correct existing errors. We also acknowledge the inconsistency in the use of uppercase and lowercase letters to denote matrices and vectors. To ensure clarity and uniformity, we have proofread the manuscript, consistently employing uppercase letter for matrices and lowercase letter for vectors.

In formulas (11) and (12), using \hat{A}_s^{-1} instead of \hat{A}_s may have an impact on the information propagation in the graph. It may change how nodes influence each other. In addition, it may introduce computational challenges as computing the inverse of a matrix can be computationally expensive.

We acknowledge the impact of multiplying by the adjacency matrix A (unnormalized) repeatedly, especially in relation to the propagation of information across layers. Indeed, in scenarios where nodes possess low degrees within sparse graphs, this progress can lead to rapidly diminishing values, potentially resulting in vanishing gradients. This issue is alleviated in our framework. On one hand, the matrix \hat{A}_s we used in formulas (11) and (12) is a normalized adjacency matrix that can help alleviate the vanishing gradient issue to a certain extent. On the other hand, the number of GNN layer is set as 1 in our model, which also help reduce the risk of gradient vanishing.

Comment 3.7: *What is the impact on overall performance of using two nonlinear activation functions in 15 and 16? How affected is the performance of the algorithm by choosing different activation functions?*

Response 3.7: We thank the reviewer for the insightful questions regarding the use of activation functions. We acknowledge the significance of activation functions in neural network architectures and their potential influence on overall performance.

In general, nonlinear activation functions are employed in deep learning models to capture nonlinear features. The use of two nonlinear activation functions in functions 15 and 16 would influence the overall loss and model accuracy. Specifically, the choice of activation functions can lead to the changes in the distribution range of corresponding representation \mathcal{Y} , thereby affecting correspondence loss, and subsequently impact the performance of the model.

In response to the reviewer's comments, we assess the performance of the model using other activation functions, such as LeakyReLU and ELU, instead of ReLU in functions 15 and 16. The results revealed that the choice of activation functions indeed affected the performance of the model. Notably, the extent of this influence varies across different datasets. Considering these findings and aiming for consistency, we have set ReLU as the default activation function in the revised model.

Decision Letter, first revision:

Dear Jinmiao,

Thank you for submitting your revised manuscript "Deciphering spatial domains from spatial multi-omics with SpatialGlue" (N METH-BC52548A). It has now been seen by the original referees and their comments are below. The reviewers find that the paper has improved in revision, and therefore we'll be happy in principle to publish it in Nature Methods, pending minor revisions to satisfy the referees' final requests and to comply with our editorial and formatting guidelines.

We ask that you add the discussion points raised by reviewer 1 to a revised manuscript. Regarding the minor points from referee 2, if these are straightforward to address, we ask that you do so experimentally. If not, please provide some comments in a rebuttal letter. We do ask that you provide a full rebuttal upon resubmission.

TRANSPARENT PEER REVIEW

Please note: we allow redactions to authors' rebuttal and reviewer comments in the interest of confidentiality. If you are concerned about the release of confidential data, please let us know specifically what information you would like to have removed. Please note that we cannot incorporate redactions for any other reasons. Reviewer names will be published in the peer review files if the reviewer signed the comments to authors, or if reviewers explicitly agree to release their name. For more information, please refer to our FAQ page.

ORCID

Sincerely,

Rita

Rita Strack, Ph.D.
Senior Editor
Nature Methods

Reviewer #1 (Remarks to the Author):

Dear authors,

Congratulations on the substantial amount of work that you have done in order to address the reviewers' feedback. And thank you for elaborating on your decisions and your thought process in your response letter.

General Comments

The general feedback on the revised manuscript mainly refers to the discussion, and maybe correction, of the potential shortcomings of the proposed method, which could help the readers to make a more informed decision when choosing a method for their dataset.

1. Both with respect to the potential pitfalls of the presented method and to the question of scalability of the benchmarked methods, it is worth noting that SpatialGlue seems to take a fixed (across modalities) amount of principal components as input in contrast to the methods that work with more information in the data (full feature sets, modality-specific number of components reflecting data complexity, etc.).

Potentially, it can have implications for losing information from modalities with more features when combined with modalities with drastically fewer features. E.g. it might be that 21-22 components, as used in this manuscript, might not be enough to capture the variability of feature-rich modalities (e.g. RNA) while more components cannot be used due to the limited number of features from another modality (e.g. proteins).

2. In addition to reporting the performance of the method for different numbers of principal components, how does running time scale when more components are used as input?

3. Does the requirement of the same amount of components mean that the method implicitly links components between modalities (i.e. PC1 in modality 1 with PC1 in modality 2, etc.)? If this is the case, does it constitute any potential problems?

4. Some methods tested together with SpatialGlue in the revised version of the manuscript have been originally designed to handle specific modalities, such as totalVI for CITE-seq data and MultiVI for scATAC+scRNA-seq. Different modalities imply different assumptions for such models, and evaluating these methods on "foreign" modalities will expectedly result in worse performance. This might be an explanation for vast differences in performance between totalVI and MultiVI on some datasets (e.g.

Figure 3b). If this is the case, it might be worth highlighting it for the reader – or dropping such methods from comparisons on modalities that are “foreign” for them.

5. Given that the adversarial loss has been removed from the revised version of the method, was it dispensable in the first place, or did its removal have to be compensated for with any additional tricks during training?

Specific Comments

1. The figure presenting the simulated data (Figure 1b, left) might not give enough clarity to the reader which spatial patterns are determined by which modality. While it is noted in the text that these modality weights “together contained the information of the ground truth”, it does not clarify the contribution of individual modalities.

2. It might be useful both for the reader and for the sake of reproducibility if the supervised metrics calculated throughout the manuscript (e.g. Figure 1e, Figure 1f, Figure 1k) are also described in more detail (e.g. in the methods section or in the supplementary text) as their calculation seems to be the basis for the method’s performance claims.

3. For the mouse brain dataset, cross-modality weights claims do not always obviously follow from the data shown in Figure 2e. For instance, the distributions of the RNA and protein weights for the ccg-11 cluster do appear quite similar.

4. What does “peak-to-gene links heatmap” mean in the context of Figure 2l? It might be unclear how exactly a set of genes/peaks was identified to produce that panel.

5. While showing difference in distribution, as demonstrated by Figure 3l, Siglec1 and, even more so, Cd209a expression seems to be largely similar across all three clusters in question. Are there any other RNA markers that did or could guide the annotation? How would RNA expression look like when visualised in space (i.e. akin Figure 3k)?

Reviewer #1 (Remarks on code availability):

The code for the SpatialGlue model as well as for the evaluation of other methods is available in the linked repository and is presented in a clean manner.

The notebooks documenting the use of SpatialGlue itself are available in another repository and rendered with ReadTheDocs (<https://spatialglue-tutorials.readthedocs.io/en/latest/>). There are also notebooks that demonstrate how other methods were used for benchmarking in the original repository. Authors might consider putting all the tutorials in one repository, e.g. moving the benchmarking tutorials into the tutorials repository or to a separate repository specifically for the paper itself.

SpatialGlue is a Python package, and as such, it is easy to install it. Authors might consider improving the namespace handling in the package to avoid imports like “from SpatialGlue import SpatialGlue”. The documentation folder common for Python packages is missing as well as the docstrings for a number of methods.

The preprocessing steps for different technologies are hardcoded into the model package itself, which assists the ease of reproducibility for the same datasets but will confuse the users that will run SpatialGlue on other datasets of the same data type.

While the revised manuscript drops the adversarial loss to make the method applicable to multiple (more than two) modalities, the code only operates with two modalities.

Reviewer #2 (Remarks to the Author):

I'd like to first commend the authors on the impressive amount of improvements they have produced in this rebuttal: a substantial expansion of the number of datasets used, a thorough benchmarking on real and simulated datasets, and a more in-depth argumentation of the currently proposed architecture using ablation studies. This has raised the quality of the manuscript significantly, in my opinion. This manuscript describes a methodology highly relevant to a broad community of readers as spatial multi-omics technologies become ever more available (i.e. new technologies, commercial implementations). I look forward to further expansions of the methodology to allow the use of adjacent slices and imaging modalities, but I think the tool in its current state is highly relevant in its current state. I have no remaining major remarks, only smaller questions/remarks which I leave to the authors'/editor's discretion to implement or not:

Minor:

Response 1.3: It's always difficult to judge analyses based on the number of clusters found, as the clustering resolutions/algorithm might differ (and even that is difficult to compare in datasets with different dimensionality/number of datapoints). It would be interesting for the authors to discuss if they are able to find the newly discovered clusters retrospectively, in a more supervised manner, in the original unintegrated datasets. If the power to detect these clusters is severely lacking prior to the integration, this would further strengthen the case for integration.

Response 2.5: I agree that increasing the dimensionality doesn't seem to bring any additional performance to the simulation datasets, as these are by definition more simplistic than real-world data. It would be interesting to show the difference between reduced dimensionality inputs and full data inputs in real datasets as these might contain real non-linearities, even at a higher computational cost.

Reviewer #2 (Remarks on code availability):

The code is available and properly documented, including installation instructions and usage tutorials. I have not been able to extensively test it, but it seems to be usable to the community.

Author Rebuttal, first revision:

Summary of changes

Dear Editor,

We have perused the reviewers' comments and undertaken additional work to address their concerns to improve both the manuscript and the SpatialGlue package. The following summarizes the work performed:

1. **Code and package:** We have significantly revised the SpatialGlue repositories and tutorials. There are now new repositories for the benchmarking tutorial and for integrating of more than two modalities. The preprocessing steps have also been separated to enable user customization. We also renamed the package namespace to avoid confusion and docstrings have been added for all functions in the package.
2. **Investigating the contributions of different modalities towards identifying specific biological structures:** With the mouse brain H3K27ac dataset, we demonstrated that the four layers uncovered by SpatialGlue could not be achieved with only the RNA-seq modality. With increasing the clustering resolution, the cerebral cortex could not be delineated beyond two layers of the original annotation. When examining the DEGs (RNA) and differential peaks (CUT&Tag), we found the RNA modality to possess few DEGs and thus little differentiating power to subdivide the layers, while greater numbers of differential peaks were found between the sub-layers. This highlights the CUT&Tag modality's contribution towards resolving the cortex layers and support the use of SpatialGlue in integrating multiple modalities to analyze spatial multi-omics. The full results are presented in response 2.1.
3. **Exploring the impact of input feature dimensionality:** We performed additional experiments to assess the differences between using dimensionality reduced data versus full data as input. This comparative analysis was performed using three datasets across different tissues, i.e., mouse thymus, mouse spleen, and human lymph node, acquired with different technology platforms. Our evaluation focused on comparing both performance and time complexity. We found that using the full data did not improve performance while increasing runtime. The full results are provided in response 2.2.
4. **Time efficiency:** We evaluated how the running time scales when more components are used as input by benchmarking with a large-scale mouse brain RNA&ATAC dataset. The results are presented in response 1.2.
5. **Supervised metrics:** We have elaborated the computation details of all supervised metrics used and added them to the Supplementary file.
6. **Discussion and clarification:** We have expanded our discussion on baseline methods to include their application scope. Additionally, we have clarified descriptions that were previously identified as confusing to ensure the manuscript's clarity.

In addition, we have provided the point-by-point responses to the comments below, as well as amended the manuscript to reflect the new results and responses. If there are further questions or comments, we will be happy to address them.

Reviewer #1 (Remarks to the Author):

Dear authors,

Congratulations on the substantial amount of work that you have done in order to address the reviewers' feedback. And thank you for elaborating on your decisions and your thought process in your response letter.

Response: Thank you for your kind words and your efforts toward reviewing the manuscript. We have further revised the manuscript and SpatialGlue package based on the current set of comments from both reviewers. In addition to the work performed as described above, we have also clarified the questions posed. Should reviewers have additional suggestion and comments, we are more than happy to provide additional materials and response.

General Comments

The general feedback on the revised manuscript mainly refers to the discussion, and maybe correction, of the potential shortcomings of the proposed method, which could help the readers to make a more informed decision when choosing a method for their dataset.

Comment 1.1: Both with respect to the potential pitfalls of the presented method and to the question of scalability of the benchmarked methods, it is worth noting that SpatialGlue seems to take a fixed (across modalities) amount of principal components as input in contrast to the methods that work with more information in the data (full feature sets, modality-specific number of components reflecting data complexity, etc.).

Potentially, it can have implications for losing information from modalities with more features when combined with modalities with drastically fewer features. E.g. it might be that 21-22 components, as used in this manuscript, might not be enough to capture the variability of feature-rich modalities (e.g. RNA) while more components cannot be used due to the limited number of features from another modality (e.g. proteins).

Response 1.1: Thank you for raising this critical issue. We agree with you that having the same number of reduced data dimensions (via PCA or MLP) for all modalities can be restrictive and potentially cause information loss in some modalities. At present, the modality that is likely to cause a significant restriction is the protein modality due to the low number of protein markers that can be targeted and captured. With current technologies such as SPOTS (Ben-Chetrit et al., 2023), up to 32 proteins can be simultaneously captured. While a larger number of proteins and consequently dimensions is preferred, 32 is not overly restrictive based on our experience with the examples presented. Moreover, we can expect the number of proteins captured to increase with technological advances and consequently eliminate this restriction. In the future, we will upgrade our model to work with full feature sets and modality-specific number of latent dimensions (PCA or MLP-derived embeddings) reflecting respective data modality complexity.

Reference

Ben-Chetrit, N. et al. Integration of whole transcriptome spatial profiling with protein markers. *Nat. Biotechnol.* (2023) doi:10.1038/s41587-022-01536-3.

Comment 1.2: In addition to reporting the performance of the method for different numbers of principal components, how does running time scale when more components are used as input?

Response 1.2: We thank the reviewer for the suggestion. We agree that the impact of the number of input components on runtime performance can be an important consideration. Here, we tested SpatialGlue on a large dataset, the mouse brain RNA-ATAC P22 (9,215 cells), with the number of input principal components varying from 10 to 1,000 (Figure R1). SpatialGlue's time complexity increases slowly with the number of principal components, showing that it scales well when more components are used. We have added these results to Extended Data Fig.1 and discussed this in the manuscript (last paragraph of manuscript).

Figure R1. Evaluating the impact of the number of principal components on the running time of Spatial on the mouse brain RNA-ATAC P22 dataset.

Comment 1.3: Does the requirement of the same amount of components mean that the method implicitly links components between modalities (i.e. PC1 in modality 1 with PC1 in modality 2, etc.)? If this is the case, does it constitute any potential problems?

Response 1.3: Thank you for this question. We agree that using the same number of components for each modality can give the impression that there might be links between the corresponding components of each modality. Currently, within the existing design of SpatialGlue, there is no explicit link between the components of each modality. Any similarity or correspondence within the learned representations arises solely from the nature of the dataset, not the algorithm.

Comment 1.4: Some methods tested together with SpatialGlue in the revised version of the manuscript have been originally designed to handle specific modalities, such as totalVI for CITE-seq data and MultiVI for scATAC+scRNA-seq. Different modalities imply different assumptions for such models and evaluating these methods on “foreign” modalities will expectedly result in worse performance. This might be an explanation for vast differences in performance between totalVI and MultiVI on some datasets (e.g. Figure 3b). If this is the case, it might be worth highlighting it for the reader – or dropping such methods from comparisons on modalities that are “foreign” for them.

Response 1.4: We would like to thank the reviewer for the insightful comments. Indeed, some of the competing methods have been originally designed with specific data modalities in mind and were not tested for others. Therefore, the design restrictions of methods are relevant in explaining their performance during the benchmarks. It is also fairer to only test methods on data types that they are designed to handle. In particular, totalVI and MultiVI have been designed for specific data modalities (totalVI for CITE-seq data and MultiVI for scATAC+scRNA-seq). We have highlighted this in Supplementary file (Table S2) where we listed the applicability of methods to data types. Specifically, for totalVI and MultiVI, we did not test totalVI on the mouse brain epigenome-transcriptome data (spatial ATAC-RNA-seq, and RNA-seq with CUT&Tag (H3K27ac histone modification) datasets) and this restriction was mentioned in the text (the third paragraph of page 4). For MultiVI, we did test it on data with a protein modality as it has been stated in the original publication (third paragraph) that it can handle “surface protein expression with tagged antibodies”. Consequently, we included MultiVI in the comparison. In view of the relatively poor results achieved by MultiVI, we have added discussion

on the possible factors that could have caused the poor performance observed (the paragraph 2 of page 6).

Comment 1.5: Given that the adversarial loss has been removed from the revised version of the method, was it dispensable in the first place, or did its removal have to be compensated for with any additional tricks during training?

Response 1.5: Thank you for your insightful questions regarding the exclusion of adversarial loss from the revised model. The decision to remove adversarial loss was not taken lightly. Initially, the adversarial loss within our model aims at constraining representations from different modalities within a uniform latent space. This constraint would ensure better alignment between a spot's representations across different modalities. Upon rigorous evaluation, we recognized that the adversarial component might constrain the extension of the model from two to three or more modalities while contributing minimally to the model's overall performance. This led to us to reconsider its utility within our framework. Although the removal of adversarial loss resulted in a marginal decrease in performance, we implemented a series of compensatory strategies to mitigate this impact. These measures include revising the learning rate and training duration, fine-tuning the weight factors of various loss components, and optimizing feature preprocessing. These measures enable our model to maintain the performance of its predecessor that utilized adversarial loss.

Specific Comments

Comment 1.6: The figure presenting the simulated data (Figure 1b, left) might not give enough clarity to the reader which spatial patterns are determined by which modality. While it is noted in the text that these modality weights "together contained the information of the ground truth", it does not clarify the contribution of individual modalities.

Response 1.6: Thank you for highlighting this omission. We have added clarification to the main manuscript (second paragraph of page 3) to clearly outline the contribution of each modality towards each factor in the simulated data.

Comment 1.7: It might be useful both for the reader and for the sake of reproducibility if the supervised metrics calculated throughout the manuscript (e.g. Figure 1e, Figure 1f, Figure 1k) are also described in more detail (e.g. in the methods section or in the supplementary text) as their calculation seems to be the basis for the method's performance claims.

Response 1.7: Thank you for your constructive suggestion. According to your suggestion, we have added the details of supervised metrics in the Supplementary file, including Homogeneity, Mutual Information, V-measure, AMI, ARI, and NMI. We provide computational formulas together with definition for each supervised metric.

Comment 1.8: For the mouse brain dataset, cross-modality weights claims do not always obviously follow from the data shown in Figure 2e. For instance, the distributions of the RNA and protein weights for the ccg-11 cluster do appear quite similar.

Response 1.8: Thank you for pointing this out. Indeed, visual inspection of the cross-modality weights and the plotted unimodal data can differ due to their computation. The plotted unimodal clusters were obtained by clustering on the PCA(RNA, protein)/LSI(ATAC) reduced dimension data from individual modalities. In contrast, the cross-modality weights within SpatialGlue are computed based on the latent representations learned by SpatialGlue. Such latent representation of each modality can be enhanced by the information from the other modalities as the information contained in one

modality is used to learn the representation of the other modality. Consequently, the significance (or weight) of specific cellular features (or modalities) in the latent space may be different from that of the original feature modalities, leading to potential discrepancies between the SpatialGlue's learned weights and the plotted clusters of the unimodal data. We have added text in the manuscript to clarify this (first paragraph of page 5).

Comment 1.9: What does “peak-to-gene links heatmap” mean in the context of Figure 2l? It might be unclear how exactly a set of genes/peaks was identified to produce that panel.

Response 1.9: We apologize for the lack of clarity on the terms used. Here, peak-to-gene links refer to the correlations computed between peaks (ATAC-seq modality) and gene expression (RNA-seq modality), as accomplished using the ArchR package (Granji et al, 2021). Specifically, to produce peak-to-gene links heatmap in Figure 2l, we first obtained 62,012 peak-to-gene links (P2GLinks) or correlations between peaks and genes using the ‘addPeak2GeneLinks’ function in ArchR. Figure 2l plots the z-score normalized expression heatmaps of correlated peaks (left) and gene expression (right). The heatmap rows were clustered by k-means clustering for the default 25 clusters. This side-by-side plotting illustrates the correspondence between peaks accessibility and gene expression. We have revised the Supplementary file to make clear the nature of the heatmap plotted (‘Downstream analyses’ section).

Reference

Granja, Jeffrey M., et al. ArchR is a scalable software package for integrative single-cell chromatin accessibility analysis. *Nature genetics* 53.3 (2021): 403-411.

Comment 1.10: While showing difference in distribution, as demonstrated by Figure 3l, Siglec1 and, even more so, Cd209a expression seems to be largely similar across all three clusters in question. Are there any other RNA markers that did or could guide the annotation? How would RNA expression look like when visualised in space (i.e. akin Figure 3k)?

Response 1.10: We thank the reviewer for the questions. We agree that the Siglec1 and Cd209a expression values across the different macrophage clusters are highly similar and are not the best markers in differentiating them. Within literature, Cd209a and Siglec1 (CD169) are the commonly used markers for distinguishing the respective macrophage subsets. They are primarily detected as proteins via assays such as antibody staining. For example, CD169 expression in the protein modality show clear differences between the macrophage subsets (Figure 3j). Conversely in the RNA-seq modality, we agree that their expression values show much smaller differences. We have checked the RNA-seq expression of other subset markers from literature (Satoshi, et al., 2019) but only found Cd209a and Siglec1 to show relevant expression differences. We have also perused the literature for RNA-seq specific markers but regrettably have not been successful. We plotted spatial clustering of marker genes Cd209a and Siglec1 in the mouse spleen sample (Figure R2), but the plots do not reveal clear expression patterns.

Figure R2. Spatial clustering of marker genes Cd209a and Siglec1 in mouse spleen replicate 1 sample.

References

Fujiyama, Satoshi, et al. "Identification and isolation of splenic tissue-resident macrophage sub-populations by flow cytometry." *International Immunology* 31.1 (2019): 51-56.

Reviewer #1 (Remarks on code availability):

The code for the SpatialGlue model as well as for the evaluation of other methods is available in the linked repository and is presented in a clean manner.

The notebooks documenting the use of SpatialGlue itself are available in another repository and rendered with ReadTheDocs (<https://spatialglue-tutorials.readthedocs.io/en/latest/>). There are also notebooks that demonstrate how other methods were used for benchmarking in the original repository. Authors might consider putting all the tutorials in one repository, e.g. moving the benchmarking tutorials into the tutorials repository or to a separate repository specifically for the paper itself.

SpatialGlue is a Python package, and as such, it is easy to install it. Authors might consider improving the namespace handling in the package to avoid imports like "from SpatialGlue import SpatialGlue". The documentation folder common for Python packages is missing as well as the docstrings for a number of methods.

The preprocessing steps for different technologies are hardcoded into the model package itself, which assists the ease of reproducibility for the same datasets but will confuse the users that will run SpatialGlue on other datasets of the same data type.

While the revised manuscript drops the adversarial loss to make the method applicable to multiple (more than two) modalities, the code only operates with two modalities.

Response: Thank you for in-depth comments on improving the code repository and accompanying documentation. Based on your suggestion and comments, we have made the following improvements as follows:

- We have created a separate repository (https://github.com/JinmiaoChenLab/SpatialGlue_notebook) for benchmarking tutorials and added the link into the manuscript.
- We have improved the namespace in the package to avoid confusing and added docstrings for each function in the package. The documentation folder can be found at <https://github.com/JinmiaoChenLab/SpatialGlue>.
- We agree that the preprocessing steps are hardcoded into the package itself and may confuse users. Therefore, we have separated out the preprocessing steps. Using the latest package and tutorials, users can explicitly view the preprocessing steps when they run SpatialGlue on their datasets.
- In response to the suggestion, we have created a new package (SpatialGlue_3M) that can handle three modalities (https://github.com/JinmiaoChenLab/SpatialGlue_3M).

Reviewer #2 (Remarks to the Author):

I'd like to first commend the authors on the impressive amount of improvements they have produced in this rebuttal: a substantial expansion of the number of datasets used, a thorough benchmarking on real and simulated datasets, and a more in-depth argumentation of the currently proposed architecture using ablation studies. This has raised the quality of the manuscript significantly, in my opinion. This manuscript describes a methodology highly relevant to a broad community of readers as spatial multi-omics technologies become ever more available (i.e. new technologies, commercial implementations). I look forward to further expansions of the methodology to allow the use of adjacent slices and imaging modalities, but I think the tool in its current state is highly relevant in its current state. I have no remaining major remarks, only smaller questions/remarks which I leave to the authors'/editor's discretion to implement or not.

Response: Thank you for your highly positive comments on SpatialGlue and the manuscript. Your comments in the review process have been highly valuable to us in improving our work. We are happy to present SpatialGlue and want it to be useful to the larger community. We are also excited to continually develop and expand SpatialGlue in terms of capabilities to handle advances on the experimental front and thus ensure continued relevance.

Minor:

Comment 2.1 (Response 1.3): It's always difficult to judge analyses based on the number of clusters found, as the clustering resolutions/algorithm might differ (and even that is difficult to compare in datasets with different dimensionality/number of datapoints). It would be interesting for the authors to discuss if they are able to find the newly discovered clusters retrospectively, in a more supervised manner, in the original unintegrated datasets. If the power to detect these clusters is severely lacking prior to the integration, this would further strengthen the case for integration.

Response 2.1: Thank you for your insightful suggestion. We acknowledge that it is difficult to judge the analysis based on the number of clusters found. Following your comments, we conducted experiments to explore the potential for discovering new clusters within a single data modality from two perspectives: expanding the numbers of clusters and conducting differential expression gene (DEG) or differential expression peaks (DE peaks) analysis in a supervised manner. For this purpose, we utilized the mouse brain H3K27ac dataset.

Firstly, we attempted to delineate spatial domains within the RNA and H3K27ac data modalities by incrementally increasing the number of clusters from 16 to 20. Figure R3 shows that a mere increase in the number of clusters did not facilitate the discovery of additional distinct structures within the cortical layer in the RNA modality. Similarly, Figure R4 shows that increasing the number of clusters did not enable the genu of corpus callosum (ccg) layer to be identified in the raw H3K27ac data.

Secondly, we performed DEG analysis to evaluate the raw data's power to detect new clusters. On the mouse brain dataset, the original study identified only two layers, namely cluster 6 and 9, which differentiated the cortex into an inner and outer layer (Figure R5 (a), left). On the other hand, the clusters identified by SpatialGlue showcased four cortical layers, captured by clusters 1, 5, 6, 12, with clear boundaries (Figure R5 (b), right). Notably, cluster 6 of the original annotation was subdivided into clusters 5 and 12 by SpatialGlue's clustering, while the original cluster 9 was segmented into clusters 1 and 6. We then performed DEG analysis with the raw RNA data as features and the SpatialGlue's clusters as labels. This analysis aimed to identify differentially expressed genes across clusters, i.e., 'cluster 1 vs cluster 6', 'cluster 5 vs cluster 12', and 'clusters 1 & 6 vs clusters 5 & 12'. The heatmap (Figure R5 (b), up) revealed a scarcity of differentially expressed genes between clusters 1

and 6, as well as between clusters 5 and 12, indicating the original RNA data may lack sufficient power to solely enable the differentiation of these clusters.

Figure R3. Spatial clustering with different number of clusters on the raw RNA data of the mouse brain H3K27ac dataset.

Figure R4. Spatial clustering with different number of clusters on the raw H3K27ac data in the mouse brain H3K27ac dataset.

Meanwhile, the combined group 'clusters 5 & 12 vs clusters 1 & 6' showed a significant number of differentially expressed genes, underscoring that the original RNA data contains sufficient differences to enable the cerebral cortex to be divided into two clusters (clusters 6 and 9 in Figure R5 (a), left). These findings were further validated using volcano plots (Figure R5 (c)), where we observed limited significant genes in both groups 'cluster 1 vs cluster 6' (Figure R5 (c), left) and 'cluster 5 vs cluster 12' (Figure R5 (c), middle), with an imbalance in the quantities of up-regulated and down-regulated genes. In contrast, the combined group displays more statistically significant DEGs (Figure R5 (c) right). These findings highlighted that using only RNA data allowed two clusters within the cortical layer to be identified but fell short in enabling further differentiation. In contrast, Figure R5 (b)(bottom) shows a significant number of differentially expressed peaks between clusters 1 and 6, clusters 5 and 12, and the combined group (clusters 5 & 12 and clusters 1 & 6) in the H3K27ac data, demonstrating the H3K27ac data can better separate these four cortical layers than the RNA data.

Figure R5. Differentially expressed genes (DEG) analysis of the mouse brain H3K27ac dataset. (a) Spatial clustering identified using the RNA data only (left) and the latent integrated representation obtained by SpatialGlue model (right), respectively. (b) Heatmaps of DEGs and DE peaks across clusters. (c) Volcano plots of DEGs across clusters.

Comment 2.2 (Response 2.5): I agree that increasing the dimensionality doesn't seem to bring any additional performance to the simulation datasets, as these are by definition more simplistic than real-world data. It would be interesting to show the difference between reduced dimensionality inputs and full data inputs in real datasets as these might contain real non-linearities, even at a higher computational cost.

Response 2.2: Thank you for your great suggestion. We agree that experimental datasets are more complex than simulation datasets, and might contain real non-linearities, even at a higher computational cost. Based on your suggestions, we tested SpatialGlue on real-world datasets to observe the difference between reduced dimensionality inputs (SpatialGlue) and full data inputs (SpatialGlue-full). Here we employ three RNA & protein datasets, i.e., 10x Genomics human lymph node, Stereo-seq mouse thymus, and SPOTS mouse spleen. All these datasets have obvious differences in the numbers of features across modalities. Figure R6 shows spatial clustering of SpatialGlue and its variant SpatialGlue-full on the human lymph node, mouse thymus, and mouse spleen datasets, respectively. SpatialGlue could better predict cell type or spatial domains (Figure R6 (a) and (b)) and exhibited better preservation of the original features in the latent space than its variant (Figure R6 (c)). To assess the influence of input feature dimensionality on runtime, we compared the time complexity of SpatialGlue and its variant SpatialGlue-full using all three datasets. Figure R6 (d) shows that SpatialGlue-full took comparable or even longer time than SpatialGlue in three datasets, indicating that taking full features as inputs will cause higher time complexity with the model than the reduced dimensionality feature inputs.

Figure R6. Performance comparison between SpatialGlue and its variant SpatialGlue-full on three RNA & protein datasets. (a) Spatial clustering of human lymph node, mouse thymus, and mouse spleen node datasets respectively. (b) Quantitative evaluation of SpatialGlue and its variant with the human lymph node dataset and supervised metrics. (c) Comparison of Jaccard similarity scores of SpatialGlue and its variant in these three datasets. (d) Comparison of time complexity between SpatialGlue and its variant with these three datasets.

Reviewer #2 (Remarks on code availability):

The code is available and properly documented, including installation instructions and usage tutorials. I have not been able to extensively test it, but it seems to be usable to the community.

Response: Thank you for the positive comments on the code and tutorials. We have further updated our repository by refining the model package, including an improved namespace, support for more than two modalities, and enhanced preprocessing steps. We plan to continuously improve SpatialGlue to meet the needs of users and future technological advances in spatial multi-omics.

Final Decision Letter:

Dear Jinmiao,

I am pleased to inform you that your Article, "Deciphering spatial domains from spatial multi-omics with SpatialGlue", has now been accepted for publication in Nature Methods. The received and accepted dates will be May 10, 2023 and May 20, 2024. This note is intended to let you know what to expect from us over the next month or so, and to let you know where to address any further questions.

Over the next few weeks, your paper will be copyedited to ensure that it conforms to Nature Methods style. Once your paper is typeset, you will receive an email with a link to choose the appropriate publishing options for your paper and our Author Services team will be in touch regarding any additional information that may be required. It is extremely important that you let us know now whether you will be difficult to contact over the next month. If this is the case, we ask that you send us the contact information (email, phone and fax) of someone who will be able to check the proofs and deal with any last-minute problems.

Please note that *Nature Methods* is a Transformative Journal (TJ). Authors may publish their research with us through the traditional subscription access route or make their paper immediately open access through payment of an article-processing charge (APC). Authors will not be required to make a final decision about access to their article until it has been accepted. Find out more about Transformative Journals

Authors may need to take specific actions to achieve compliance with funder and institutional open access mandates. If your research is supported by a funder that requires immediate open access (e.g. according to Plan S principles) then you should select the gold OA route, and we will direct you to the compliant route where possible. For authors selecting the subscription publication route, the journal's standard licensing terms will need to be accepted, including self-archiving policies. Those licensing terms will supersede any other terms that the author or any third party may assert apply to any version of the manuscript.

You may wish to make your media relations office aware of your accepted publication, in case they consider it appropriate to organize some internal or external publicity. Once your paper has been scheduled you will receive an email confirming the publication details. This is normally 3-4 working days in advance of publication. If you need additional notice of the date and time of publication,

please let the production team know when you receive the proof of your article to ensure there is sufficient time to coordinate. Further information on our embargo policies can be found here: <https://www.nature.com/authors/policies/embargo.html>

If you are active on Twitter/X, please e-mail me your and your coauthors' handles so that we may tag you when the paper is published.

Best regards,
Rita

Rita Strack, Ph.D.
Senior Editor
Nature Methods